

# Evidence of an Ozone Mini-Hole Structure in the Early Hunga Plume Above the Indian Ocean

Tristan Millet [1], Hassan Bencherif [1], Thierry Portafaix [1], Nelson Bègue [1], Alexandre Baron [2], Valentin Duflot [1,*], Cathy Clerbaux [3,4], Pierre-François Coheur [4], Andrea Pazmino [3], Michaël Sicard [1,5], Jean-Marc Metzger [6], Guillaume Payen [6], Nicolas Marquestaut [6], and Sophie Godin-Beekmann [3]

[1]LACy, Laboratoire de l'Atmosphère et des Cyclones (UMR 8105 CNRS, Université de La Réunion, Météo-France) Saint-Denis de La Réunion, France
[2]Cooperative Institute for Research in Environmental Sciences, and NOAA Chemical Sciences Laboratory, Boulder, USA
[3]LATMOS/IPSL, Sorbonne Université, UVSQ, CNRS, Paris, France
[4]Spectroscopy, Quantum Chemistry and Atmospheric Remote Sensing, Université Libre de Bruxelles (ULB), Brussels, Belgium
[5]CommSensLab-UPC, Universitat Politècnica de Catalunya, Barcelona, Spain
[6]Observatoire des Sciences de l'Univers de La Réunion (OSUR), CNRS/Université de La Réunion/Météo-France, UAR 3365, Saint-Denis, France
[*]now at : Department for Atmospheric and Climate Research, NILU – Norwegian Institute for Air Research, Kjeller, Norway

**Correspondence:** Tristan Millet (tristan.millet@univ-reunion.fr)

**Abstract.** On 15 January 2022, the Hunga volcano (20.5° S, 175.4° E) erupted, releasing significant amounts of aerosols, water vapor ($H_2O$) and a moderate quantity of sulfur dioxide ($SO_2$) into the stratosphere. Due to the general stratospheric circulation of the southern hemisphere, this volcanic plume traveled westward and impacted the Indian Ocean and Reunion (21.1° S, 55.5° E) a few days after the eruption. This study aims to describe current observations of an ozone mini-hole in the

first week following the eruption. The Ozone Mapping and Profiler Suite Limb Profiler (OMPS-LP) aerosol extinction profiles were used to investigate the vertical and latitudinal extension of the volcanic plume over the Indian Ocean. The volcanic aerosol plume was also observed with an aerosol lidar and a sun-photometer located at Reunion. The impact of this plume on stratospheric ozone was then investigated using the Microwave Limb Spectrometer (MLS) and Infrared Atmospheric Sounding Interferometer (IASI) ozone profiles and total ozone maps. Results show that the volcanic plume was observed over Reunion

at altitudes ranging from 26.8 to 29.7 km and spanned more than 20 degrees of latitude on 22 January while over the Indian Ocean. Ozone maps reveal an ozone mini-hole structure, with a maximum Total Column Ozone (TCO) anomaly of -38.97 ± 25.39 DU from IASI on 21 January. The MLS profiles impacted by the Hunga water vapor plume show an average ozone anomaly of -0.43 ppmv with a standard deviation of 0.66 ppmv at the 14.68 hPa pressure level.

## 1  Introduction

Due to its high oxidizing potential and contribution to the radiative budget, ozone plays an undeniable role in the Earth's atmosphere (IPCC, 2013; WMO, 2018; IPCC, 2021). In the stratosphere, ozone serves as a protective shield for the biosphere by absorbing the majority of solar ultraviolet radiation (UVR) in the 280–315 nm range. This shielding action protects ecosys-



tems and human health from the harmful effects of UV-B radiation, which can lead to adverse health issues such as cataracts, melanoma, and skin aging, while deteriorating materials (Pitts et al., 1977; Matsumura and Ananthaswamy, 2004; Bernhard et al., 2020). In the past decades, anthropogenic emission of chlorofluorocarbons (CFCs) and halons (Br) was found to be responsible for the rapid decline in stratospheric ozone (Rowland, 1996). Within the stratosphere, CFCs are indeed photo-dissociated into chlorine compounds which are known to efficiently deplete ozone (Solomon, 1999). Following the ratification of the Montreal Protocol in 1987, CFC emissions were gradually restricted, and previous research and reports show that the ozone layer is expected to return to its 1970s levels from the middle to the end of the century, depending on the latitude (Dhomse et al., 2018; WMO, 2022). On the contrary, tropospheric ozone is a secondary pollutant that directly harms ecosystems, reduces crop productivity, and has negative effects on human health (Mills et al., 2018; Nuvolone et al., 2018). Photochemical formation of tropospheric ozone is driven by the combination of solar radiation and ozone precursors, including Volatile Organic Compounds (VOCs), nitrogen oxides ($NO_x$) and aerosols (Jacob, 1999; Ivatt et al., 2022). Ozone in the troposphere can therefore be enhanced by anthropogenic activities such as agriculture, industry and transport, that release $NO_x$ and aerosols.

Explosive volcanic eruptions can influence both stratospheric and tropospheric ozone concentrations, and thus play a role in global chemistry and radiative forcing (Robock, 2000). Previous major eruptions, such as that of Mount Pinatubo (1991) and El Chichón (1982) are well-documented examples of events that have altered global atmospheric chemistry (Hofmann and Solomon, 1989; Gobbi et al., 1992; McCormick et al., 1995; WMO, 1999; Guo et al., 2004). These eruptions release substantial amounts of aerosols and sulfur dioxide ($SO_2$) which can alter ozone chemistry. The primary impact of volcanic aerosols on ozone is associated with the activation of chlorine compounds on volcanic particles. When $SO_2$ is released during eruptions, it undergoes transformation into sulfuric acid particles ($H_2SO_4$) which can contribute to ozone depletion through heterogeneous chemistry. Studies have highlighted the relationship between $SO_2$ and chlorine in causing ozone decline post-eruption (Tie and Brasseur, 1995). Additionally, reactive anthropogenic chlorine compounds may be enhanced in volcanically perturbed regions, leading to further ozone depletion (Hofmann and Solomon, 1989). Justifiably, McCormick et al. (1995) reported that tropical column ozone decreased by 6–8 % in the months following the Mount Pinatubo eruption. They observed that losses were greatest below 28 km, amounting to 20 % in the 24–25 km altitude range.

Because of the implied ozone losses and radiative forcing anomalies, the injection of volcanic plumes into the stratosphere can also influence atmospheric temperatures. Ramaswamy et al. (2006) observed increases in global lower stratospheric temperatures following the major eruptions of El Chichón and Mount Pinatubo. Moreover, it was determined that the ozone depletion in the aerosol layer caused by the Mount Pinatubo eruption reduced stratospheric heating by 30 % (Kirchner et al., 1999). Despite this reduction, the radiative anomalies caused by the presence of stratospheric aerosols induced a global stratospheric warming of 3-4 K and tropospheric cooling (Stenchikov et al., 1998; Kirchner et al., 1999).

Moderate and major eruptions may also contribute to the amplitude and dimension of the ozone hole over Antarctica. Following the Mount Pinatubo eruption, Hofmann and Oltmans (1993) observed unusually low total ozone values of 105 DU over the South Pole Station. This deeper ozone hole was attributed to enhanced PSCs volume driven by extra stratospheric sulfuric acid availability, offering more surface for halogen-ozone reactions. Ivy et al. (2017) reported an increase of the 2015 Antarctic ozone hole by $4.5\times10^6$ km$^2$, primarily attributed to volcanic aerosols from the Calbuco eruption. Similarly, Zhu



et al. (2018) reported volcanic sulfate aerosols penetration from the Calbuco eruption into the Antarctic polar vortex, resulting in earlier ozone loss and an increase in the area of the ozone hole. Yook et al. (2022) also hypothesized a link between the eruption of La Soufrière in 2021 and the longevity of the 2021 ozone hole. Hence, numerous research papers focused on ozone chemistry and atmospheric forcings following eruption events.

This study focuses on the January 2022 Hunga eruption with a particular emphasis on the ozone-related data in the week following the eruption. The main eruption likely released more energy than the 1991 Mount Pinatubo eruption and caused the largest stratospheric aerosol disturbance since that event (Wright et al., 2022; Sellitto et al., 2022). Its consequences have been under intensive scrutiny and studies revealed it injected $\sim$0.5 Tg of $SO_2$ and $146 \pm 5$ Tg of water vapor ($H_2O$) into the stratosphere, corresponding to an increase of $\sim$10 % of the global stratospheric $H_2O$ burden (Sellitto et al., 2022; Zuo et al., 2022; Millán et al., 2022). The main eruption's aerosol column extended through the troposphere and stratosphere, and even reached the lower mesosphere (Carr et al., 2022). As a result of the main austral summer stratospheric circulation and the prevalent phase of the QBO, the first signs of the Hunga aerosol plume's passage over Reunion were noticed only 4 days after the main eruption (Baron et al., 2023; Legras et al., 2022). Evan et al. (2023) and Zhu et al. (2023) attribute the initial low ozone levels observed with satellite data to the lofting of ozone-poor tropospheric air masses. However, the subsequent ozone depletion observed in the following days is attributed to chemical processes. Specifically, they highlight the role of heterogeneous chlorine activation on humidified volcanic aerosols and gas-phase ozone-depleting reactions. The significant increase in stratospheric humidity and the resulting radiative cooling (up to 2-6 K, as reported by Zhu et al. (2023)) facilitated the rapid conversion of $SO_2$ to sulfate aerosols in less than two weeks (Legras et al., 2022; Asher et al., 2023). This increase in aerosol surface area likely accelerated heterogeneous chlorine activation on sulfate aerosols and led to notable ozone depletion despite elevated non-polar temperatures (Evan et al., 2023; Zhu et al., 2023). While heterogeneous reactions played a crucial role in ozone loss, Zhu et al. (2023) also emphasized the importance of gas-phase reactions. Balloon-borne observations performed at Reunion on 22 January by Evan et al. (2023) justifiably provided evidence of chlorine activation within the volcanic plume. In fact, Zhu et al. (2023) and Evan et al. (2023) identified key gas-phase mechanisms contributing to ozone loss: photolysis of $Cl_2$ followed by its reaction with ozone to form ClO, enhanced $HO_x$ cycle activity due to high $H_2O$ concentrations, strengthened interactions between the $HO_x$ and $ClO_x$ cycles, and the slowing down of the $NO_x$ cycle. Thus, Evan et al. (2023) showed that 5 % of stratospheric ozone was depleted in the week following the eruption over the Indian Ocean, with the most significant losses during periods of peak stratospheric humidification.

In the first two weeks following the eruption, the volcanic aerosol and water vapor plumes caused a stratospheric cooling because water vapor radiative cooling dominated local stratospheric heating rates (Sellitto et al., 2022; Legras et al., 2022; Wang et al., 2022). After the conversion of the initial $SO_2$ into sulfates, the remaining aerosol plume consisted of two concentrated patches (Legras et al., 2022). The dilution and dispersion of the water vapor and aerosol plumes lead to a net warming of the climate system and persisting low temperatures within the stratosphere (Sellitto et al., 2022; Coy et al., 2022). During the first month following the eruption, the aerosol and water vapor plumes mostly overlapped and coincided in altitude ($\sim$ 26 km or $\sim$ 20 hPa) (Schoeberl et al., 2022; Coy et al., 2022). The aerosol plume circumnavigated the Earth within a single week (Khaykin et al., 2022) and diluted pole-to-pole within three months (Taha et al., 2022). Over subsequent months, the aerosol





plume slowly decreased in altitude because of gravitational settling (Legras et al., 2022), whereas the altitude of the water vapor plume was found to increase because of diabatic transport from the Brewer-Dobson circulation (Legras et al., 2022; Schoeberl et al., 2022; Sicard et al., 2024). Using satellite data, Fleming et al. (2024) specified that the global radiative impacts were strongest during the period from March to June 2022, with a warming of ∼1 K in the lower stratosphere and a cooling of ∼3 K in the mid-stratosphere. Over the Indian Ocean basin, Sicard et al. (2024) showed that aerosols and water vapor caused a net cooling (-0.54 ± 0.29 Wm$^{-2}$) on the Earth's radiation budget in the thirteen months after the eruption.

The present paper describes the currently available observations from the Microwave Limb Sounder (MLS) (Waters et al., 2006; Livesey et al., 2008) and the Infrared Atmospheric Sounding Interferometer (IASI) (Aires et al., 2002; Blumstein et al., 2004) as well as ground-based data available at Reunion. The objective of the present manuscript can be summarized in two main points: firstly, we describe the current ozone observations and demonstrate the appearance of a structure resembling an ozone mini-hole; and secondly, we show the zonal displacement of the volcanic aerosol and H$_2$O plumes and the dynamics of its advection with the help of satellite data and analyses.

This article is organized as follows: Sect. 2 describes the instruments and observations that were used for the present study, as well as the data processing and methodology. Section 3 presents the results regarding the volcanic aerosol plume, the ozone and H$_2$O data over the Indian Ocean as well as the dynamics of the advection. Finally, the conclusion in Sect. 4 summarizes the results of this research.

## 2 Instrumentation and Method

In this study, we combined ground-based and satellite observational data to investigate the impacts on ozone before and after the Hunga eruption. Additionally, we used numerical assimilation data for the dynamical aspect. This section outlines the different types of data used in our analysis.

### 2.1 Ozone measurements

A stratospheric DIfferential Absorption Lidar (DIAL) has been operated since January 2013 at the Reunion Atmospheric Physics Observatory (OPAR, 2160 m asl) (Baray et al., 2013; Portafaix et al., 2015). Lidar observations have the advantage of providing high temporal and vertical resolutions (Pazmiño, 2006), with an accuracy of ∼5 % below 20 km, ∼3 % in the 20–30 km altitude range and 15–30 % above 45 km (Godin-Beekmann et al., 2003). This instrument can retrieve ozone concentration profiles at altitudes ranging from 15 to 45 km. However, although the Maïdo DIAL system recorded data during the initial passage of the volcanic plume in January 2022, the corresponding signal-to-noise ratio (SNR) was extremely low and the ozone profiles were not reliable. As a result, stratospheric DIAL ozone profiles used in this paper were recorded before the Hunga eruption, from January 2013 to December 2021. The 470 ozone profiles obtained during this period were used to determine the background ozone level and were compared with profiles from satellites. As part of the Network for the Detection of Atmospheric Composition Change (NDACC), this DIAL data can be accessed at the following link: https://ndacc.larc.nasa.gov/ (last accessed on 23 January 2024).



The Système d'Analyse par Observation Zénithale (SAOZ), an instrument also integrated into the NDACC, is a ground-based spectrometer which measures the sunlight scattered from the zenith sky within the 300 to 650 nm range (Pommereau and Goutail, 1988). Differential Optical Absorption Spectrometry (DOAS) is utilized to analyze observations, enabling the retrieval of daily ozone and nitrogen columns at sunrise and sunset with a total accuracy of 6 % and 14 %, respectively (Boynard et al., 2018). Operating at an altitude of 80 m asl in Saint-Denis, Reunion, since 1993, a SAOZ instrument has provided Total Column

Ozone (TCO) observations at this subtropical site for over three decades. Unfortunately, SAOZ data during the passage of the aerosol plume over Reunion are unreliable because of an unrealistic representation of the Air Mass Factor (AMF), leading to biased TCO retrievals. Consequently, SAOZ data for January 2022 were excluded. However, data outside this time period and climatological values of TCO for the month of January (262.35 ± 4.02 DU) were kept to illustrate the background January ozone TCO. The SAOZ data used in this work can be downloaded from this website: http://saoz.obs.uvsq.fr/ (last accessed on

23 January 2024).

In this study, satellite observations of ozone profiles and TCO were used in complement to ground-based data, offering a global coverage and a consistent measurement frequency. The MLS instrument is a radiometer on the Aura satellite, launched in July 2004. The Aura satellite follows a helio-synchronous orbit and passes the equator at 01:45 pm solar time on its ascending node. In order to calculate atmospheric parameters like temperature and atmospheric component concentrations, MLS measures

thermal radiation emitted from the Earth's atmospheric limb ahead of its orbital path at spectral wavelengths ranging from 0.12 to 2.5 mm (Waters et al., 2006; Livesey et al., 2008). According to Millán et al. (2022), observations close to the Hunga plume should be studied using MLS data at level 2 and version 4 (v4), instead of the latest version (v5). Indeed, MLS v4 relies only on profile retrievals from $O_2$ signals whereas v5 also uses the $H_2O$ line. This inclusion may degrade results in regions of enhanced humidity, which are common in our study. Additionally, their study indicates that the quality of ozone and

temperature measurements are not affected by the aerosol plume (Millán et al., 2022).

Following these recommendations, the MLS data for January 2022 are sourced exclusively from level 2 v4 measurements (Livesey et al., 2020). MLS profiles influenced by the Hunga eruption were selected based on a criterion from Evan et al. (2023). Locations showing v4 water vapor profiles with mixing ratio values exceeding 100 ppmv within the 10 to 100 hPa range were identified as being impacted by the Hunga eruption. Applying this criterion resulted in a total of 113 ozone and water vapor

profiles between 15 and 23 January. To evaluate the similarity between v4 and v5 MLS ozone profiles during unperturbed conditions, we calculated the differences for the remaining 2190 co-located v4 and v5 ozone profiles that did not meet the criterion. The maximum $1\sigma$ standard deviation found in the stratosphere was close to 0.05 ppmv, demonstrating the similarity of these two versions during background conditions. To compare the 113 impacted profiles to background profiles obtained under unperturbed conditions, we employed MLS level 2 v5 data (Livesey et al., 2022). All ozone and water vapor profiles

obtained within a 5-degree radius of each January 2022 impacted profiles were collected. This procedure was undertaken for the months of January from 2013 to 2021 to derive the monthly averaged background profiles. This specific time period was chosen to align with the lidar time series. This procedure was repeated for both ascending and descending nodes on each measuring day at the locations of profiles meeting the established criterion. In most of the stratosphere, specifically between 1 and 68 hPa, MLS ozone volume mixing ratio profiles have accuracy and precision that are both lower than 10 % (Livesey et al.,



2022). In accordance with the recommendations made in the MLS data quality and description documents, all quality flags (quality, convergence, status and precision) were used on the raw profiles, and data lying outside the recommended range (261 to 0.001 hPa, or approximately 11 to 90 km) were not used (Livesey et al., 2022, 2020). MLS observations can be accessed through NASA's data portal (https://disc.gsfc.nasa.gov/, last accessed on 23 January 2024).

IASI is a Fourier Transform spectrometer installed on the three Metop satellites (Clerbaux et al., 2009; Coheur et al., 2009).
This instrument retrieves ozone profiles by analyzing day and night nadir radiances within the thermal infrared spectrum from 6.62 to 15.5 $\mu$m. In the present study, we used data obtained from the Fast-Optimal Retrievals on Layers for IASI (FORLI-O3) ozone products (Hurtmans et al., 2012), which have been extensively validated (Boynard et al., 2018). Specifically, to study the impact of the Hunga eruption on ozone levels in January 2022, we employed a combination of daily TCO observations from IASI instruments onboard Metop-B and Metop-C, operational since 2013 and 2019, respectively. To obtain average TCO maps
during unperturbed conditions, we used monthly TCO data exclusively from IASI on Metop-B. Given that IASI on Metop-B has been providing measurements since March 2013, we used the average of TCO maps spanning from January 2014 to 2021 as a representative of ozone background. Unlike daily TCO, monthly TCO data points from IASI are re-sampled to be distributed on a regular grid. Therefore, to compute anomalies, we performed a re-sampling of daily data to align with the monthly grid. At each grid location, the nearest daily IASI TCO observation within a 0.5° radius was interpolated. If the closest observations
lie beyond this radius limit, then no value was kept for this grid point. Consequently, TCO anomalies from IASI represent the difference between the background ozone levels (from monthly data) and a re-sampled combination of Metop-B and Metop-C daily data during the Hunga event. To observe the spatial correlation between the ozone and water vapor anomaly with the aerosol and $SO_2$ plume, we also employed daily $SO_2$ observations from Metop-B and Metop-C (Clarisse et al., 2012, 2014). The IASI products employed in this work can be accessed on the AERIS platform: https://iasi.aeris-data.fr (last accessed on
23 January 2024).

## 2.2  Aerosol measurements

In addition to the DIAL system, the OPAR is equipped with several other active remote sensing systems, including a Rayleigh-Mie lidar for aerosol profile measurements (Baron et al., 2023). In this study we used aerosol extinction profiles together with the corresponding stratospheric Aerosol Optical Depth (sAOD) at 532 nm as derived from the Rayleigh-Mie lidar
measurements at the Reunion observatory. The data used in this study are publicly accessible via this webpage: https://geosur.osureunion.fr/geonetwork/srv/eng/catalog.search#/metadata/f2c35798-47b7-433c-8927-46cf7babca83. The L2 ready-to-use data set in netCDF format can be accessed from Baron (2023) (last accessed on 23 January 2024).

Aerosol optical properties can also be retrieved using sun-photometers. These remote sensing sun-tracking radiometers perform regular and frequent measurements of the direct solar spectral irradiance, typically at wavelengths between 340 and
1640 nm. By comparing the ground solar irradiance to the estimated top of the atmosphere irradiance, they can determine the total AOD, a quantity that describes the opacity of the atmosphere to radiation. Therefore, a sun-photometer gives a measure of aerosol abundance in the atmospheric column above the study site. In the present study, we used AOD data from a Cimel sun-photometer located in Saint-Denis campus, which has been operating since December 2003 in the framework of the AErosol



RObotic NETwork (AERONET) program. We used level 2.0 v3 AERONET data for the period from December 2003 to January
2022. AERONET data of level 2.0 is quality-controlled with near-real time automatic cloud-screening in addition to having
pre- and post-field calibrations. According to Giles et al. (2019), the $1\sigma$ uncertainty for the near-real time AERONET AOD
measurement is up to 0.02. AERONET data are accessible from https://aeronet.gsfc.nasa.gov/ (last accessed on 23 January
2024).

The Ozone Mapping and Profiler Suite Limb Profiler (OMPS-LP) monitors the Earth limb ahead of its orbit path to pro-
vide high vertical resolution ozone and aerosol profiles. The instrument measures limb scattering radiances in the 290–1000
nm wavelength range over the sunlit portion of the atmosphere using three vertical slits. This instrument has been making
observations onboard the Suomi National Polar-orbiting Partnership (Suomi NPP) spacecraft since January 2012, following
a helio-synchronous orbit with an equatorial passing time of 01:30 pm solar time on its ascending node. With the goal to
study the spatial extension of the plume, we used OMPS-LP aerosol extinction profiles at 745 nm. According to Taha et al.
(2021), extinction coefficients at 745 nm have relative accuracy and precision of 10 % and 15 %, respectively. OMPS data were
downloaded from the following link: https://ozoneaq.gsfc.nasa.gov/ (last accessed on 05 March 2024).

## 2.3 Reanalyses and model

To investigate the origin of the air-masses in our study region, we used the HYbrid Single Particle Lagrangian Integrated
Trajectory (HYSPLIT) model in its passive and backward mode (Draxler and Hess, 1997, 1998). Developed by the National
Oceanic and Atmospheric Administration (NOAA), this model uses meteorological fields to compute and simulate trajectories
of air-masses. Because the long-lasting Hunga atmospheric effects appear to be concentrated within the stratosphere, and since
stratospheric circulation is stable and stratified (Sellitto et al., 2022; Zuo et al., 2022; Millán et al., 2022), we used a single
HYSPLIT simulation to highlight the trajectories of air-masses in the stratosphere over the Indian Ocean. Thus, using meteo-
rological fields from the Global Data Assimilation System (GDAS), we ran a 240 hours back-trajectory simulation of 9 distinct
air parcels with terminal altitudes distributed equitably between 22 and 26 km (National Oceanic and Atmospheric Administra-
tion (NOAA), 2023). These trajectories were chosen to have their endpoint at the location of Saint-Denis, Reunion. HYSPLIT
trajectories can be obtained by running simulations through the following link: https://www.ready.noaa.gov/HYSPLIT_traj.php
(last accessed on 23 January 2024).

Additionally, to examine dynamical processes in the stratosphere during the advection of the volcanic plume, we used ERA5
analyses of Ertel's Potential Vorticity (EPV) at the 600 K isentropic level ($\sim$24 km in altitude, i.e. the altitude of maximum
tropical ozone) for the period from 15 to 23 January, in conjunction with GDAS data (utilized for driving HYSPLIT). Daily
maps of EPV were obtained by averaging instantaneous maps. According to Hoskins et al. (1985), the EPV on isentropic
surfaces behaves as a dynamical tracer when diabatic effects are absent. Many authors have demonstrated its utility in studying
isentropic transport in the stratosphere (Holton et al., 1995; Bencherif et al., 2003; Semane et al., 2006; Bencherif et al., 2011).
EPV maps on the 600 K isentropic level were downloaded from ECMWF's data archive (MARS).

EPV maps over the study area were analysed further using the DYnamical Barrier Localization (DYBAL) algorithm, which
allows the localization of the subtropical barrier (Portafaix et al., 2003). The detection of a dynamical barrier is based on the



EPV gradient in equivalent latitude coordinates as defined by Nakamura (1996), with its position characterized by a local maximum of the EPV gradient (Nash et al., 1996; Manney et al., 2022, 2023). Here we applied the DYBAL code to maps extracted from the ERA5 EPV fields on the 600 K isentropic surface. The ability of DYBAL to detect the position and the deformation of the dynamical barriers was previously highlighted by several studies (Portafaix et al., 2003; Morel et al., 2005; Bencherif et al., 2007).

### 2.4 Inter-comparison

Prior to drawing any conclusions based on the MLS ozone profiles, it is essential to verify their agreement with precise local lidar observations during unperturbed conditions. For this inter-comparison process, we determined daily MLS ozone profiles by averaging all recovered profiles within a 5-degree region around the lidar site, setting the inter-comparison radius to a maximum of 5°. We used MLS ozone profiles obtained from both ascending and descending Aura orbits, with acquisition time near Reunion around 10:15 or 21:45 UTC, respectively. On the other hand, the 470 ground-based DIAL lidar profiles are only nocturnal (recorded at Reunion, i.e. approximately between 16:00 and 01:00 UTC, averaging around 18:30 UTC). Thus, the maximum temporal difference between MLS and lidar profiles is approximately 8 hours. Despite the non-overlapping acquisition times, we compared DIAL night profiles to daily MLS profiles. Although we obtained 470 DIAL profiles, the 5° inter-comparison radius limits the number of available MLS profiles, allowing inter-comparison on a total of 340 days. As a result, the profile comparison is based on the following formula:

$$\text{Relative}_{\text{bias}}(z) = 100 \times \frac{O_{3\ \text{MLS}} - O_{3\ \text{DIAL}}(z)(z)}{O_{3\ \text{DIAL}}(z)}, \tag{1}$$

where $O_{3\ \text{MLS}}(z)$ represents the MLS ozone value at an altitude $z$ and $O_{3\ \text{DIAL}}(z)$ represents the stratospheric DIAL ozone value at the same altitude. Other statistical quantities were also determined, namely the number of profiles (N), the p-value, the coefficient of correlation (r), the linear regression (in the form $y = ax$) and the relative Root-Mean-Square Dispersion (RMSD) (see Appendix A). These statistical quantities were used to assess the differences and similarities between different ozone data at different layers.

Additionally, to compare IASI data with SAOZ measurements recovered at Reunion under unperturbed conditions, we derived a daily TCO time series from Metop-B at Reunion, spanning March 2013 to December 2021. The inter-comparison utilized all data points from both datasets within this time period, irrespective of date and time, including all sunrise and sunset measurements.

## 3 Results and discussion

### 3.1 Aerosol plume

The Hunga main eruption occurred on 15 January 2022 and ejected a large quantity of $H_2O$ and a moderate amount of $SO_2$ into the stratosphere (Khaykin et al., 2022; Sellitto et al., 2022; Zuo et al., 2022; Millán et al., 2022). Following the austral





summer's general stratospheric circulation, the volcanic plume then traveled westward and reached the Indian Ocean and the African continent within days (Baron et al., 2023). The aerosol plume's transport across the Indian Ocean was captured by

OMPS aerosol extinction profiles. Panels (b) to (e) of Fig. 1 present OMPS extinction coefficient profiles at 745 nm over different locations of the Indian Ocean as a function of latitude and altitude during the passage of the volcanic plume. Panel (a) presents the background aerosol distribution obtained prior to the plume's arrival over the Indian Ocean. At the bottom left of each panel are given the date and time of retrieval, and the black dots correspond to the instrument's estimation of the tropopause height. Panel (f) traces the satellite tracks corresponding to data in panels (a) to (e). Thus, this figure describes

the latitudinal and vertical extent of the volcanic plume as observed by the satellite instrument during its passage over the Indian Ocean on 22 January, the date when ozone impacts at Reunion were considered highest (Evan et al., 2023). During unperturbed conditions (see Fig. 1a), the aerosol distribution shows that the largest values of the extinction coefficient are kept below the tropopause. Aerosol presence in the stratosphere is negligible compared to the troposphere. However, the presence of the volcanic plume becomes clearly visible on the other panels, where large extinction coefficient values ($> 10^{-3}$) lie above

the tropopause level and become comparable to those typically observed in the upper troposphere. On 22 January (Fig. 1b to 1e), the volcanic plume is clearly visible in the stratosphere over the Indian Ocean between 5° S and 25° S, reaching altitudes greater than 35 km. Note that this result only characterizes the vertical and latitudinal extent of the volcanic plume, but it does not describe the longitudinal dimension of the plume. Equivalent observations can also be obtained for 21 January (not shown). Similar results were found by Taha et al. (2022) as they outlined the presence of a volcanic plume located at an altitude

exceeding 36 km. Additionally, they reported that the high sensitivity of OMPS LP enabled to monitor the volcanic plume at altitudes above 36 km for a duration of up to 90 days.

Figure 2 shows the Hunga aerosol plume as seen by two quasi-colocalized instruments operating at the Maïdo observatory (lidar) and the Saint-Denis campus (sun-photometer). It is important to highlight that the two instruments are 20 km apart with an approximately 2000 m difference in elevation. Even though the AOD measured by the sun-photometer cannot be

directly compared to the sAOD recorded by the lidar instrument, both sets of observations hold significant information about the passage of the volcanic plume. Figure 2a depicts the evolution of the lidar aerosol extinction profiles at 532 nm between 21 and 23 January, and Fig. 2b shows the evolution of the lidar sAOD at 532 nm (in black) and sun-photometer level 2.0 total AOD (in red) at 532 nm for the second half of January 2022 with their respective uncertainties. The sun-photometer AOD at 532 nm was obtained from the conversion of the AOD at 675 nm using the Angström exponent measurements between 440

and 675 nm. The blue line represents the multi-year average of the sun-photometer level 2.0 data calculated from 2003 to 2021, and the shaded blue region is the corresponding $\pm 1\sigma$ (standard deviation). This multi-year average represents an average of AOD data which is grouped into months, irrespective of the years. Note that different horizontal axes are used for panels (a) and (b), and the common observation periods are shown in gray in both panels.

Results show that the maximum total and stratospheric optical depths recorded by both instruments in January 2022 are very

high in comparison to the multi-year mean AOD of $0.05 \pm 0.02$. This is expected, as Reunion is a pristine region where January usually experiences low AOD levels (Duflot et al., 2022). After 20 January, total AOD values start to dramatically increase until 23 January, when they culminated at $0.57 \pm 0.02$ before gradually decreasing to return to background levels. Similarly to the





sun-photometer measurements, the Maïdo lidar reveals a large amount of aerosols after 21 January, with sAOD values rising up to $0.84 \pm 0.13$. A significant aerosol layer was seen by the lidar on two consecutive nights at altitudes of 29.7 km and 26.8 km,

with maximum extinction coefficients of $0.53 \pm 0.08$ and $0.68 \pm 0.06$, respectively. Note that sun-photometer measurements are obtained during the day, while lidar observations are only performed during nighttime. As such, observations from these two instruments cannot overlap as they do not operate simultaneously. A detailed study of the lidar observation of the Hunga plume can be found in Baron et al. (2023). Our results support their research, suggesting that the bulk of the Hunga aerosol plume passed over Reunion from 21 to 23 January.

### 295  3.2  Maïdo DIAL ozone profiles

Figure 3 shows the multi-year Maïdo DIAL ozone profiles. The black line depicts the altitude of the ozone maximum. A total of 470 profiles were obtained during the period from January 2013 to December 2021. The figure shows an ozone layer that is located between 22 and 27 km, and highlights the variation of the vertical distribution of ozone at Reunion. With a predominant annual cycle, the ozone maximum is at its highest altitude during austral summer (in December at 26.3 km), and at its lowest

altitude during austral winter (in August at 23.7 km). This behavior, observed in subtropical (e.g., Reunion which is located at the edge of the tropical barrier in the stratosphere) and tropical locations, is primarily attributed to dynamical processes. Notably, tropical upwelling, as part of the Brewer-Dobson Circulation (BDC), transports ozone from the equator (where it is primarily produced) to higher latitudes (Butchart, 2014; Plumb and Eluszkiewicz, 1999; Weber et al., 2011). During austral summer (winter), Reunion is closest to the ascending (descending) branch of the BDC, which explains why the ozone layer is

highest (lowest) in altitude.

### 3.3  Inter-comparison results

Prior to obtaining results relative to ozone measurements and stratospheric transport, we conducted a statistical analysis to evaluate the differences between lidar and MLS observations, and between IASI and SAOZ data, during unperturbed conditions. Thus, we compared MLS v5 ozone concentration profiles obtained over Reunion to the Reunion stratospheric DIAL ozone

concentration profiles from January 2013 to December 2021, as well as SAOZ TCO to IASI TCO from March 2013 to December 2021. Results are presented in Fig. 4. The continuous line and the shaded area in Fig. 4a represent the mean relative bias and the standard error, respectively. This standard error represents standard deviation divided by the square root of the number of individual comparisons (which varies as a function of altitude). The aforementioned statistical quantities are also shown in the figure. These mean relative bias profiles were obtained by averaging the relative bias values as derived from Eq. (1) across

all available ozone profiles. Statistical results (correlation coefficient, linear regression and relative RMSD) presented in the following paragraphs were obtained from the comparison of all data points, irrespective of the altitude level, date and time.

Concerning ozone profiles, the best agreements are found in the 20–40 km altitude range, with higher and increasing deviations below 20 km and above 40 km. In the altitude range from 20 to 40 km, MLS has a relative bias and error (with respect to DIAL measurements) of $1.22 \pm 0.37$ %. In this altitude range, the standard error is low because of the high number of

available comparison profiles (up to a maximum of 340). Above 40 km, the bias decreases to $-3.73 \pm 2.55$ %, whereas below





20 km, it shows an average of $0.06 \pm 2.30$ %. There is a positive bias that appears at $\sim$20 km of altitude, with $11.22 \pm 1.67$ %. The increased difference and error at altitudes greater than 40 km is partly due to the lidar SNR decrease and the reduced number of lidar profiles reaching altitudes greater than 45 km. The decrease in SNR requires additional signal filtering, which introduces a bias (Godin et al., 1999). Consequently, the lidar mean measurement error increases from $\sim$ 10 % at 40 km to $\sim$

50 % at 47.5 km. Additionally, out of the 470 lidar profiles, 410 reached 40 km, 132 reached 45 km and only 6 reached 47.5 km. Note also that the increased difference and error at altitudes lower than 20 km may be due to the reduced satellite accuracy and precision (see Table 3.18.1 of Livesey et al. (2022)) and the lower number of lidar profiles for these altitudes. Indeed, out of the 470 profiles, 453 start below 20 km, 409 before 17.5 km and 131 before 15 km. For these reasons, the MLS mean bias profile seems to under-estimate ozone concentrations by $20.73 \pm 1.89$ % at 16.70 km. Over the whole altitude range, the

correlation coefficient (r = 0.99) indicates an excellent correlation between the lidar and MLS, and the linear regression ($y = 1.01\ x$) shows that MLS profiles tend to slightly over-estimate ozone concentrations, irrespective of the altitude. Finally, a low relative dispersion (RMSD = 7.56 %) further demonstrates the agreement between MLS and the DIAL profiles.

Concerning TCO data, a high number of comparison points (N = 5619) indicates a very low relative dispersion (RMSD = 3.26 %) and an elevated correlation (r = 0.87) between IASI and SAOZ datasets. The linear regression ($y = 1.02\ x$) shows that

IASI TCO tends to slightly over-estimate SAOZ TCO.

Therefore, the MLS ozone concentration profiles seem to be in good agreement with lidar observations in the 20–40 km altitude range, which includes the altitudes of the Hunga volcanic plume (26–30 km) being our main focus in this study. The IASI and SAOZ TCO also exhibit low dispersion and a high degree of correlation throughout the comparison period.

### 3.4    Effects of the volcanic plume on ozone

Based on the excellent correlation and agreement between satellite (MLS and IASI) and ground-based instruments (stratospheric lidar and SAOZ) over Reunion, it appears relevant to use satellite ozone products to investigate the changes in the distribution of ozone over the study region.

Figure 5 shows snapshots of the evolution of TCO reduction, in correlation with the $SO_2$ plume, following the passage of the volcanic plume over the Indian Ocean from 15 to 23 January. It depicts daily maps of TCO (panels (a1) to (a9)), TCO

anomalies (panels (b1) to (b9)) and total $SO_2$ (panels (c1) to (c9)). All maps are overlaid with blue contours of the $SO_2$ plume, indicating regions where the $SO_2$ total column is greater than 30 DU. $SO_2$ maps are complemented by the MLS satellite track (light blue circles) comprising the MLS profiles with high $H_2O$ values which met the criterion selection (green circles). The successive locations of the $SO_2$ plume and the impacted MLS profiles (representing the $H_2O$ anomaly) highlight an east-to-west displacement of the $H_2O$ and $SO_2$ plumes. The convergence of the $SO_2$ and $H_2O$ plumes supports previous studies,

and the rapid disappearance of the high $SO_2$ anomaly indicates its rapid conversion into sulfates under the influence of $H_2O$ (Legras et al., 2022; Schoeberl et al., 2022).

This zonal movement is also clearly visible on TCO and TCO anomalies from IASI, highlighting a correlation between ozone, $H_2O$ and $SO_2$. The first appearing important negative ozone anomaly linked to the Hunga appears on 16 January at $\sim$160 °E, with $-31.52 \pm 22.75$ DU, where minimum TCO values are $222.87 \pm 27.58$ DU. This first ozone anomaly is attributed





to the lofting of ozone-poor tropospheric air-masses (Zhu et al., 2023; Evan et al., 2023). TCO anomalies are not retrieved over Australia on 18 January because of the presence of clouds. The anomaly then reappears on the western Australian coast on 19 January at ∼115 °E, with -23.02 ± 22.94 DU and minimum TCO values of 234.92 ± 26.30 DU. This anomaly then appears to grow larger in size and amplitude as the $SO_2$ plume spreads, reaching Reunion on 21 January. The rapid conversion of $SO_2$ molecules to sulfate particles in the first days following the eruption increased the aerosol surface area, resulting in ozone depletion through heterogeneous chemistry (Zhu et al., 2023; Evan et al., 2023). Consequently, the reduction of the $SO_2$ anomaly correlates with the increase in ozone anomaly. Thus, on 21 January, IASI recorded a minimum TCO value of 214.22 ± 25.62 DU and a maximum TCO anomaly of -38.97 ± 25.39 DU. In comparison to the SAOZ climatological TCO for the month of January (262.35 ± 4.02 DU), this IASI TCO anomaly lies more than 10 times below the usual variability, representing a TCO reduction of ∼ 18 %. The IASI anomaly map for 21 January suggests the appearance of a structure resembling an ozone mini-hole extending over approximately 30 degrees of longitude and latitude. The presence of clouds on 22 January hindered the retrieval of IASI data between Reunion and Madagascar, but large anomalies were still visible in the region on 23 January. At this date, IASI recorded a minimum TCO value of 227.29 ± 24.84 DU and a maximum TCO anomaly of -37.69 ± 24.84 DU. The ozone anomaly then exited the Indian Ocean (not shown). Therefore, the anomaly maps and MLS satellite track emphasize that the study region was subject to TCO and $H_2O$ anomalies over the latitudinal band from 30° S to 10° S, with a zonal westward transition of the ozone minimum. Similarly to Evan et al. (2023), they indicate the co-localization of the $H_2O$ and ozone anomalies as the Hunga plume passed over the Indian Ocean.

MLS profiles which met the criterion selection were studied further. For each of the 113 $H_2O$ and ozone profiles, we computed the difference from their corresponding background average profiles. Subsequently, these individual differences were averaged, and results are presented in panels (a) and (b) of Fig. 6, where the thick black line indicates the mean value and the blue shaded region the ± 1σ standard deviation with respect to the mean value. Panel (c) shows the vertical correlation between ozone loss and water vapor excess. These results show that the selection criterion holds profiles with a distinguishable ozone loss and water vapor excess at the 14.68 hPa pressure level (highlighted with a horizontal black line). The largest ozone anomaly reads -0.43 ± 0.66 ppmv and the largest water vapor anomaly is 125.74 ± 51.82 ppmv. The right panel indicates that the largest anti-correlation (r = -0.68) is obtained at the level with maximum ozone anomaly. As Fig. 5 revealed local TCO minima, Fig. 6 shows that these minima are due to reduction in stratospheric ozone (in the range 46–12 hPa, corresponding to 21–30 km) and that this is linked to the $H_2O$ excess. This observation confirms previous research (Evan et al., 2023; Zhu et al., 2023) and indicates that the ozone anomaly is linked to a reduction of the ozone layer.

### 3.5 Transport of air-masses in the stratosphere

The Lagrangian HYSPLIT model was used to investigate the origin of the air-masses responsible for the ozone anomaly over the Indian Ocean following the Hunga eruption. Back-trajectories were run from the location of Reunion on 21 January at 00:00, and for 9 distinct altitudes ranging between 22 and 26 km. Figure 7 shows the result of the HYSPLIT simulation, where the darkest trajectory represents air-masses at 26 km, and the lightest trajectory represents air-masses at 22 km. Figure 7 shows that all back-trajectories are zonal, moving westward and passing over the location of the Hunga eruption. The results of the



HYSPLIT back-trajectories simulation are consistent with the lidar measurements made in Reunion (see Fig. 2), as well as
with the ozone anomalies over the region of study as depicted in Fig. 5. Additionally, the latter shows a westward transition of
ozone anomalies in the stratosphere over the Indian Ocean.

To support these results, we used ERA5 EPV contours to highlight the dynamics of the stratosphere at the 600 K potential
temperature level ($\sim$24 km). Daily results from 15 to 23 January are shown in panels (a) to (i) of Fig. 8, where EPV is
expressed in potential vorticity units (PVU, with 1 PVU = $10^{-6}$ m$^2$ s$^{-1}$ K kg$^{-1}$). In this figure are also superimposed the
MLS satellite tracks with the profiles meeting the criterion selection for large water vapor level detection. The red dashed line
corresponds to the position of the subtropical barrier as detected by the DYBAL algorithm, and the star represents the location
of Reunion. When it enters the Indian Ocean on 18 January, the bulk of the stratospheric water vapor anomaly lies to the
north of the subtropical barrier, itself located at an average global latitude of 26.6° S. In the course of its westward transport
toward Madagascar, the anomaly stays north of the subtropical barrier. This region appears to show no marked discontinuity in
the EPV field during this period, allowing isentropic transport from east to west over the 600 K isentropic surface. When the
anomaly exited the region of study on 22 January, the subtropical barrier (red dashed line in panel (h)), located at an average
global latitude of 24.8° S, was still distinctly south of the anomaly.

The EPV contour map in panel (j) represents the average EPV from 15 to 23 January. The quasi-linear contours of mean
EPV further confirms that no sharp EPV discontinuity was present. Therefore, the westward transport was made possible by
the austral summer general stratospheric circulation, as well as the strong EPV gradient poleward.

## 4 Conclusions

The main eruption of the Hunga volcano released significant amounts of aerosols, water vapor, and a moderate quantity of
sulfur dioxide into the atmosphere (Sellitto et al., 2022; Zuo et al., 2022; Millán et al., 2022), resulting in substantial anomalies
within the stratosphere. OMPS aerosol extinction profiles revealed that the volcanic plume extended through the stratosphere,
from 5° S and 25° S, and reached altitudes greater than 35 km over the Indian Ocean. These results are supported by the Maïdo
aerosol lidar, which observed the plume during two consecutive nights a few days after the eruption, indicating that the core of
the plume was passing over Reunion at an altitude ranging from 26.8 to 29.7 km.

The ozone anomaly associated with the volcanic plume was investigated using MLS and IASI ozone data. Based on these
results, we state that the advection of the volcanic aerosol and water vapor plumes had an impact on ozone levels over the
Indian Ocean, as an ozone mini-hole structure was found to extend over large areas of the studied region, as emphasized by
IASI on 21 January when it showed a TCO anomaly of -38.97 $\pm$ 25.39 DU. MLS profiles impacted by the water vapor anomaly
showed that the ozone reduction occurred at the level of the ozone layer. Specifically, the average ozone anomaly reads -0.43
ppmv with a standard deviation of 0.66 ppmv at the 14.68 hPa pressure level.

The dynamics of the stratosphere responsible for the volcanic plume's advection over the Indian Ocean were studied using
ERA5 EPV maps. The HYSPLIT simulation highlighted the presence of the summer's westward stratospheric flow. ERA5 EPV
maps indicated that the region situated between Australia and Reunion showed no EPV discontinuity at the 600 K isentropic





level during the passage of the volcanic plume. Thus, due to isentropic transport and the summer's westward stratospheric flow, the volcanic plume traveled westward and reached the Indian Ocean and Reunion within days, confirming the results of previous studies.

425 This study showed the evolution of the localization of the early ozone and water vapor anomaly in the Hunga volcanic aerosol plume in the Indian Ocean. We examined the impact of this aerosol and water vapor plume on stratospheric ozone reduction, yet the daily measurement frequency of satellite observations hinders the acquisition of a detailed understanding of the short-term variations and nuances in ozone during the event. This would require the use of meso-scale models that incorporate a comprehensive representation of stratospheric chemistry and dynamics, as well as the perturbations caused by 430 the Hunga.

**Appendix A: Statistical parameters**

Statistical parameters were used for comparisons between satellite (MLS and IASI) and ground-based instruments (stratospheric lidar and SAOZ). Here we chose to use the correlation coefficient (r) and the Root-Mean-Square Dispersion (RMSD) to assess the differences and agreements between both datasets in different layers of the atmosphere. They are based on the 435 following equations:

$$r = \frac{\sum_{i=1}^{N}(O_{3\ GRD_i} - \overline{O_{3\ GRD}}) \times (O_{3\ SAT_i} - \overline{O_{3\ SAT}})}{\sqrt{\sum_{i=1}^{N}(O_{3\ GRD_i} - \overline{O_{3\ GRD}})^2} \times \sqrt{\sum_{i=1}^{N}(O_{3\ SAT_i} - \overline{O_{3\ SAT}})^2}}, \tag{A1}$$

$$RMSD = \sqrt{\frac{1}{N}\sum_{i=1}^{N}(O_{3\ GRD_i} - O_{3\ SAT_i})^2}, \tag{A2}$$

where $N$ is the number of available observations (ozone profiles or TCO), $O_{3\ GRD_i}$ represents the ground-based data compared to satellite ($O_{3\ SAT_i}$) observations, and the index $i$ iterates over the available observations at different time steps. The overline 440 indicates an average of $N$ observations. For each coincident profile, individual values of r and RMSD are obtained. The r and RMSD values reported in Sect. 3 are the averages derived from the individual r and RMSD values across all compared profiles.

*Data availability.* Reunion aerosol lidar used in this steady are accessible from https://doi.org/10.5281/zenodo.7790284 (last accessed on 23 January 2024). Reunion ozone lidar measurements are available through the NDACC page (https://ndacc.larc.nasa.gov/, last accessed on 23 January 2024). MLS data can be downloaded using NASA's data portal (https://disc.gsfc.nasa.gov/, last accessed on 23 January 2024). IASI 445 and data are accessible from https://iasi.aeris-data.fr (last accessed on 23 January 2024). SAOZ data can be downloaded from http://saoz.obs. uvsq.fr/ (last accessed on 23 January 2024). AERONET Version 3 Level 2 data are available through this link: https://aeronet.gsfc.nasa.gov/



(last accessed on 23 January 2024). OMPS data can be accessed from https://ozoneaq.gsfc.nasa.gov/ (last accessed on 05 March 2024). HYSPLIT back-trajectories can be obtained from https://www.ready.noaa.gov/HYSPLIT_traj.php (last accessed on 23 January 2024).

*Author contributions.* TM was the project leader; HB was the supervisor of the project; HB and NB participated in the methodology and
interpretation of the results; all co-authors participated in the review of the manuscript.

*Competing interests.* The authors declare that they have no conflict of interest.

*Acknowledgements.* The authors acknowledge the CNRS-NRF IRP ARSAIO (Atmospheric Research in Southern Africa and Indian Ocean) project for supporting research activities, as well as the Conseil Régional de la Réunion for the Ph.D. scholarship of Tristan Millet. The authors thank NASA for facilitating easy access and providing documentation for OMPS ans MLS data. The authors extend their thank to
ECMWF for providing access to ERA5 data, to the NOAA–ARL for supplying the HYSPLIT transport and dispersion model and to IASI for providing access and documentation related to their data. The authors are appreciative of the PIs for providing data and their respective teams for maintaining the lidars and AERONET stations used in the present article. The authors acknowledge the support of the European Commission through the REALISTIC project (GA 101086690). The projects OBS4CLIM (Equipex project funded by ANR: ANR-21-ESRE-0013), EECLAT and AOS (CNES) are acknowledged. The authors acknowledge the CNRS (INSU), Météo France, and the Université de
la Réunion for funding the infrastructure OPAR (Observatoire de Physique de l'Atmosphère à la Réunion) and OSU-R (Observatoires des Sciences de l'Univers à la Réunion, UAR 3365) for managing it. The federation Observatoire des Milieux Naturels et des Changements Globaux (OMNCG) of the OSU-R is also acknowledged. Lucien Froidevaux and Natalya Kramarova are warmly thanked for providing valuable insights into MLS and OMI data, respectively. Finally, the first author expresses heartfelt gratitude to Krzysztof Wargan for his significant contribution in providing valuable remarks that have improved the quality of the article.



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

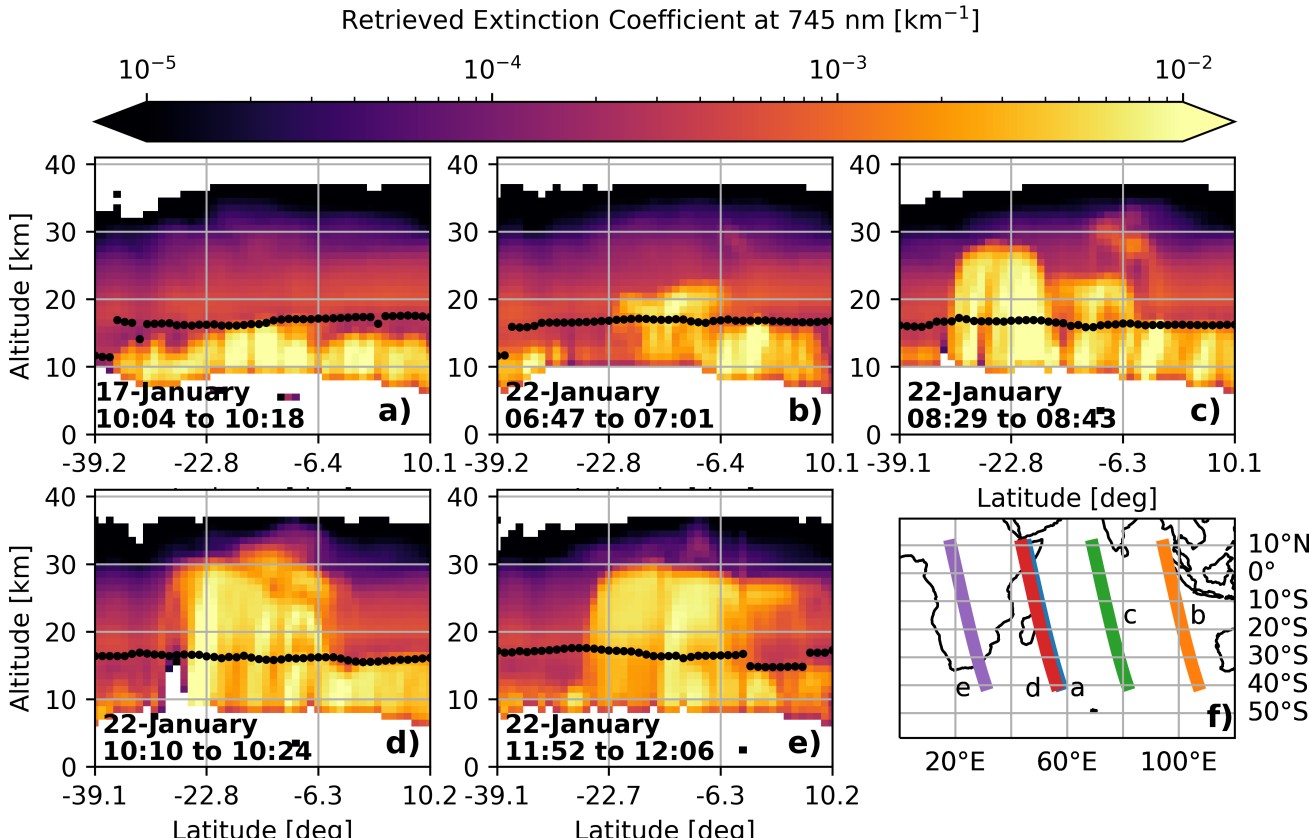

**Figure 1.** OMPS-LP aerosol extinction height-latitude cross-sections over the Indian Ocean at 745 nm for **a**) background conditions prior to the passage of the volcanic plume on 17 January and **b–e**) during the passage of the plume on 22 January. Panel **f**) shows the satellite track corresponding to each overpass. The superimposed black dots on panels **a–e**) indicate the instrument's estimation of the tropopause height.

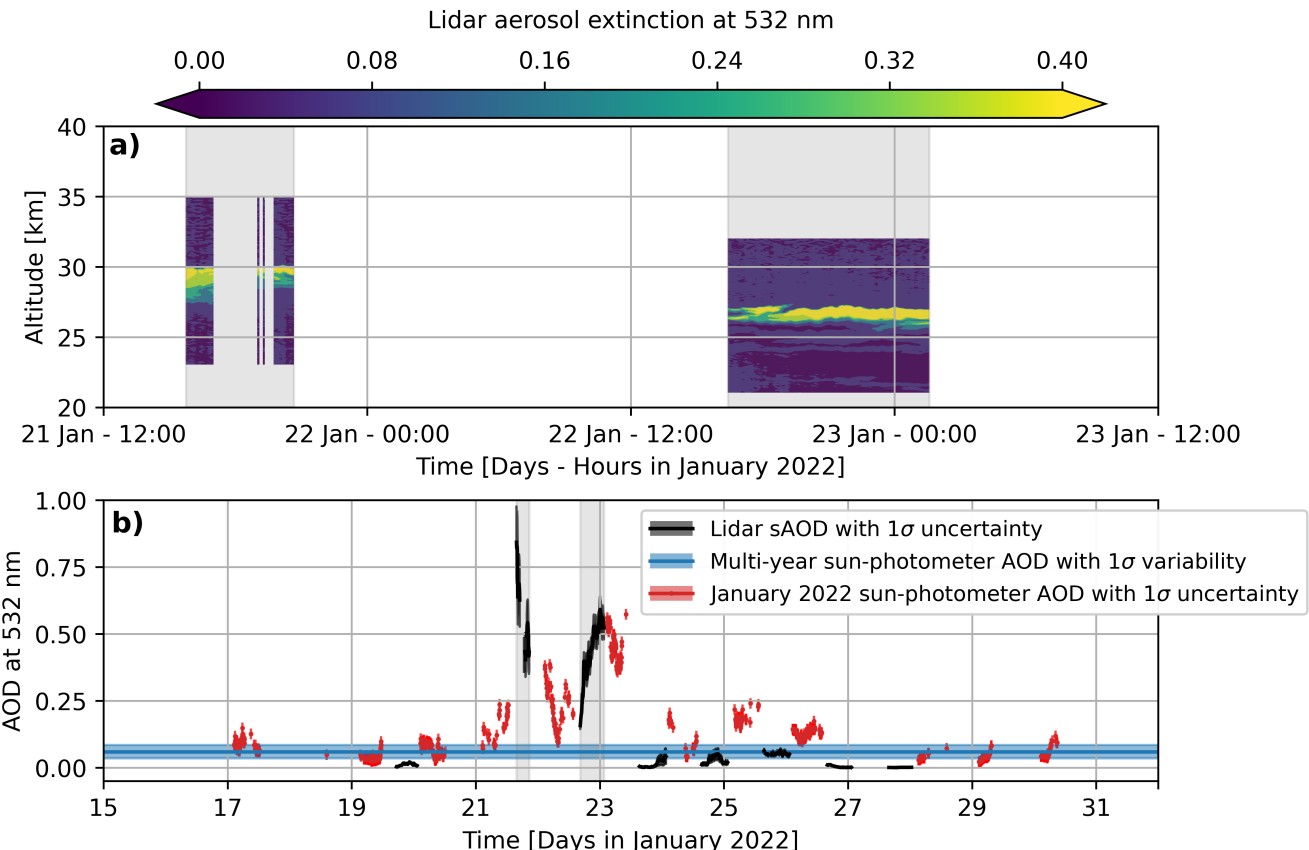

**Figure 2.** Aerosol lidar extinction profiles at 532 nm (**a**) and aerosol lidar sAOD in black with level 2.0 sun-photometer total AOD in red and their associated uncertainties (**b**). The blue line and shaded area represent average and standard deviation values given by level 2.0 sun-photometer data from 2003 to 2021. The common observation periods in both panels are visually represented with gray regions.





**Figure 3.** Multi-year stratospheric DIAL profiles obtained from observations between 2013 and 2021 at Reunion. The black dotted line shows the altitude of the ozone maximum.



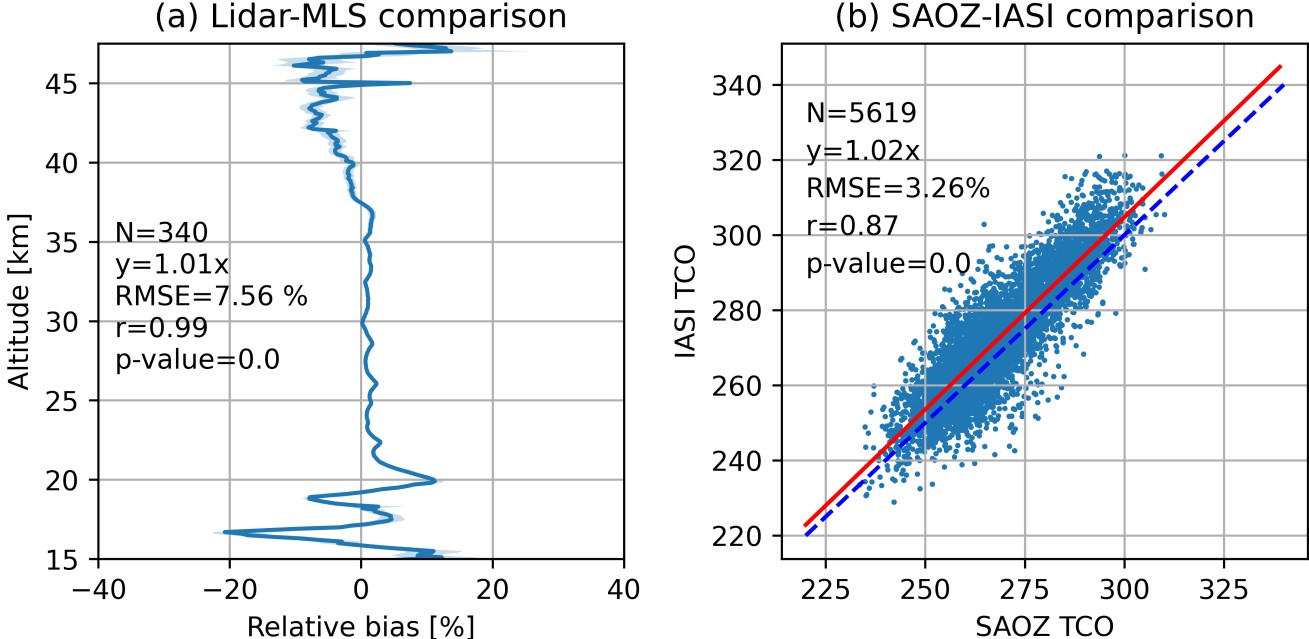

**Figure 4. a**) Mean relative bias (solid line) and $\pm\,1\sigma$ standard error (shaded area) comparing nocturnal DIAL ozone profiles to the corresponding MLS ozone profiles between January 2013 and December 2021. **b**) Direct comparison between SAOZ TCO and IASI TCO from data points obtained between March 2013 and December 2021. Statistical results presented in the left side each panel were obtained from the comparison of all data points, irrespective of the altitude level, date and time.



**Figure 5.** Daily evolution of TCO (**a1-a9**) and TCO anomaly (**b1-b9**) observed by IASI alongside the satellite track of MLS (light blue dots) and the total $SO_2$ column from IASI (**c1-c9**) between 15 and 23 January. Green dots on the MLS track represent the location of profiles with exceptionally high $H_2O$ meeting the criterion selection. The red contour indicates the regions where total $SO_2$ column is greater than 30 DU. The black and white star represents the location of Reunion. Each row corresponds to a distinct day, and the date of observation is indicated for each row in the right column.



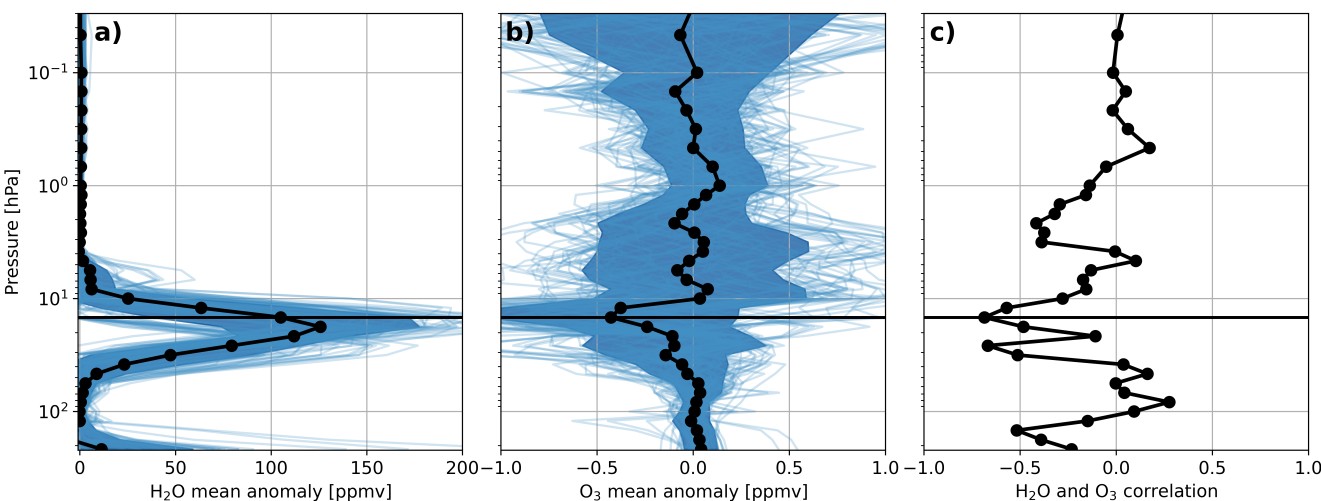

**Figure 6.** Average anomalies (thick black line) and $\pm 1\sigma$ standard deviation (shaded blue region) in **a**) water vapor and **b**) ozone profiles determined using MLS profiles which met the criterion selection. Panel **c**) shows the correlation between ozone reduction and water vapor excess. The horizontal black line in all three panels represents the location of the 14.68 hPa pressure level.



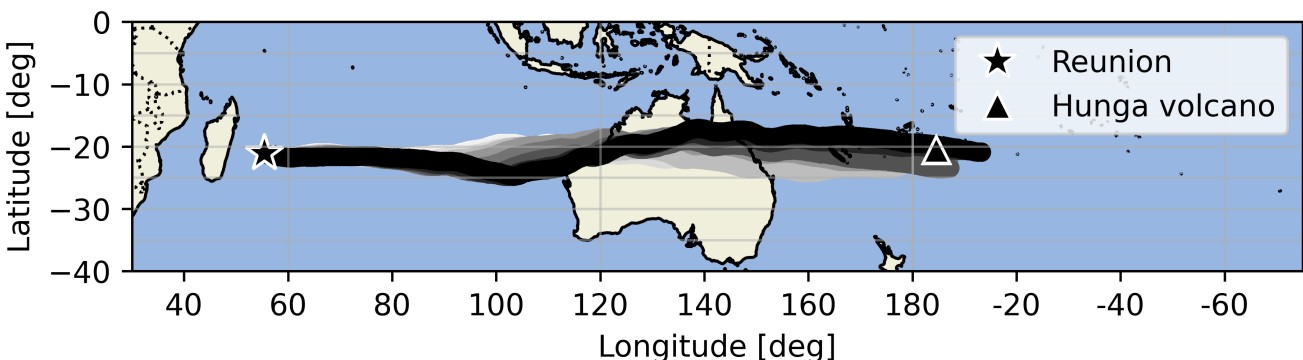

**Figure 7.** HYSPLIT back-trajectories of 240 hours ending on 21 January at 00:00 UTC at the location of Saint-Denis, Reunion, between 22 and 26 km height. The star and triangle symbols indicate the ending point and the Hunga volcano location, respectively. The back-trajectories are displayed with thick lines of different shades of grey, ranging from white for the 22 km height to black for 26 km.

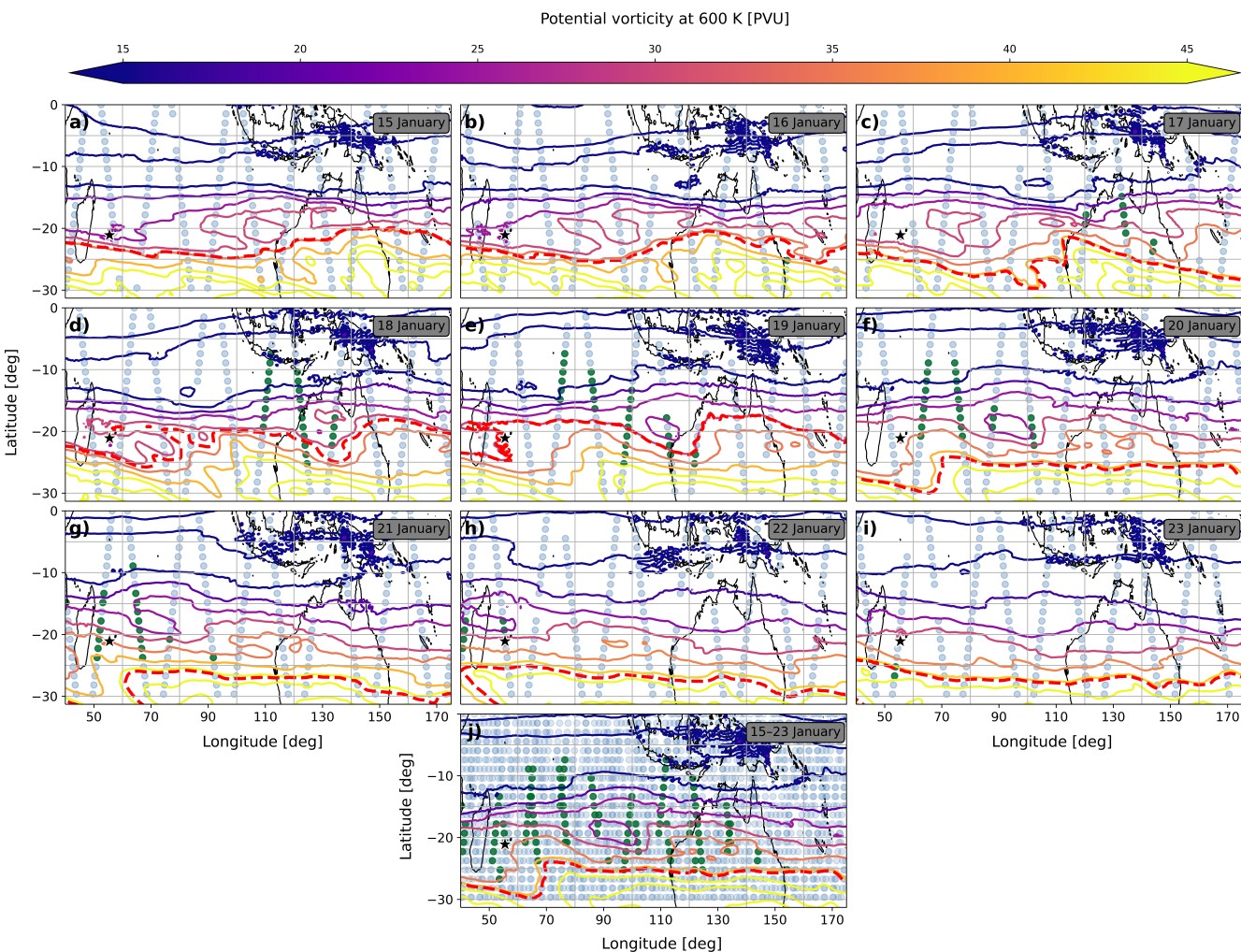

**Figure 8.** Daily averages of ERA5 hemispheric EPV from 15 January to 23 January and the corresponding MLS satellite track. Panel **j)** shows the average of daily maps and the superimposition of all MLS satellite tracks. The red thick line represents the DYBAL estimated position of the subtropical barrier and the black and white star represents the location of Reunion.