# Peer review of "Evidence of a Transient Ozone Depletion Event in the Early Hunga Plume Above the Indian Ocean"

_EGUsphere, 2024_

## Referee Comment (RC1)

**Review of "Evidence of an Ozone Mini-Hole Structure in the Early Hunga Plume Above the Indian Ocean" by Millet et al.**

This paper uses IASI total column ozone (TCO) measurements and MLS vertically resolved ozone profiles to investigate low ozone observed over the Indian Ocean in the week following the Hunga eruption. First, measurements of aerosol from OMPS-LP and two ground-based instruments are used to characterize the passage of the volcanic plume over Reunion. Then IASI and MLS data are compared to ground-based measurements (SAOZ TCO and DIAL profiles, respectively) obtained at Reunion under background conditions to confirm their suitability for the study. Negative anomalies in IASI TCO are linked to a negative anomaly in MLS profiles that peaked around 15 hPa, where the excess water vapor injected by Hunga was maximum. Transport of Hunga-influenced air masses was explored through HYSPLIT back trajectories and inspection of ERA5 PV maps.

In my opinion, both the analysis performed in this study and the presentation thereof are seriously flawed. While it is possible that some of these individual issues could be addressed with more work, others are fundamental in nature. Taken together, I believe that these deficiencies should preclude publication of the manuscript in anything resembling its present form. However, I realize that sometimes manuscripts are published even when a reviewer feels that rejection is warranted. Therefore, in addition to summarizing my major concerns, I have made the effort to describe in detail specific substantive issues that would need to be addressed before the manuscript could be re-considered. I have also listed a number of minor points of clarification as well as grammar/ typo corrections at the end.

Major comments:

- The term "ozone mini-hole" has a specific meaning – it refers to a transient natural synoptic-scale phenomenon that arises, mainly in midlatitudes, through dynamical and transport processes (a combination of uplift and horizontal advection of ozone-poor air). Total column ozone decreases rapidly during a mini-hole event but returns to its initial levels as the weather systems pass. Ozone mini-holes are unrelated to photochemical processes. Thus, the region of low ozone described and attributed to Hunga in this paper is NOT an "ozone mini-hole". This wording needs to be changed throughout the manuscript, including the title.

- The authors acknowledge that the Hunga plume adversely affected the DIAL and SAOZ ozone retrievals, and therefore the post-eruption data from those ground-based instruments are not used in their analyses. The impact of the extreme stratospheric hydration from Hunga on MLS retrievals is also discussed (although that description needs some clarification, as noted in the specific points below). In contrast, while cloud contamination is noted, the potential effects from Hunga on IASI data are not mentioned, and those measurements are presented with no Hunga-related caveats whatsoever. It is hard to believe that the IR measurements from IASI would be completely unaffected by either the enhanced gas-phase $SO_2$ following the eruption or the sulfate aerosol that formed from it within the first week. Indeed, as the

manuscript shows, the region of low ozone is highly aligned with the region of initially high SO$_2$. The conversion to sulfate is then inferred from the reduction of SO$_2$ in the region of low ozone. Whether this is a real atmospheric feature or a measurement artifact is not clear. It is essential that discussion of the IASI data quality in the wake of the eruption be added.

- Further to the preceding point, I find Figure 5 and the associated discussion unconvincing. I have several technical criticisms of the figure/text, detailed in the specific points below. But the big-picture issue is that the depiction of anomalies in the IASI maps is not compelling. Anomalies of apparently comparable magnitude can be seen in many parts of the displayed area, including in the vicinity of Reunion on 15 January before the arrival of the Hunga plume, so the anomalies being spotlighted by the authors hardly stand out. Most of the maximum anomalies quoted in the text are marginal, and some are not significant at even 1$\sigma$. Moreover, the focus on maximum anomalies is puzzling. Since in most cases the exact location of these points is not specified, it is not even certain that they occurred in the region near Reunion and not elsewhere in the study area. It is not clear why a regional average anomaly on each day was not computed and related to the passage of the plume.

- The maximum anomaly in January 2022 TCO from IASI (about −39 DU) was linked to the average ozone anomaly in Hunga-influenced profiles measured by MLS, which peaked at 15 hPa. There are several issues with this aspect of the study, starting with relating maximum anomalies in TCO to average anomalies in vertically resolved ozone. In addition, the 15-hPa average ozone anomaly from MLS is not significant (−0.4 ± 0.7 ppmv, 1$\sigma$), so these results are even less convincing than those based on column ozone. Most importantly, it is not possible to reconcile the magnitudes of the two sets of anomalies, as illustrated in the figure embedded below. The black line shows a climatological MLS ozone profile (for 2005, a representative year) calculated over the region 10°S–30°S; its associated stratospheric / mesospheric (100–0.001 hPa) burden is 233.9 DU. The red line shows the same climatological profile perturbed with an anomaly like the one indicated in this study (−0.4 ppmv at 15 hPa). This modified profile has an ozone burden of 228.7 DU, only about 5 DU less than the original profile. The purple line shows the climatological profile with 0.5 ppmv subtracted between 40 and 1 hPa, effectively perturbing the bulk of the stratospheric ozone layer. The associated burden for this profile is 217.5 DU, a reduction of about 16 DU. Finally, the green line shows the results for 1 ppmv subtracted from the climatological profile over 40–1 hPa. In this case, the reduction in the burden is 32.6 DU, still about 6 DU less than the maximum anomaly in TCO reported in this paper. The key point is that the entire stratospheric ozone layer would have to be substantially perturbed to achieve an anomaly in total ozone of the magnitude asserted here. If indeed TCO was truly reduced by as much as 39 DU, then the decrease must have occurred in the troposphere rather than the stratosphere, in which case it is very unlikely to have been related to the Hunga eruption.

[Figure]

- I fail to see the point of much of the discussion in Section 3.5 on the transport of Hunga-influenced air masses over the Indian Ocean. Several previous studies tracked the early dispersion of the plume, including Millán et al. (2022), Legras et al. (2022), and Khaykin et al. (2022); moreover, its presence over Reunion within a week has already been established by Baron et al. (2023) and Evan et al. (2023). Even if the authors felt that further confirmation was needed, the HYSPLIT trajectory calculations would have been sufficient. Instead, maps of ERA5 PV are shown and the fact that they reveal no "marked discontinuity" in the PV field during this period is argued to be evidence that east-to-west isentropic transport at 600 K was possible. It is not clear what kind of atmospheric feature it is thought may have impeded such transport. An issue that is overlooked in this discussion is that ERA5 does not assimilate water vapor measurements, and thus it did not accurately capture post-eruption perturbations in stratospheric circulation, as discussed for MERRA-2 by Coy et al. (2022).

- Fundamentally, the raison d'être for this manuscript is not clear. Much of the analysis centers on evaluation of MLS and IASI ozone data through comparisons with DIAL and SAOZ measurements made at Reunion under background conditions. The statement is made "Based on the excellent correlation and agreement between satellite (MLS and IASI) and ground-based instruments (stratospheric lidar and SAOZ) over Reunion, it appears relevant to use satellite ozone products to investigate the changes in the distribution of ozone over the study region." But this is hardly a surprising result – both MLS and IASI are very well-characterized data sets that have already been employed extensively in similar kinds of studies, including over the region in question. In fact, arguably the entire intercomparison portion of this study was unnecessary. On the other hand, validation of the satellite measurements – in particular those from IASI – under perturbed post-eruption conditions would have been valuable, but that was not possible using the Reunion data as noted above. Furthermore, this work seems to have provided no additional scientific insights beyond those already presented in the papers by Baron et al. (2023), Evan et al. (2023), and Zhu et al.

(2023). Indeed, as the authors note, Baron et al. (2023) presented the lidar data and talked about the passage of the Hunga plume over Reunion. Evan et al. (2023) presented MLS (and other) data, including ozone, over Reunion during the same timeframe. Evan et al. (2023) and Zhu et al. (2023) elucidated the mechanisms giving rise to the observed low ozone (conclusions that this paper makes no attempt to add to). Nothing in this current study is new, other than the addition of total column ozone measurements, whose reliability in this particular region at this particular time has not been adequately addressed, as noted above.

- Throughout the manuscript, numerical results are reported with what seems to me to be an unjustifiably high degree of precision. As just one example, the maximum anomaly in IASI TCO is stated to be −38.97 ± 25.39 DU. This "false precision" needs to be removed.

Specific substantive issues:

- L1: Most of the aerosols of stratospheric significance were not emitted directly by the volcano, but rather arose through subsequent $SO_2$ conversion to sulfate.
- L16: It is not clear why the 2018 WMO Ozone Assessment is referenced for this general statement, rather than the most recent Report from 2022, which is cited elsewhere in this manuscript.
- L30-42: I have several comments on this paragraph:
  - It is stated in the first sentence that eruptions can influence tropospheric ozone, but the rest of the paragraph does not elaborate on this point at all, and it is not clear why it is relevant to this paper (unless the observed reduction in ozone is in fact occurring in the troposphere). The connection to tropospheric ozone needs to either be explained better or omitted altogether. Moreover, for clarity, in L36 "contribute to ozone depletion" should be "contribute to stratospheric ozone depletion".
  - It is stated that eruptions release substantial amounts of aerosols, but the volcanic aerosols of most consequence for the stratosphere are those formed subsequently by the conversion of $SO_2$ to sulfate, not those (e.g., ash) emitted directly by the volcanoes.
  - Literature citations are inadequate. It is not sufficient to cite only Tie and Brasseur (1995), Hofmann and Solomon (1989), and McCormick et al. (1995) for these points – many more references than these would be relevant in each case. At the very least, an "e.g.," needs to be added in front of all of these references.
  - It is not clear what is meant by the sentence "*Additionally*, reactive anthropogenic chlorine compounds may be enhanced in volcanically perturbed regions, leading to *further* ozone depletion" [emphasis added]– how is this different from "the activation of chlorine compounds on volcanic particles", "ozone depletion through heterogeneous chemistry", and "relationship between $SO_2$ and chlorine in causing ozone decline post-eruption" that have already been mentioned in the preceding three sentences?
- L59: Wright et al. (2022) is not the most suitable reference for the Hunga aerosol perturbation; in addition to Sellitto et al. (2022), other appropriate work to cite for this point include Khaykin et al. (2022) and Taha et al. (2022) – both already cited elsewhere. Wright et

al. (2022) is pertinent to the statement about the comparative energy release by Hunga, so it should be moved to that part of the sentence.

- L61: Sellitto et al. (2022) is not really the best reference for the magnitude of the Hunga water vapor injection; it should be replaced here by Khaykin et al. (2022) and Vömel et al. (2022, https://doi.org/10.1126/science.abq2299).

- L63-65: The sentence "As a result of the main austral summer stratospheric circulation and the prevalent phase of the QBO, the first signs of the Hunga aerosol plume's passage over Reunion were noticed only 4 days after the main eruption" is problematic for several reasons. First, it's not clear what "main" means in this context (and the word "main" is used in three other places in the paragraph in reference to the eruption, so it is confusing). Second, the QBO is mentioned, but its influence is not made clear – was the QBO in an easterly or westerly phase at the time of the eruption, thus did it delay or accelerate the plume's arrival over Reunion? I believe that the authors mean that the prevailing westward flow brought the plume to the region of Reunion very quickly, such that it could be observed by instruments there within a short period of 4 days, but the wording is ambiguous and could be misinterpreted. Third, this is the first mention of Reunion in the main text. Since its importance to this work has not yet been established, it comes out of the blue and is a bit jarring. A lead-in sentence introducing Reunion and giving the reader a hint about its role in this work would be good. Otherwise, the relevance of the following information is unclear.

- L67-79: This discussion of the results of Evan et al. (2023) and Zhu et al. (2023) could be better organized – it jumps back and forth between heterogeneous and gas-phase reactions, making it difficult to follow. More importantly, some of the results of those studies are misstated. First, the Hunga-induced stratospheric cooling enhanced heterogeneous reaction rates but was not a factor in the rapid conversion of $SO_2$ to sulfate aerosols, as is implied by the current wording (L69-70). (Also, a reference to the earlier paper by Zhu et al. (2022, https://doi.org/10.1038/s43247-022-00580-w) should be added for the impact of abundant OH from the Hunga hydration on the rapid sulfate formation.) Second, I was puzzled by the emphasis on photolysis of $Cl_2$ (L76), as this is not part of the conclusions about gas-phase chemistry reported by Zhu et al. (2023) as is suggested, but then I found a similar sentence in the paper by Evan et al. (2023). However, Evan et al. are talking about the negative HCl anomaly arising from *heterogeneous* chlorine activation on sulfate. Their statement about $Cl_2$ photolysis is made in connection with the colocated positive anomaly seen in daytime ClO. It is not correct that this is a "key gas-phase mechanism contributing to ozone loss".

- L80-93: Although the longer-term evolution of the Hunga water vapor and aerosol plumes is certainly interesting, it is not clear what relevance any of this has to the ozone distribution in the first week following the eruption, which is the focus of this study. If such discussion is retained, then it needs to be much more comprehensive in its summation of the existing literature on Hunga's radiative impact. Moreover, if the radiative effects from Hunga in subsequent months are covered here, then why are its chemical effects ignored?

- L95 (and also L136): Livesey et al. (2008), which is a conference proceeding, is not a suitable reference for Aura MLS. The paper by Waters et al. (2006) is sufficient.

- L98-99 and L102: The phrase "dynamics of its advection" seems strange to me, since "dynamics" and "advection" are essentially synonyms. I suppose that the authors mean that

they will show details of the plume's transport, but this should be clarified. Moreover, it is not clear what "its" in this sentence is referring to – grammatically it does not make sense.

- L111-116: It is not sufficient to simply state that the temporal and vertical resolution of the lidar data is "high". This information should be specified, especially the vertical resolution. Moreover, it is not appropriate to characterize a data set consisting of a total of 470 profiles obtained over a 9-year period as having "high" temporal resolution.

- L132-135: Aspects of the MLS description need to be improved. The term "consistent measurement frequency" is ambiguous – initially I thought it was referring to spectral frequency. Thus, "spatial sampling" would be better. Also, it is not clear what "consistent" means in this context (and the MLS orbit ground tracks do differ slightly from one day to the next) – I would delete this word. It is not quite correct to refer to MLS as "a radiometer" (the instrument actually consists of seven radiometers); this is an unnecessary detail that it would be better to omit.

- L136-140: The recommendations of Millán et al. (2022) are slightly mischaracterized. That paper stated that the reliability of MLS measurements *inside* the Hunga plume (not "close to" it) was degraded in the first few weeks immediately following the eruption, because of the enormous enhancement in $H_2O$ concentrations. The statement that "MLS v4 relies only on profile retrievals from $O_2$ signals whereas v5 also uses the $H_2O$ line" is unclear – this statement refers specifically to how information about *instrument pointing* (required for the retrieval of atmospheric composition profiles) is obtained in the two versions. This should be clarified. In addition, Millán et al. (2022) indicated that the standard MLS data quality screening protocols should NOT be implemented for the v4 $H_2O$ data during that initial post-eruption period. On the other hand, such filtering should still be performed for the $O_3$ measurements, whose quality, as noted in L139, was unaffected by Hunga. The description of the MLS v4 data handling is unclear on this issue – since both the v4 and the v5 MLS Data Quality Documents are referenced in L157, the implication is that the v4 data (both $H_2O$ and $O_3$) were screened, but the data filtering recommendations should be followed and the approach taken should be stated explicitly.

- L144-146: Some rearrangement of this discussion is needed. The numbers of MLS profiles being examined here – 113 influenced by Hunga and 2190 in unperturbed conditions between 15 and 23 January – only make sense in the context of a limited region (since MLS measures ~3500 profiles per day). However, the information that the comparison is restricted to a 5-degree radius around Reunion is not provided until much later in the paragraph.

- L147: Is the standard deviation calculated separately at each pressure level or over the whole 10–100 hPa range? In other words, is the maximum value of 0.05 ppmv quoted here never exceeded at any single level? If the standard deviation is being calculated over the entire profile, then a larger value (at, say, the level of peak ozone anomaly) could be getting "diluted" in the overall profile standard deviation.

- L152: I am interpreting the statement "repeated for both ascending and descending nodes" to mean that for the comparison between perturbed and unperturbed conditions, background values were calculated separately for the measurements obtained on the two sides of the orbit. Since ozone at these altitudes does not display large diurnal variations, I am wondering why it was considered necessary to derive both daytime and nighttime background profiles.

- L206-207: While the three references cited in this sentence are pertinent to the statement that persistent Hunga effects were confined to the stratosphere, they are completely unsuitable for the point that they immediately follow, which is that the stratospheric circulation is stable and stratified. In fact, no such statement is needed to justify the use of trajectory calculations (a very common technique). I recommend deleting everything in this sentence up to "we used HYSPLIT".
- L218-219: Similarly, PV is so widely used now for characterizing isentropic transport in the stratosphere that not only is such a list of citations unnecessary, but also the one provided is so seemingly arbitrary and self-referential that it does more harm than good. This sentence should be deleted.
- Section 2.4: I have several concerns about the DIAL/MLS intercomparison and its description:
  - MLS retrievals are output on a pressure grid, whereas the comparison with the lidar measurements uses altitude as a vertical coordinate. How the MLS measurements are placed onto an altitude grid needs to be explained.
  - As noted above, the vertical resolution of the lidar measurements is not given, but I presume that it is much higher than that of the MLS ozone profiles (which is ~2.5–3 km in the lower stratosphere). Simply sampling the finer profile at the MLS retrieval surfaces is not the best approach. To make a truly fair comparison between high-vertical-resolution profiles and coarser-resolution MLS data, it is necessary to follow the guidance in the MLS Data Quality Document to apply the MLS averaging kernels to and perform a least-squares "smoothing" of the high-resolution data set (Sections 1.8 and 1.9 of the MLS Quality Document, respectively). Although performing such a procedure may not make a substantial difference to the bottom-line results, this issue is nevertheless worth some investigation. At the very least, an experiment in which the lidar profiles are smoothed over ±1.5 km (i.e., boxcar smoothing) should be conducted to explore the impact on the comparisons with MLS.
  - The notation in the numerator of Equation (1) is confusing: $O_{3\,MLS} - O_{3\,DIAL}(z)(z)$. Why is the first term not written $O_{3\,MLS}(z)$ (as in L240), rather than putting both "(z)"s at the end?
- L267 and Figure 1: It is difficult to judge where 5°S and 25°S are located on these panels, as the x-axis latitude grid is odd. Instead of placing the vertical lines at the x-axis major tick marks, it would be more helpful to draw vertical lines at 5°S and 25°S. I also request that minor tick marks be added to both x- and y-axes.
- L281-282 and Figure 2: I do not understand what "This multi-year average represents an average of AOD data which is grouped into months, irrespective of the years" means – does the blue line in Figure 2 show the overall average over the 2003–2021 period, or is it just the January mean over all the years in that period? The uncertainty error bars on the red and black lines in Figure 2b are nearly impossible to see, even when zooming in on the panel. Greater contrast between the lighter and darker shades is needed. I also request that minor tick marks be added to both x- and y-axes.
- L320-332 and Figure 4: This discussion requires clarification in several respects:
  - Saying "the bias decreases to −3.73" makes it sound as though the bias has gotten smaller, whereas it has changed sign but actually is larger in magnitude.
  - Given the large oscillations in the differences below 20 km, the average bias carries little meaning, so there is no real benefit in stating it.

- o Livesey et al. (2022) discuss the presence of known systematic vertical oscillations in the MLS ozone retrievals in the UTLS; these likely play a role in the differences below 20 km seen here. However, they do not explain the rather large relative bias at 20 km.
  - o Given the stated caveats on the lidar data, it is not justifiable to say "the MLS mean bias profile seems to under-estimate ozone concentrations by 20.73±1.89%". All that can be said with confidence is that there is a relative bias of ~21% between the two data sets.
  - o Similarly, the linear regression shows that "MLS profiles tend to slightly over-estimate ozone concentrations" *relative to DIAL*.
  - o I request that minor tick marks be added to both x- and y-axes.
- L345-371 and Figure 5: I have several issues with Figure 5 and the accompanying discussion:
  - o The east-to-west movement of the plume during this time period was shown also by Millán et al. (2022).
  - o Schoeberl et al. (2022) is not an appropriate reference for the rapid conversion of $SO_2$ to sulfate aerosol – Zhu et al. (2022) and Asher et al. (2023) are better citations for that.
  - o It is not clear what "important" means in the context of the negative ozone anomaly.
  - o I am confused about what the error bars on the daily minimum TCO and maximum TCO anomaly values represent – please clarify.
  - o Why is the IASI TCO anomaly in January 2022 (which is derived relative to the 2014–2021 IASI climatology) being compared to the SAOZ climatological January TCO rather than to the IASI January climatology?
  - o The color palettes used for the TCO maps need improvement, especially the anomaly one. For one thing, the color bar saturates below –10 DU, making it impossible for the reader to judge the ~20–40 DU maximum negative anomalies noted in the text. Although a bright blue color is used for those largest negative anomalies, the contrast between it and the color used for the negative anomalies with magnitude smaller than 10 DU is too weak to be readily discernible without extreme magnification. In addition, positive anomalies should be easily distinguishable from negative ones – as it is now, the zero line falls in the middle of a continuum of bluish-greenish colors.
  - o The overlaid contours depicting $SO_2$ are red, not blue.
  - o It makes little sense to highlight the MLS profiles with high $H_2O$ values in green, when the orbit tracks are overlaid on $SO_2$ maps plotted using a yellow-green color palette.
  - o The black and white star denoting the location of Reunion is very difficult to spot.
- L379: While the largest anti-correlation may coincide with the maximum ozone anomaly, the r value a couple of levels below is nearly as large.
- Figure 7: Why are the trajectories plotted using such thick lines? The individual trajectories are completely indistinguishable. Perhaps that is the point, but it could still be easily made using differently colored lines of more moderate thickness.
- L392-402 and Figure 8: As stated in the general comments, I do not see the added value of this discussion. In addition, I have some specific comments:
  - o Of what possible relevance is the location of the *global* average latitude of the subtropical barrier on these dates? In the context of this study, only its location in the region of the Indian Ocean is important.
  - o The anomaly does not stay completely north of the subtropical barrier – some solid green dots are clearly present poleward of the barrier on 18 and 19 January (panels (d) and (e)).

- It is stated that the anomaly exits the region on 22 January, but a couple of green dots still appear on the map on both 22 and 23 January, and on 23 January (panel (i)) they fall south of the subtropical barrier.
- The red line marking the subtropical barrier is dashed, not solid as implied in the caption.
- Using red for the subtropical barrier is a poor choice since it is overlaid on contours of similar color (ranging from purple to orange).
- As in other plots, it is very difficult to spot the black and white star indicating Reunion.

- L407: As noted earlier, most of the aerosols of stratospheric significance were not emitted directly by the volcano, but rather arose through $SO_2$ conversion to sulfate.
- L417: It is not appropriate to say "the ozone reduction occurred at the level of the ozone layer". The ozone layer is a broad feature, extending over roughly 15–35 km altitude. MLS showed substantial (but not significant) anomalies only in a narrow layer around 15 hPa.
- L431-441: The statistical quantities described here are common and widely used, so I am not convinced that this Appendix is really needed.

Minor points of clarification, wording suggestions, and grammar / typo corrections:

- The word "highlight" is used about a dozen times throughout the manuscript, but in many cases its usage is inappropriate. I suggest alternatives in the specific comments below.
- L4: Why are the observations used here (obtained in January 2022) referred to as "current"?
- L8: Spectrometer --> Sounder
- L9: add ", respectively" after "maps", to avoid giving the impression that vertically resolved profiles and total column ozone are provided by both instruments
- L24: "end of the century" – clarify by changing "the" to "this" or adding "21$^{st}$" in front of "century"
- L25: On the contrary --> In contrast
- L29: enhanced by anthropogenic activities such as agriculture, industry and transport, that release $NO_x$ and aerosols --> enhanced by anthropogenic activities that release $NO_x$ and aerosols, such as agriculture, industry and transport
- L32: add a comma after "(1982)"
- L34: add a comma after "($SO_2$)"
- L36: add a comma after "($H_2SO_4$)"
- L39: delete "Justifiably" (it is not clear what this word is intended to convey here, but it is not appropriate)
- L50: "PSCs" – acronym not defined
- L53: volcanic sulfate aerosols penetration --> penetration of volcanic sulfate aerosols
- L60: add a comma after "scrutiny"
- L64: QBO" – acronym not defined
- L74: again, "justifiably" is inappropriate and should be deleted
- L94, 97: Why are the observations used here (obtained in January 2022) referred to as "currently available" and "current"?
- L95: add a comma after "2004)"

- L99: analyses --> reanalyses
- L102: add a comma after "Ocean"
- L105: how could Hunga have had any impacts on ozone before the eruption?
- L111: resolutions --> resolution
- L114: add a comma after "low"
- L131: satellite observations of ozone profiles and TCO were used in complement to ground-based data --> satellite observations of ozone profiles and TCO complement the ground-based data; a global --> global
- L133: solar time --> local solar time
- L134: calculate --> observe
- L144: 113 ozone and water vapor profiles --> 113 Hunga-influenced ozone and water vapor profiles
- L150: each January 2022 impacted profiles --> each of the January 2022 impacted profiles
- L154: accuracy and precision that are both lower than 10 % --> accuracy and precision that are both better than 10 % ("lower" can be interpreted to mean "worse")
- L167: as a representative --> as being representative
- L172: correlation between … anomaly with --> correlation between … anomaly and
- L188: located in --> located at the
- L205: to compute and simulate trajectories --> to compute trajectories
- L206 and L208: "air masses" should not be hyphenated
- L208: simulation to highlight the trajectories --> simulation of the trajectories
- L210-211: the NOAA citation is out of place – it should immediately follow "GDAS" in L209
- L209: 240 hours --> 240-hour
- L210: equitably --> equally
- L214: a citation is needed for ERA5 (e.g., Hersbach et al., 2020)
- L222: localization --> identification
- L226: of the dynamical barriers --> of dynamical barriers
- L233: time --> times
- L260: ozone impacts --> impacts on ozone (as written, it sounds like the effects are from ozone rather than on it)
- L263: compared to the troposphere --> compared to that in the troposphere
- Figure 1 caption: background conditions prior to the passage of the volcanic plume on 17 January --> background conditions on 17 January prior to the passage of the volcanic plume (as written, it sounds like the plume passed over the area on 17 January)
- L270: enabled to monitor the --> enabled monitoring of the
- L272: colocalized --> colocated
- L273: highlight --> emphasize
- L274: AOD --> total AOD
- L278: AOD (in red) at 532 nm --> AOD at 532 nm (in red); also, add a comma after "2022"
- Figure 2 caption: both --> the two
- L284: both --> the two
- L297: total … were --> total … was
- L298: delete the comma after "km"; highlights --> illustrates

- L300: add a comma after "Reunion"
- L305: add "at that time" at the end of the sentence
- L307: relative to --> for
- L312: standard deviation --> the standard deviation
- L317: agreements are --> agreement is
- Figure 4 caption: MLS ozone --> daily MLS v5 ozone; side each --> side of each
- L319: high number --> large number
- L321: 20 km of altitude, with --> 20 km altitude of
- L327: lower number --> smaller number
- L328: 453 start below 20 km, 409 before 17.5 km and 131 before 15 km --> only 453 extend below 20 km, 409 below 17.5 km, and 131 below 15 km
- L333: high number --> large number; indicates --> leads to
- L334: "elevated correlation (r=0.87)" – elevated over what? The comparison here is unclear, especially since r=0.99 for the DIAL/MLS comparison
- L336: seem to be --> are
- L337: being --> that are
- L347: circles) comprising the MLS profiles with high $H_2O$ values which met the criterion selection (green circles) --> circles), with the MLS profiles with high $H_2O$ values that met the criterion selection marked by green circles
- L348: highlight --> indicate
- L352: visible on --> visible in
- L353: The first appearing important negative ozone anomaly linked to the Hunga appears --> the first important negative ozone anomaly linked to Hunga appears (although see above for a comment on this sentence, in particular the use of the word "important")
- L354: with --> with a value of
- L370: transition --> shift; Similarly to Evan et al. (2023), they indicate the co-localization of --> Similar to the findings of Evan et al. (2023), our results indicate the colocation of
- L372 and L395: criterion selection --> selection criterion
- Figure 6 caption: ozone profiles determined using MLS profiles which met the criterion selection --> ozone from v4 MLS profiles that met the selection criterion; also, the thin blue lines presumably represent the 113 individual profiles going into the average, but that should be explicitly stated in the caption
- L376: the correlation is between $H_2O$ and ozone itself, not ozone loss (some of the anomalies are positive); also, "holds" is not quite the right word here – "leads to" would be better
- L377: highlighted with --> marked by
- L377-378: largest ozone anomaly reads --> largest average ozone anomaly is; largest water vapor --> largest average water vapor
- L379: As Fig. 5 --> While Fig. 5
- L382: this sentence basically says nothing – by definition, a negative anomaly in stratospheric ozone is linked to a reduction of the ozone layer
- L383, L384, and L386 (twice): "air masses" should not be hyphenated
- L385, L388, L398, and Figure 7 caption (twice): "back trajectories" should not be hyphenated
- L386: delete ", and"

- L392: highlight --> investigate
- L395: large water vapor level detection --> detection of enhanced water vapor
- L397: enters the Indian Ocean --> enters the region of the Indian Ocean
- L410: from 5°S and 25°S --> from 5°S to 25°S
- L415: emphasized --> observed
- L416: add a comma after "January"
- L417: reads --> is
- L420: highlighted --> reflected; summer --> austral summer
- L425-426: evolution of the localization of the early ozone and water vapor anomaly in the Hunga volcanic aerosol plume in the Indian Ocean --> evolution of the colocated ozone and water vapor anomalies in the early Hunga volcanic plume over the Indian Ocean
- L430: the Hunga --> Hunga
- L434: both --> the two

---

## Author Comment (AC1)

**Response to Referee 1 Comments**

We would first like to express our sincere thanks and appreciation to Referee 1 for their thorough and detailed review. The comments identified flaws and unclear points in the article, providing an excellent opportunity to improve its overall quality.

Our responses follow the structure of the review document and are divided into three sections: 1) responses to major comments, 2) responses to specific substantive issues, and 3) responses to minor issues. Referee comments are written in black and authors answers are in blue.

**Major comments:**

**Point 1**: The term "ozone mini-hole" has a specific meaning – it refers to a transient natural synoptic-scale phenomenon that arises, mainly in midlatitudes, through dynamical and transport processes (a combination of uplift and horizontal advection of ozone-poor air). Total column ozone decreases rapidly during a mini-hole event but returns to its initial levels as the weather systems pass. Ozone mini-holes are unrelated to photochemical processes. Thus, the region of low ozone described and attributed to Hunga in this paper is NOT an "ozone mini-hole". This wording needs to be changed throughout the manuscript, including the title.

**Response 1**: Indeed, the ozone anomalies discussed in the manuscript are primarily chemical, not dynamical. To clarify this, we now refer to them as "Transient Ozone Depletion Event" and have updated the title and manuscript accordingly.

**Point 2**: The authors acknowledge that the Hunga plume adversely affected the DIAL and SAOZ ozone retrievals, and therefore the post-eruption data from those ground-based instruments are not used in their analyses. The impact of the extreme stratospheric hydration from Hunga on MLS retrievals is also discussed (although that description needs some clarification, as noted in the specific points below). In contrast, while cloud contamination is noted, the potential effects from Hunga on IASI data are not mentioned, and those measurements are presented with no Hunga-related caveats whatsoever. It is hard to believe that the IR measurements from IASI would be completely unaffected by either the enhanced gas-phase SO2 following the eruption or the sulfate aerosol that formed from it within the first week. Indeed, as the manuscript shows, the region of low ozone is highly aligned with the region of initially high SO2. The conversion to sulfate is then inferred from the reduction of SO2 in the region of low ozone. Whether this is a real atmospheric feature or a measurement artifact is not clear. It is essential that discussion of the IASI data quality in the wake of the eruption be added.

**Response 2**: The Hunga eruption caused significant disturbances in the IASI spectra, particularly within the first two days after the eruption, when pronounced temperature and water vapor anomalies were directly observed in the radiance spectra (see Wright et al., 2022). Increases in SO2 were also detected by IASI, with conversion to sulfates occurring more rapidly and efficiently compared to other volcanic eruptions. In contrast to UV-visible instruments like OMI, which reported significant ozone perturbations following the eruption (later attributed to interference from SO2/H2SO4), no similar disturbances were observed in the IASI O3 retrievals immediately after the event. Since the spectral ranges of ozone and SO2 do not overlap in the IASI ozone retrieval, there is no reason to expect any bias. While sulfate aerosols may share some spectral range with ozone, the retrieval algorithm can distinguish between the two, as sulfate aerosols exhibit strong absorption features, and ozone

variations are directly measured through its absorption lines.

It is important to note that the retrieval algorithms for H2O, SO2, O3, and other gases rely on a priori profiles and error covariance matrices to constrain the possible concentration values. For unusual events like the Hunga eruption, some covariance matrices may be inadequate, particularly for H2O, since water vapor is not typically found at such high altitudes. However, for ozone, the algorithm was specifically designed to monitor the ozone hole, so it should account for the vertical variability between 15 and 40 km effectively.

Elements of this discussion were added into the manuscript following the IASI description.

**Point 3**: Further to the preceding point, I find Figure 5 and the associated discussion unconvincing. I have several technical criticisms of the figure/text, detailed in the specific points below. But the big-picture issue is that the depiction of anomalies in the IASI maps is not compelling. Anomalies of apparently comparable magnitude can be seen in many parts of the displayed area, including in the vicinity of Reunion on 15 January before the arrival of the Hunga plume, so the anomalies being spotlighted by the authors hardly stand out. Most of the maximum anomalies quoted in the text are marginal, and some are not significant at even $1\sigma$. Moreover, the focus on maximum anomalies is puzzling. Since in most cases the exact location of these points is not specified, it is not even certain that they occurred in the region near Reunion and not elsewhere in the study area. It is not clear why a regional average anomaly on each day was not computed and related to the passage of the plume.

**Response 3**: In the original version of the manuscript, all negative anomalies below -10 DU were shown, regardless of their associated uncertainties. Many of these anomalies were not statistically significant at the $1\sigma$ level, which compromised the interpretation of the results and led to confusion. Regarding the update of Figure 5, we must highlight an important point. During the review process, we discovered that the IASI anomalies had been incorrectly calculated, resulting in significant over-estimations. Specifically, the IASI data files analyzed during the review included ozone anomalies expressed in two distinct units: mol./m$^2$ (consistent with the column ozone unit) or a relative unit, representing a percentage of the column ozone amount. The anomaly unit was not uniform across all data files. As the first author, I mistakenly assumed that all anomalies were expressed in mol./m$^2$. Consequently, the anomalies were overestimated by a factor of 10. After correcting this issue, the column ozone anomalies are now accurately calculated and fall within the 1–1.5 % range. For the Hunga-related ozone anomalies, we now obtain values that are significant at the $2\sigma$ level, greatly improving both the quality and relevance of Figure 5. We have revised the text associated with Figure 5 to enhance clarity and ensure that the location of the anomaly is explicitly specified whenever it is mentioned. However, in response to the referee's final comment: given that the transient depletion event changes both position and shape each day, and that the evolving atmospheric conditions result in varying observations and a different number of data points, it is not feasible to compute a regional average for this anomaly. At best, we could indeed derive an average total ozone content for the entire study region or centered around specific latitudes. However, since the number of available IASI data points varies each day, such an average would not accurately represent ozone variability related to the passage of the plume.

**Point 4**: The maximum anomaly in January 2022 TCO from IASI (about −39 DU) was linked to the average ozone anomaly in Hunga-influenced profiles measured by MLS, which peaked at 15 hPa. There are several issues with this aspect of the study, starting with relating maximum anomalies in TCO to average anomalies in vertically resolved ozone. In addition, the 15-hPa average ozone anomaly from MLS is not significant (-0.4 $\pm$ 0.7 ppmv, $1\sigma$), so these results are even less convincing than those based on column ozone. Most importantly, it is not possible to reconcile the magnitudes of the two

sets of anomalies, as illustrated in the figure embedded below. The black line shows a climatological MLS ozone profile (for 2005, a representative year) calculated over the region 10 °S–30 °S; its associated stratospheric / mesospheric (100–0.001 hPa) burden is 233.9 DU. The red line shows the same climatological profile perturbed with an anomaly like the one indicated in this study (-0.4 ppmv at 15 hPa). This modified profile has an ozone burden of 228.7 DU, only about 5 DU less than the original profile. The purple line shows the climatological profile with 0.5 ppmv subtracted between 40 and 1 hPa, effectively perturbing the bulk of the stratospheric ozone layer. The associated burden for this profile is 217.5 DU, a reduction of about 16 DU. Finally, the green line shows the results for 1 ppmv subtracted from the climatological profile over 40-1 hPa. In this case, the reduction in the burden is 32.6 DU, still about 6 DU less than the maximum anomaly in TCO reported in this paper. The key point is that the entire stratospheric ozone layer would have to be substantially perturbed to achieve an anomaly in total ozone of the magnitude asserted here. If indeed TCO was truly reduced by as much as 39 DU, then the decrease must have occurred in the troposphere rather than the stratosphere, in which case it is very unlikely to have been related to the Hunga eruption.

**Response 4**: Indeed, a direct comparison between IASI total column ozone measurements and MLS stratospheric ozone measurements is not appropriate, as the two instruments sample different atmospheric layers and use distinct observation geometries and techniques. We answer this comment in two main points:

- 1) The referee's comment prompted us to refine the selection criterion for better representation of the impacts of the sulfate aerosol clouds on ozone. Upon revisiting the MLS profiles selected by the initial criterion, we realized that it resulted in ozone profiles with anomalies occurring in two distinct pressure ranges. These profiles are displayed in Figure 1 below (page 4). Profiles exhibiting a significant ozone decrease around the 15 hPa level (approximately 28.5 km) are shown in the left panel, those with a decrease near the 26 hPa level (approximately 24.8 km) are shown in middle panel, while those with a decrease at both levels are shown in the right panel.

  This observation aligns with findings from Legras et al. (2022), particularly in their Figures 2 and 4e, which indicate that "the aerosol plume was initially formed of two clouds at 30 and 28 km" on January 15. CALIOP data in their Figure 2 suggests that by January 28, these clouds have

descended to approximately 27 and 25 km, indicating a loss of roughly 3 km in altitude after 14 days. Our MLS data, collected between January 16 and 23, captures the aerosol cloud during this interval, suggesting that the distinct ozone loss regions observed in Figure 1 are associated with these two aerosol clouds.

The initial selection criterion produced ozone anomalies across these distinct regions, regardless of whether they were caused by one aerosol cloud or the other, resulting in averaged anomalies that masked the individual impacts (hence the non-significant mean ozone anomaly that we found). To emphasize the ozone loss from a specific aerosol cloud, we modified our selection criterion. The new selection criterion represents a sub-ensemble of the previous one. As before, locations with water vapor mixing ratios exceeding 100 ppmv between 10 and 100 hPa were identified. The corresponding ozone profiles were then categorized into four groups: 1) negative ozone anomaly at 15 hPa (associated with the upper aerosol cloud), 2) negative anomaly at 26 hPa (associated with the lower aerosol cloud), 3) anomalies at both 15 and 26 hPa (both aerosol clouds), and 4) no negative anomaly. This method yields significant negative ozone anomalies in the first three cases. This refined criterion allows us to characterize and quantify the impact on each sulfate cloud on ozone levels over the Indian Ocean between January 15 and 23. This significant improvement has been added as one of the article's objectives within the introduction.

[Figure]

Figure 1: Anomaly profiles selected from a modified criterion, showing ozone decrease at the 15 hPa level (left), the 26 hPa level (middle) and at both levels (right).

- 2) Our goal is not to directly correlate IASI anomalies with MLS ozone anomalies at ∼15 hPa, nor to suggest that all ozone loss observed by IASI should match MLS measurements in magnitude. Previous research has already demonstrated stratospheric ozone loss following the eruption; our focus is to show how ozone observations can capture this impact within the stratosphere, where the bulk of the volcanic plume was injected. Because of the difference in observation geometries and techniques between IASI and MLS, the magnitude of the anomalies between both instruments might not match perfectly. However, they do match in showing clear ozone loss in the same pressure range. Our revised Figure 6 shows that ozone loss was particularly present in the 10–30 hPa range; but seems to extend up to ∼ 1 hPa. Similarly, IASI partial ozone columns in these pressure ranges show a clear ozone decrease. This result is illustrated in Figure 2 of

this document, which shows IASI ozone partial columns for different pressure ranges (specified in the upper right corner of each panel) for January 21 only, where the transient ozone depletion is best seen. These maps do not show ozone anomalies, but they do consistently show lower ozone levels over an extended area of the Indian Ocean. Note that the minimum and maximum values for the colorbar change for each panel. The pressure ranges shown in this figure (from 25.5 hPa to 6.6 hPa) are the ones where the impact on ozone is most visible, but it could be extended to lower and higher pressure levels. However, the troposphere does not seem to show signs of ozone reduction, at least not as much as the stratosphere. This is illustrated in Figure 3 of this document (page 6), where we show the tropospheric and stratospheric ozone columns for January 21. These columns were separated using IASI's estimation of the tropopause height. The region with the lowest stratospheric ozone column corresponds to the location of the ozone depletion event. However, the IASI pixels within this area do not coincide with the lowest tropospheric ozone values.

[Figure]

Figure 2: IASI partial column ozone for different pressure ranges (specified in the top right corner) for January 21, when the TODA is best captured.

[Figure]

Figure 3: Tropospheric (left) and stratospheric (right) ozone columns derived from IASI on January 21.

**Point 5**: I fail to see the point of much of the discussion in Section 3.5 on the transport of Hunga-influenced air masses over the Indian Ocean. Several previous studies tracked the early dispersion of the plume, including Millán et al. (2022), Legras et al. (2022), and Khaykin et al. (2022); moreover, its presence over Reunion within a week has already been established by Baron et al. (2023) and Evan et al. (2023). Even if the authors felt that further confirmation was needed, the HYSPLIT trajectory calculations would have been sufficient. Instead, maps of ERA5 PV are shown and the fact that they reveal no "marked discontinuity" in the PV field during this period is argued to be evidence that east-to-west isentropic transport at 600 K was possible. It is not clear what kind of atmospheric feature it is thought may have impeded such transport. An issue that is overlooked in this discussion is that ERA5 does not assimilate water vapor measurements, and thus it did not accurately capture post-eruption perturbations in stratospheric circulation, as discussed for MERRA-2 by Coy et al. (2022).

**Response 5**: The ERA5 PV maps were included to provide an alternative dynamical perspective alongside the HYSPLIT simulations. However, it is true that they do not significantly enhance the understanding of the dynamics and dispersion of the plume, as this has already been addressed in the studies cited by the referee. Therefore, as suggested, we have decided to exclude this result.

**Point 6**: Fundamentally, the raison d'être for this manuscript is not clear. Much of the analysis centers on evaluation of MLS and IASI ozone data through comparisons with DIAL and SAOZ measurements made at Reunion under background conditions. The statement is made "Based on the excellent correlation and agreement between satellite (MLS and IASI) and ground-based instruments (stratospheric lidar and SAOZ) over Reunion, it appears relevant to use satellite ozone products to investigate the changes in the distribution of ozone over the study region." But this is hardly a surprising result – both MLS and IASI are very well characterized data sets that have already been employed extensively in similar kinds of studies, including over the region in question. In fact, arguably the entire intercomparison portion of this study was unnecessary. On the other hand, validation of the satellite measurements – in particular those from IASI – under perturbed post-eruption conditions would have been valuable, but that was not possible using the Reunion data as noted above. Furthermore, this work seems to have provided no additional scientific insights beyond those already presented in the papers by Baron et al. (2023), Evan et al. (2023), and Zhu et al. (2023). Indeed, as the authors note, Baron et al. (2023) presented the lidar data and talked about the passage of the Hunga plume over Reunion. Evan et al. (2023) presented MLS (and other) data, including ozone, over Reunion during the same timeframe. Evan et al. (2023) and Zhu et al. (2023) elucidated the mechanisms giving rise to the observed low ozone (conclusions that this paper makes no attempt to add to). Nothing in this current study is new, other than the addition of total column ozone measurements, whose reliability in this particular region at this particular time has not been adequately addressed, as noted above.

**Response 6**: The intercomparison is essential for obtaining reliable results on ozone depletion during the Hunga event, given the specific nature of lidar data. It also validates the agreement between Maïdo DIAL and MLS ozone profiles, which had not been previously assessed. Calculating averages and variabilities for the background period (2013–2021) is crucial for accurately characterizing Hunga-related anomalies, as these cannot be derived without a clear and coherent representation of mean observations. This study adds a new perspective by identifying ozone decreases linked to each of the aerosol clouds, while also presenting IASI SCO maps for the first week following the eruption, which had not been done before. Furthermore, in contrast to previous studies–such as Baron et al. (2023) and Evan et al. (2023), which focused on static balloon profiles, or Zhu et al. (2023), which uses models–this research provides a global perspective for the Indian Ocean, relying solely on observations. The goal is not to elucidate the mechanisms behind low ozone, but to demonstrate how ozone anomalies are revealed in IASI and MLS data, and how they relate to the two initial sulfate aerosol clouds.

**Point 7**: Throughout the manuscript, numerical results are reported with what seems to me to be an unjustifiably high degree of precision. As just one example, the maximum anomaly in IASI TCO is stated to be -38.97 $\pm$ 25.39 DU. This "false precision" needs to be removed.

**Response 7**: Numerical results have been rounded to avoid false precision where necessary. However, double precision was maintained for AOD and sAOD results because their background values typically fall within the order of $10^{-1}$ or $10^{-2}$. Similarly, statistical results (such as coefficients of determination and regression slopes) were retained at double precision for accurate comparisons between datasets.

**Specific substantive issues:**

**Point L1**: Most of the aerosols of stratospheric significance were not emitted directly by the volcano, but rather arose through subsequent SO2 conversion to sulfate.

**Response L1**: We acknowledge the referee's point that mentioning the volcano's emission of aerosols could be misleading in the context of ozone loss, as only stating "aerosols" implies ash, which is less relevant compared to sulfates or SO2. We have therefore removed all references to this in the revised manuscript.

**Point L16**: It is not clear why the 2018 WMO Ozone Assessment is referenced for this general statement, rather than the most recent Report from 2022, which is cited elsewhere in this manuscript.

**Response L16**: This point has been amended.

**Points L30-42**: I have several comments on this paragraph:

- **Point 1**: It is stated in the first sentence that eruptions can influence tropospheric ozone, but the rest of the paragraph does not elaborate on this point at all, and it is not clear why it is relevant to this paper (unless the observed reduction in ozone is in fact occurring in the troposphere). The connection to tropospheric ozone needs to either be explained better or omitted altogether. Moreover, for clarity, in L36 "contribute to ozone depletion" should be "contribute to stratospheric ozone depletion".

- **Response 1**: Since our article does not concern tropospheric ozone, we recognize that adding this information may be misleading. We omitted this information in the new version of the article.

- **Point 2**: It is stated that eruptions release substantial amounts of aerosols, but the volcanic aerosols of most consequence for the stratosphere are those formed subsequently by the conversion of SO2 to sulfate, not those (e.g., ash) emitted directly by the volcanoes.

- **Response 2**: Similar to Point L1, this comment has been amended.

- **Point 3**: Literature citations are inadequate. It is not sufficient to cite only Tie and Brasseur (1995), Hofmann and Solomon (1989), and McCormick et al. (1995) for these points – many more references than these would be relevant in each case. At the very least, an "e.g.," needs to be added in front of all of these references.

- **Response 3**: Of course, we agree many more references could be cited to support paragraph. As suggested, we have added 'e.g.' to indicate that the references provided are examples.

- **Point 4**: It is not clear what is meant by the sentence "*Additionally*, reactive anthropogenic chlorine compounds may be enhanced in volcanically perturbed regions, leading to *further* ozone depletion" [emphasis added]– how is this different from "the activation of chlorine compounds on volcanic particles", "ozone depletion through heterogeneous chemistry", and "relationship between SO2 and chlorine in causing ozone decline post-eruption" that have already been mentioned in the preceding three sentences?

- **Response 4**: We thank the referee for pointing out this repetition. In the revised manuscript, this paragraph was shortened to avoid repetitive information.

**Point L59**: Wright et al. (2022) is not the most suitable reference for the Hunga aerosol perturbation; in addition to Sellitto et al. (2022), other appropriate work to cite for this point include Khaykin et al. (2022) and Taha et al. (2022) – both already cited elsewhere. Wright et al. (2022) is pertinent to the statement about the comparative energy release by Hunga, so it should be moved to that part of the sentence.

**Response L59**: We thank the referee for pointing this out. This point has been amended.

**Point L61**: Sellitto et al. (2022) is not really the best reference for the magnitude of the Hunga water vapor injection; it should be replaced here by Khaykin et al. (2022) and Vömel et al. (2022, https://doi.org/10.1126/science.abq2299).

**Response L61**: This point has been amended.

**Point L63-65**: The sentence "As a result of the main austral summer stratospheric circulation and the prevalent phase of the QBO, the first signs of the Hunga aerosol plume's passage over Reunion were noticed only 4 days after the main eruption" is problematic for several reasons. First, it's not clear what "main" means in this context (and the word "main" is used in three other places in the paragraph in reference to the eruption, so it is confusing). Second, the QBO is mentioned, but its influence is not made clear – was the QBO in an easterly or westerly phase at the time of the eruption, thus did it delay or accelerate the plume's arrival over Reunion? I believe that the authors mean that the prevailing westward flow brought the plume to the region of Reunion very quickly, such that it could be observed by instruments there within a short period of 4 days, but the wording is ambiguous and could be misinterpreted. Third, this is the first mention of Reunion in the main text. Since its importance to this work has not yet been established, it comes out of the blue and is a bit jarring. A lead-in sentence introducing Reunion and giving the reader a hint about its role in this work would be good. Otherwise, the relevance of the following information is unclear.

**Response L63-65**: The referee's interpretation of the sentence is correct. Our original intent was to highlight that the combination of the austral summer stratospheric eastward circulation and the QBO easterly anomaly allowed Hunga's plume to reach Reunion quickly. However, we chose to omit this detail because: 1) it was not an essential information (it was the only mention of the QBO in the article), and 2) including it would require additional, unnecessary context. Specifically, we would need to explain (see Stocker et al., 2024) that the QBO phase during the eruption likely induced an easterly anomaly above ~30 km, which accelerated the plume detected at ~34 km over Reunion four days later (Baron et al., 2023). Below 30 km, the QBO produced a westerly anomaly. We recognize that the word "main" was overused and have replaced it with more precise terms. Additionally, we have added a preceding sentence to better explain the significance Reunion in the context of our study, making it clearer for the reader.

**Point L67-79**: This discussion of the results of Evan et al. (2023) and Zhu et al. (2023) could be better organized – it jumps back and forth between heterogeneous and gas-phase reactions, making it difficult to follow. More importantly, some of the results of those studies are misstated. First, the Hunga-induced stratospheric cooling enhanced heterogeneous reaction rates but was not a factor in the rapid conversion of SO2 to sulfate aerosols, as is implied by the current wording (L69-70). (Also, a reference to the earlier paper by Zhu et al. (2022, https://doi.org/10.1038/s43247-022-00580-w) should be added for the impact of abundant OH from the Hunga hydration on the rapid sulfate formation.) Second, I was puzzled by the emphasis on photolysis of Cl2 (L76), as this is not part of the conclusions about gas-phase chemistry reported by Zhu et al. (2023) as is suggested, but then I found a similar sentence in the paper by Evan et al. (2023). However, Evan et al. are talking about the negative HCl anomaly arising from *heterogeneous* chlorine activation on sulfate. Their statement about Cl2 photolysis is made in connection with the colocated positive anomaly seen in daytime ClO. It is not correct that this is a "key gas-phase mechanism contributing to ozone loss".

**Response L67-79**: The information regarding Hunga-induced stratospheric cooling was not accurately conveyed and may have implied incorrect facts. We have rephrased it for clarity. We acknowledge the confusion caused by intertwining the explanation on 1) heterogeneous chemistry and 2) gas-phase chemistry. We have revised the text to provide a clearer, more distinct explanation of ozone loss. We acknowledge that we misinterpreted the information from Evan et al. (2022) and mistakenly included it in our study. As suggested, we have corrected this by deleting it from the "key gas-phase mechanism" list.

**Point L80-93**: Although the longer-term evolution of the Hunga water vapor and aerosol plumes is certainly interesting, it is not clear what relevance any of this has to the ozone distribution in the first week following the eruption, which is the focus of this study. If such discussion is retained, then it needs to be much more comprehensive in its summation of the existing literature on Hunga's radiative impact. Moreover, if the radiative effects from Hunga in subsequent months are covered here, then why are its chemical effects ignored?

**Response L80-93**: Indeed, the radiative impacts of the Hunga eruption are not relevant to our study. Since this information does not contribute to the analysis, we have removed this section.

**Points L95 and L136**: Livesey et al. (2008), which is a conference proceeding, is not a suitable reference for Aura MLS. The paper by Waters et al. (2006) is sufficient.

**Response L95 and L136**: We replaced the "Livesey et al. (2008)" reference by "Waters et al. (2006)" in these two lines.

**Point L98-99 and L102**: The phrase "dynamics of its advection" seems strange to me, since "dynamics" and "advection" are essentially synonyms. I suppose that the authors mean that they will show details of the plume's transport, but this should be clarified. Moreover, it is not clear what "its" in this sentence is referring to – grammatically it does not make sense.

**Response L98-99 and L102**: We thank the referee for pointing out this wording issue. This phrase has been revised in the article's objectives to more clearly convey the aim of showing the zonal displacement of the anomalies.

**Point L111-116**: It is not sufficient to simply state that the temporal and vertical resolution of the lidar data is "high". This information should be specified, especially the vertical resolution. Moreover, it is not appropriate to characterize a data set consisting of a total of 470 profiles obtained over a 9-year period as having "high" temporal resolution.

**Response L111-116**: In the revised version of the manuscript we included information regarding the lidar's vertical resolution, which varies with altitude. We omitted the information relative to temporal resolution as we recognize it was not appropriately articulated.

**Point L132-135**: Aspects of the MLS description need to be improved. The term "consistent measurement frequency" is ambiguous – initially I thought it was referring to spectral frequency. Thus, "spatial sampling" would be better. Also, it is not clear what "consistent" means in this context (and the MLS orbit ground tracks do differ slightly from one day to the next) – I would delete this word. It is not quite correct to refer to MLS as "a radiometer" (the instrument actually consists of seven radiometers); this is an unnecessary detail that it would be better to omit.

**Response L132-135**: The word "consistent" was omitted. Our intention was to convey that, unlike most ground-based instruments, satellite observations enable regular, constant data acquisition, thus providing continuous monitoring. We also removed the term "radiometer" for clarity.

**Point L136-140**: The recommendations of Millán et al. (2022) are slightly mischaracterized. That paper stated that the reliability of MLS measurements *inside* the Hunga plume (not "close to" it) was degraded in the first few weeks immediately following the eruption, because of the enormous enhancement in H2O concentrations. The statement that "MLS v4 relies only on profile retrievals from O2 signals whereas v5 also uses the H2O line" is unclear – this statement refers specifically to how information about *instrument pointing* (required for the retrieval of atmospheric composition profiles) is obtained in the two versions. This should be clarified. In addition, Millán et al. (2022) indicated that the standard MLS data quality screening protocols should NOT be implemented for the v4 H2O data during that initial post-eruption period. On the other hand, such filtering should still be performed for the O3 measurements, whose quality, as noted in L139, was unaffected by Hunga. The description of the MLS v4 data handling is unclear on this issue – since both the v4 and the v5 MLS Data Quality Documents are referenced in L157, the implication is that the v4 data (both H2O and O3) were screened, but the data filtering recommendations should be followed and the approach taken should be stated explicitly.

**Response L136-140**: We have replaced "close to" with "inside" to align with Millán et al. (2022). Additionally, we have clarified the explanation of the differences between the v4 and v5 profiles. In response to the referee's last point, we assure that we did not implement the screening protocols for the v4 H2O data, and we appreciate the referee for highlighting this oversight in our manuscript. While all screening procedures were properly applied to the v4 ozone profiles, as well as the v5 H2O and ozone profiles, they were not applied to the v4 H2O profiles.

**Point L144-146**: Some rearrangement of this discussion is needed. The numbers of MLS profiles

being examined here – 113 influenced by Hunga and 2190 in unperturbed conditions between 15 and 23 January – only make sense in the context of a limited region (since MLS measures ∼3500 profiles per day). However, the information that the comparison is restricted to a 5-degree radius around Reunion is not provided until much later in the paragraph.

**Response L144-146**: This discussion has been revised for clarity on this point.

**Point L147**: Is the standard deviation calculated separately at each pressure level or over the whole 10–100 hPa range? In other words, is the maximum value of 0.05 ppmv quoted here never exceeded at any single level? If the standard deviation is being calculated over the entire profile, then a larger value (at, say, the level of peak ozone anomaly) could be getting "diluted" in the overall profile standard deviation.

**Response L147**: The standard deviation mentioned in L147 refers to the maximum value observed at any pressure level within the 10–100 hPa range, rather than an average over this range. We clarified this information by changing the sentence to: "The maximum standard deviation did not exceed 0.1 ppmv in any of the 10 to 100 hPa pressure levels". Throughout the manuscript, we updated $1\sigma$ standard deviations and uncertainties to $2\sigma$, where possible, to reflect a 95 % confidence level. Thus, for this sentence, the standard deviation of 0.05 ppmv ($1\sigma$) was updated to 0.1 ppmv ($2\sigma$).

**Point L152**: I am interpreting the statement "repeated for both ascending and descending nodes" to mean that for the comparison between perturbed and unperturbed conditions, background values were calculated separately for the measurements obtained on the two sides of the orbit. Since ozone at these altitudes does not display large diurnal variations, I am wondering why it was considered necessary to derive both daytime and nighttime background profiles.

**Response L152**: We included the phrase "repeated for both ascending and descending nodes" to provide a comprehensive description of our methodology for deriving background profiles. However, this was not meant to imply that we processed the ascending and descending profiles separately. In fact, we combined all profiles from both nodes within a single day to produce an average profile, rather than distinct daytime or nighttime profiles. To clarify, we have reworded the sentence to: "All ozone and water vapor profiles within a 5-degree radius of each of the January 2022 impacted profiles were collected, regardless of the satellite's ascending or descending node".

**Point L206-207**: While the three references cited in this sentence are pertinent to the statement that persistent Hunga effects were confined to the stratosphere, they are completely unsuitable for the point that they immediately follow, which is that the stratospheric circulation is stable and stratified. In fact, no such statement is needed to justify the use of trajectory calculations (a very common technique). I recommend deleting everything in this sentence up to "we used HYSPLIT".

**Response L206-207**: This point has been amended.

**Point L218-219**: Similarly, PV is so widely used now for characterizing isentropic transport in the stratosphere that not only is such a list of citations unnecessary, but also the one provided is so seemingly arbitrary and self-referential that it does more harm than good. This sentence should be deleted.

**Response L218-219**: This point has been amended.

**Points Section 2.4**:

- **Point 1**: MLS retrievals are output on a pressure grid, whereas the comparison with the lidar

measurements uses altitude as a vertical coordinate. How the MLS measurements are placed onto an altitude grid needs to be explained.

- **Response 1**: To place MLS measurements on a pressure grid, we downloaded MLS geopotential height profiles, which associate geopotential height to each MLS pressure levels. Since geopotential height can be easily converted into height above mean sea level (using an approximation of variation of gravity with altitude), we used geopotential height profiles to associate a height above mean sea level (i.e. altitude) to each pressure level. This clarification was added into the manuscript.

- **Point 2**: As noted above, the vertical resolution of the lidar measurements is not given, but I presume that it is much higher than that of the MLS ozone profiles (which is ∼2.5–3 km in the lower stratosphere). Simply sampling the finer profile at the MLS retrieval surfaces is not the best approach. To make a truly fair comparison between high-vertical-resolution profiles and coarser-resolution MLS data, it is necessary to follow the guidance in the MLS Data Quality Document to apply the MLS averaging kernels to and perform a leastsquares "smoothing" of the high-resolution data set (Sections 1.8 and 1.9 of the MLS Quality Document, respectively). Although performing such a procedure may not make a substantial difference to the bottom-line results, this issue is nevertheless worth some investigation. At the very least, an experiment in which the lidar profiles are smoothed over ∼1.5 km (i.e., boxcar smoothing) should be conducted to explore the impact on the comparisons with MLS.

- **Response 2**: As suggested by the referee, we proceeded to modify the methodology of the comparison between DIAL and MLS profiles by incorporating MLS averaging kernels. Doing so reduced the number of points available for the comparison and changed the appearance of the mean relative bias profile (Figure 4 of the manuscript). However, it did not undermine the agreement between both datasets. In fact, this approach has improved statistical metrics, notably reducing the RMSD and increasing the correlation.

- **Point 3**: The notation in the numerator of Equation (1) is confusing: O3 MLS - O3 DIAL(z)(z). Why is the first term not written O3 MLS(z) (as in L240), rather than putting both "(z)"s at the end?

- **Response 3**: We thank the referee for pointing out this typo. The equation was corrected in the revised version of the manuscript.

**Point L267 and Figure 1**: It is difficult to judge where 5°S and 25°S are located on these panels, as the x-axis latitude grid is odd. Instead of placing the vertical lines at the x-axis major tick marks, it would be more helpful to draw vertical lines at 5°S and 25°S. I also request that minor tick marks be added to both x- and y-axes.

**Response L267 and Figure 1**: Figure 1 was updated to show regular tick labels and marks, as suggested. We also added vertical lines at 5°S and 25°S to better visualize the location of our study region.

**Point L281-282 and Figure 2**: I do not understand what "This multi-year average represents an average of AOD data which is grouped into months, irrespective of the years" means – does the blue line in Figure 2 show the overall average over the 2003–2021 period, or is it just the January mean over all the years in that period? The uncertainty error bars on the red and black lines in Figure 2b are nearly impossible to see, even when zooming in on the panel. Greater contrast between the lighter and darker shades is needed. I also request that minor tick marks be added to both x- and y-axes.

**Response L281-282 and Figure 2**: The blue line represents the January mean over 2003–2021. This precision was added into the manuscript. We recognize that the error bars were difficult to see properly and tried a different approach. Minor tick marks were also added to both x- and y-axes.

**Point L320-332 and Figure 4**: This discussion requires clarification in several respects:

- **Point 1**: Saying "the bias decreases to -3.73" makes it sound as though the bias has gotten smaller, whereas it has changed sign but actually is larger in magnitude.

- **Response 1**: The revised comparison methodology, now incorporating averaging kernels, has slightly altered the results. The average bias in this altitude range is now $0.24 \pm 2.12$ %.

- **Point 2**: Given the large oscillations in the differences below 20 km, the average bias carries little meaning, so there is no real benefit in stating it.

- **Response 2**: With the new mean bias profile using averaging kernels, differences below 20 km are significantly reduced and we continue to report these values.

- **Point 3**: Livesey et al. (2022) discuss the presence of known systematic vertical oscillations in the MLS ozone retrievals in the UTLS; these likely play a role in the differences below 20 km seen here. However, they do not explain the rather large relative bias at 20 km.

- **Response 3**: We thank the referee for pointing to this reference.

- **Point 4**: Given the stated caveats on the lidar data, it is not justifiable to say "the MLS mean bias profile seems to under-estimate ozone concentrations by 20.73±1.89%". All that can be said with confidence is that there is a relative bias of ∼21% between the two data sets.

- **Response 4**: The revised comparison methodology, now incorporating averaging kernels, has slightly altered the results. The relative difference pointed out by this comment is now considerably reduced and no longer mentioned.

- **Point 5**: Similarly, the linear regression shows that "MLS profiles tend to slightly over-estimate ozone concentrations" *relative to DIAL*.

- **Response 5**: The revised comparison methodology, now incorporating averaging kernels, has slightly altered the results. The linear regression slope is now below unity, so we simply state that "MLS profiles tend to slightly under-estimate ozone concentrations relative to DIAL."

- **Point 6**: I request that minor tick marks be added to both x- and y-axes.

- **Response 6**: Minor tick marks were added to both x- and y-axes.

**Point L345-371 and Figure 5**: I have several issues with Figure 5 and the accompanying discussion:

- **Point 1**: The east-to-west movement of the plume during this time period was shown also by Millán et al. (2022).

- **Response 1**: Millán et al. (2022) was among the first to publish on HTHH material, focusing primarily on H2O, with some attention to SO2 and HCl, but no mention of ozone. In contrast, our work examines H2O, SO2, and O3 during the first 10 days after the eruption.

- **Point 2**: Schoeberl et al. (2022) is not an appropriate reference for the rapid conversion of SO2

to sulfate aerosol – Zhu et al. (2022) and Asher et al. (2023) are better citations for that.

- **Response 2**: We have replaced the Schoeberl et al. (2022) reference with the suggested alternatives.

- **Point 3**: It is not clear what "important" means in the context of the negative ozone anomaly.

- **Response 3**: We recognize that the signification of "important" is vague. We simply omitted this word in the revised version.

- **Point 4**: I am confused about what the error bars on the daily minimum TCO and maximum TCO anomaly values represent – please clarify.

- **Response 4**: The error bars on TCO anomalies are measurement uncertainties. This information was added into the manuscript.

- **Point 5**: Why is the IASI TCO anomaly in January 2022 (which is derived relative to the 2014–2021 IASI climatology) being compared to the SAOZ climatological January TCO rather than to the IASI January climatology?

- **Response 5**: The SAOZ climatology was selected to emphasize local data collected over a significant period (1994–2021). For completeness, as suggested, we have also included the January IASI climatology.

- **Point 6**: The color palettes used for the TCO maps need improvement, especially the anomaly one. For one thing, the color bar saturates below -10 DU, making it impossible for the reader to judge the ∼20–40 DU maximum negative anomalies noted in the text. Although a bright blue color is used for those largest negative anomalies, the contrast between it and the color used for the negative anomalies with magnitude smaller than 10 DU is too weak to be readily discernible without extreme magnification. In addition, positive anomalies should be easily distinguishable from negative ones – as it is now, the zero line falls in the middle of a continuum of bluish-greenish colors.

- **Response 6**: We have revised the color palettes to address the issues raised. The IASI TCO anomaly palette now enables clear distinction between positive and negative anomalies. Additionally, the anomaly maps now display only significant anomalies at the $2\sigma$ level, with reduced scatter point sizes to enhance map readability.

- **Point 7**: The overlaid contours depicting SO2 are red, not blue.

- **Response 7**: We thank the referee for pointing out this typo. This point has been amended.

- **Point 8**: It makes little sense to highlight the MLS profiles with high H2O values in green, when the orbit tracks are overlaid on SO2 maps plotted using a yellow-green color palette.

- **Response 8**: We now use a darker color of blue for the Hunga-impacted MLS profiles.

- **Point 9**: The black and white star denoting the location of Reunion is very difficult to spot.

- **Response 9**: The size of the star was increased and Reunion is easier to spot.

**Point L379**: While the largest anti-correlation may coincide with the maximum ozone anomaly, the r value a couple of levels below is nearly as large.

**Response L379**: The results in Figure 6 have been updated with the new criterion. However, as the correlation results between water vapor and ozone yielded low values ($|r| < 0.6$) and were not essential to our study, we decided to omit them. The anti-correlation maxima identified by the referee concerning the previous version of Figure 6 are located at 15 hPa and 26 hPa, directly concerning the upper and lower aerosol clouds. Within the relevant pressure range (10–100 hPa), the new maximum anti-correlation (not shown) is at the 15 hPa level, corresponding to the highest sulfate aerosol cloud. The correlation line for the lowest sulfate aerosol cloud has a maximum at 26 hPa. This is no surprise, as the refined criterion was designed to better capture ozone depletion at these levels. Separating the impacts of each aerosol cloud enhances the accuracy and significance of these results.

**Point Figure 7**: Why are the trajectories plotted using such thick lines? The individual trajectories are completely indistinguishable. Perhaps that is the point, but it could still be easily made using differently colored lines of more moderate thickness.

**Response Figure 7**: We reduced the trajectory line widths and applied a color gradient to differentiate between them, and we hope this has improved the figure in the revised version.

**Point L392-402 and Figure 8**:

- **Point 1**: Of what possible relevance is the location of the \*global\* average latitude of the sub-tropical barrier on these dates? In the context of this study, only its location in the region of the Indian Ocean is important.

- **Response 1**: The discussion in Section 3.5 and Figure 8 has been omitted, and this point has been amended.

- **Point 2**: The anomaly does not stay completely north of the subtropical barrier – some solid green dots are clearly present poleward of the barrier on 18 and 19 January (panels (d) and (e)).

- **Response 2**: The discussion in Section 3.5 and Figure 8 has been omitted, and this point has been amended.

- **Point 3**: It is stated that the anomaly exits the region on 22 January, but a couple of green dots still appear on the map on both 22 and 23 January, and on 23 January (panel (i)) they fall south of the subtropical barrier.

- **Response 3**: The discussion in Section 3.5 and Figure 8 has been omitted, and this point has been amended.

- **Point 4**: The red line marking the subtropical barrier is dashed, not solid as implied in the caption.

- **Response 4**: The discussion in Section 3.5 and Figure 8 has been omitted, and this point has been amended.

- **Point 5**: Using red for the subtropical barrier is a poor choice since it is overlaid on contours of similar color (ranging from purple to orange).

- **Response 5**: The discussion in Section 3.5 and Figure 8 has been omitted, and this point has been amended.

- **Point 6**: As in other plots, it is very difficult to spot the black and white star indicating Reunion.

- **Response 6**: The discussion in Section 3.5 and Figure 8 has been omitted, and this point has been amended.

**Point L407**: As noted earlier, most of the aerosols of stratospheric significance were not emitted directly by the volcano, but rather arose through SO2 conversion to sulfate.

**Response L407**: Similar to Point L1, this comment has been amended.

**Point L417**: It is not appropriate to say "the ozone reduction occurred at the level of the ozone layer". The ozone layer is a broad feature, extending over roughly 15–35 km altitude. MLS showed substantial (but not significant) anomalies only in a narrow layer around 15 hPa.

**Response L417**: We thank the referee for highlighting this inappropriate wording. We now specify the pressure level of maximum ozone anomaly instead.

**Point L431-441**: The statistical quantities described here are common and widely used, so I am not convinced that this Appendix is really needed.

**Response L431-441**: Statistical equations were removed from the Appendix.

**Minor points of clarification, wording suggestions, and grammar / typo corrections:**

We thank Referee 1 for their careful review and for suggesting numerous grammar and typo corrections. All of these minor points have been addressed in the revised version.

---

## Author Comment (AC2)

**Response to Referee 2 Comments**

We would first like to express our thanks and appreciation to Referee 2 for their review. The comments identified flaws and unclear points in the article, providing an excellent opportunity to improve its overall quality.

Our responses follow the structure of the review document and are divided into three sections: 1) responses to major comments, and 2) responses to minor comments. Referee comments are written in black and authors answers are in blue.

**Major comments:**

**Point 1**: The authors present background ozone profiles from different months observed by DIAL and explain the variations in altitude with the highest ozone concentrations in Section 3.2. However, this section seems less relevant to the main topic. Additionally, Figure 3 is not particularly informative, as it merely displays typical tropical ozone concentrations in an altitude-month contour.

**Response 1**: We acknowledge the referee's perspective and have revised Figure 3 for better coherence with our study. It now presents mean January lidar and an average MLS ozone profile to illustrate background ozone levels at Reunion under unperturbed conditions, along with a representative January MLS ozone profile for the entire Indian Ocean. The close similarity between the lidar and MLS profiles over Reunion demonstrates strong correlation between the datasets. Furthermore, the similarity between the Reunion and Indian Ocean MLS profiles suggests that the Reunion data can be considered representative of the Indian Ocean. The new Figure 3 shows a good comparison between the two instruments and supports the use of MLS data to represent ozone levels across the entire Indian Ocean for this study.

**Point 2**: The MLS and IASI data are well-validated and widely used in the ozone community, so it may be unnecessary to validate them in this paper. However, it's not a negative addition. Besides, the standard deviation calculation in Figure 4a is inappropriate. The authors calculate the standard deviation for the mean relative bias, which decreases as the number of samples increases. Instead, they should calculate the standard deviation for the relative bias itself (as done in Figure 6), which would represent the variation of each individual relative bias.

**Response 2**: It is essential to ensure that (1) background conditions are accurately captured by both MLS and IASI, and (2) neither instrument exhibits bias in January 2022 due to the event's exceptional nature. It is necessary to ensure that the instruments used over the Indian Ocean perform consistently with local instruments during a recent period, confirming the absence of retrieval degradation and assessing potential instrumental bias. The comparison between the lidar and MLS profiles serves more as a demonstration of consistency between different observations (giving more robust confidence in the results) rather than a formal data validation. OMI data was excluded due to significant perturbations in its UV radiance measurements caused by stratospheric sulfate aerosols during the Hunga event. Instead, we relied on IASI data which has no known issues in its UV radiance measurements, although it has not yet been validated for this event. Regarding the referee's last point, Figure 6 aims to assess the dispersion of ozone anomaly profiles, where using the "classic" standard deviation is appropriate. However, in Figure 4, we quantify the uncertainty of the mean relative bias, making the use of standard error more appropriate. Statistically, the relative bias results below ∼15 km are based

on fewer lidar profiles (< 131), compared to ~25 km with 470 profiles, necessitating a correction for this difference to ensure a valid comparison.

**Point 3**: The biggest issue is with Figure 5 and its description: SO2 is not a reliable tracer for volcanic gases and particles. As the authors themselves mention, "the rapid conversion of SO2 molecules to sulfate particles" occurs. Therefore, there are two distinct possibilities for a region with low SO2 concentrations: (1) volcanic gases and particles did not reach this region, or (2) volcanic gases and particles did reach the region, but the SO2 was converted to sulfate. These two possibilities are entirely different and need to be clearly distinguished.

**Response 3**: SO2 is in fact a reliable tracer for volcanic gases. In the stratosphere, a concentration above $1\sigma$ of the SO2 background is very likely to be linked to a volcanic event and thus is a relatively good tracer for an eruption. That being said, SO2 is short lived and quickly converted to sulfates during the Hunga event because of the presence of water vapor. However, in Figure 5, SO2 is not used to argue for the presence or absence of SO2 or sulfates; it is employed solely to localize the plume and associated ozone anomaly, and this information is complemented by Hunga-impacted profiles provided by MLS. Because of the austral summer stratospheric dynamics, all of the regions from Figure 5 located between the HHTH and Reunion must have been impacted by SO2, sulfate or a combination of both.

**Point 4**: The authors mention a "correlation between ozone, H2O, and SO2." While the correlation between H2O and SO2 is visible, the correlation between ozone and the other two gases is unclear in Figure 5. In addition, the significant ozone anomaly (shown in blue) appears throughout the region, even before the Hunga aerosol transport (e.g., Figure 5 b1, b2), making it difficult to attribute the ozone anomaly specifically to the Hunga eruption.

**Response 4**: Indeed, in Figure 5, the correlation is best seen between H2O and SO2 (in the right-most column). As an answer to referee n°1, we have revised Figure 5 to only show IASI significant anomalies at the $2\sigma$ level, effectively removing most of the anomalies throughout the region. The revised Figure 5 now better shows the anomalies linked to the HTHH event.

**Point 5**: The largest correlation between ozone and water vapor is -0.68 in Figure 6c, which is not significant.

**Response 5**: The correlation we used in panel c) was purely indicative. The results in Figure 6 have been updated with a new criterion. However, as the correlation results between water vapor and ozone yielded low values ($|r| < 0.6$) and were not essential to our study, we decided to omit them.

**Minor comments:**

**Point 1**: In the abstract, introduction and conclusion, the authors state that "Hunga volcano eruption released significant amounts of aerosols, water vapor (H2O) and a moderate quantity of sulfur dioxide (SO2) into the stratosphere." However, volcanoes rarely release aerosols directly. Instead, sulfate aerosols form from the oxidation of SO2.

**Response 1**: We acknowledge the referee's point that mentioning the volcano's emission of aerosols could be misleading in the context of ozone loss, as only stating "aerosols" implies ash, which is less relevant compared to sulfates or SO2. We have therefore removed all references to this in the revised manuscript.

**Point 2**: Line 20: The term "halons (Br)" is vague. Halons are not equal to Br, so further explanation is needed. For example, it could be revised to "halons (including Br)".

**Response 2**: The text was revised as suggested.

**Point 3**: Line 28: The capitalization of "Volatile Organic Compounds" is unnecessary.

**Response 3**: The capitalization of "Volatile Organic Compounds" was dropped.

**Point 4**: Line 34: See my first minor comment.

**Response 4**: Amended.

**Point 5**: Some abbreviations are not explained. For example, "PSC" at Line 50 and "QBO" at Line 64 should be defined upon first use.

**Response 5**: Amended.

**Point 6**: Line 315: The phrase "altitude level, date, and time" is vague. Do you mean the time of day?

**Response 6**: The term "time" referred specifically to the time of day, which we have clarified in the revised manuscript.

**Point 7**: Figure 4: The statement "p-value = 0.0" is not meaningful in this context. The authors should either remove the p-value or provide a more meaningful value in Figure 4.

**Response 7**: We removed the p-value from Figure 4.

**Point 8**: Line 346: The unit of "DU" seems strange for SO2.

**Response 8**: The native IASI unit for SO2 is $mol.m^{-2}$. We opted to retain SO2 representation in DU to maintain consistency with ozone measurements.

---

## Author Comment (AC3)

**Response to Referee 3 Comments**

We would first like to express our thanks and appreciation to Referee 3 for their review. The comments identified flaws and unclear points in the article, providing an excellent opportunity to improve its overall quality. In the following, Referee comments are written in black and authors answers are in blue.

**Comments:**

**Point 1**: Several papers were cited by the authors which analyzed the Hunga volcanic eruption and its influences. The authors should describe better their contributions on the ozone study after the passage of Hunga plumes. What is the difference for this work with the others? The objective of the paper is not very clear.

**Response 1**: The paper's objective has been clarified both in the manuscript and here. Our aim is to provide a fresh perspective on the first 10 days following the Hunga eruption, using satellite ozone data from MLS and IASI to track the daily movement of the ozone anomaly across the Indian Ocean and identify the formation of a distinct ozone-depleted area by January 21. We have made it clear that our work is the first to use and validate IASI data for this event. Furthermore, by refining our selection criterion, we could characterize the effects of both Hunga aerosol plumes, as discussed by Legras et al. (2022), on ozone levels. Figure 6 was revised to illustrate ozone anomalies associated with each individual aerosol cloud. This new result suggests that ozone depletion was localized and followed the movement of both clouds, strengthening the link between ozone loss and sulfate aerosols.

**Point 2**: In the Introduction, the chemical influence of chlorine and sulfur compounds in ozone should be discussed further with showing some chemical reactions. After all, the objective of the paper is to show the influence of the volcanic plume on atmospheric ozone.

**Response 2**: To further discuss the chemical influence of chlorine and sulfur compounds on ozone, we added appropriate chemical reactions, as suggested.

**Point 3**: In the methodology, it should be explained what location (latitude, longitude, grid used for satellite data, . . . ) the paper analyzed. This should be put in the beginning. The methodology should be expanded to explain better the instruments and the methodology used.

**Response 3**: Suggested information was added into the manuscript.

**Point 4**: In item 2.1 - Ozone Measurements, for better understanding the work, it is needed to create separate subtitles to describe each instrument, their data and its problems with the Hunga plume.

**Response 4**: This point has been addressed by dividing Section 2.1 into subsections, with one dedicated to each instrument.

**Point 5**: In the results part, it is important to change Figure 5. How it showed the ozone total column and ozone anomaly. It was not possible to see the reduction discussed in the paper based only in this figure.

**Response 5**: Figure 5 was revised to better show the transient ozone depletion area. The revised

version of Figure 5 now only shows significant IASI anomalies at the $2\sigma$ level, allowing to better constrain the anomaly. Clarity of the information within the text was also improved.

**Point 6**: L.16: Why cited WMO (2018) and not WMO (2022)?

**Response 6**: The WMO reference has been updated to WMO (2022).

**Point 7**: L. 17: A reference for the ozone absorbing range must be cited

**Response 7**: An appropriate reference was given for the ozone absorbing range (Oprhal et al., 2016).

**Point 8**: L. 20 – 21: The authors wrote "In the past decades. . . ", however the only reference cited is from 1996.

**Response 8**: Molina and Rowland (1974) and Solomon (1988) were included to reinforce this point.

**Point 9**: L. 15 – 29: The text should be separated in two paragraphs for a better understanding in stratospheric ozone and tropospheric ozone.

**Response 9**: As suggested, the text was dived to clearly separate tropospheric ozone from stratospheric ozone.

**Point 10**: L. 30 – 41: In this part, the authors described the influence of some volcanic eruptions on ozone. However, only two are cited. They were important eruptions, but other eruptions and new studies should be analyzed also. Calbuco was only cited in line 52.

**Response 10**: Eruptions of Fuego (1974), Cerro Hudson (1991), Calbuco (2015), and their relevant references have been added to the discussion on L. 30 – 41 to provide a broader context on volcanic contributions to stratospheric ozone impacts.

**Point 11**: L. 63 – 64: "As a result of the main austral summer stratospheric circulation and the prevalent phase of the QBO. . . " – Explain better this main circulation and what phase of QBO were acting in the period analyzed.

**Response 11**: Our original intent was to highlight that the combination of the austral summer stratospheric eastward circulation and the QBO easterly anomaly allowed Hunga's plume to reach Reunion quickly. However, we chose to omit this detail because: 1) it is the only mention of the QBO in the article, and 2) including it would require additional, unnecessary context. Specifically, we would need to explain (see Stocker et al., 2024) that the QBO phase during the eruption likely induced an easterly anomaly above ~30 km, which accelerated the plume detected at ~34 km over Reunion four days later (Baron et al., 2023). Below 30 km, the QBO produced a westerly anomaly. We recognize that the word "main" was overused and have replaced it with more precise terms. Additionally, we have added a preceding sentence to better explain the significance Reunion in the context of our study, making it clearer for the reader.

**Point 12**: L. 83 – 84: Explain "the remaining aerosol plume consisted of two concentrated patches"

**Response 12**: This information was revised and clarified within the new version of the manuscript. The study by Legras et al. (2022) showed that the Hunga eruption injected two distinct volcanic clouds into the stratosphere, initially positioned at ~30 km and ~28 km altitude on 15 January. These clouds remained separated throughout our study period, gradually descending to ~27 km and ~25 km by 28 January.

**Point 13**: L. 94 – 96: What kind of observations were described in the paper? What species were analyzed from satellite and ground-based observations?

**Response 13**: This information is included in the following lines.

**Point 14**: L. 100 – 103: It is not necessary.

**Response 14**: As suggested by the referee, these lines were suppressed in the revised version of the manuscript.

**Point 15**: L. 147: "The maximum $1\sigma$ standard deviation found in the stratosphere was close to 0.05 ppmv". What is the mean ozone mixing ratio for this region? How much percentage it represents?

**Response 15**: In the stratospheric region mentioned at L147 (10—100 hPa), the mean ozone mixing ratio across all co-located v4 and v5 profiles not meeting the selection criterion is 4.4 ppmv. The $2\sigma$ standard deviation of 0.1 ppmv reflects a negligible 0.02% variation relative to this mean, and this information has been added to the manuscript. Throughout the manuscript, we updated $1\sigma$ standard deviations and uncertainties to $2\sigma$, where possible, to reflect a 95 % confidence level. Thus, for this sentence, the standard deviation of 0.05 ppmv ($1\sigma$) was updated to 0.1 ppmv ($2\sigma$).

**Point 16**: L. 153: The "established criterion" is not clear.

**Response 16**: The formulation was changed to improve clarity.

**Point 17**: L. 211: What is the lat, lon for the location of Saint-Denis?

**Response 17**: This information is now added into the manuscript. The location of Saint-Denis is approximately 20.90° S; 55.48° E.

**Point 18**: L. 231: What is the location of the lidar site? Lat, lon.

**Response 18**: This information is now added into the manuscript. The coordinates of the lidar site are: 21.08° S; 55.38° E.

**Point 19**: L. 255: Describe first what panel (a) in Figure 1 represents.

**Response 19**: The information was changed as suggested.

**Point 20**: L. 309-316: In Fig. 4 were described two kinds of data. In this paragraph, the analysis was confused. Please, rewrite it.

**Response 20**: We apologize for the confusion. This paragraph was revised as suggested.

**Point 21**: L. 322: Define SNR.

**Response 21**: SNR is defined in section 2.1 as "signal-to-noise ratio".

**Point 22**: L. 333: Standardize RMSD or RMSE.

**Response 22**: We thank the referee for pointing this typo within Figure 4. We have corrected "RMSE" to "RMSD" in the figure.

**Point 23**: L. 345: "All maps are overlaid with blue contours of the SO2 plume...". It should be red contours, not blue.

**Response 23**: We thank the referee for pointing this typo. This was corrected in the revised manuscript.

**Point 24**: L. 354: How could be seen "222.87" DU if the figure scale begins in 240 DU? Figure 5 needs to change the data range from legend and colors pattern, because it was impossible to see the reduction in ozone total column and the anomalies showed by the red areas.

**Response 24**: The colorbar was changed to clearly show the reduction in column ozone, improving the visualization of the observed anomalies.

**Point 25**: L. 352 – 371: Where is the data from this paragraph? It was not possible to see the number cited based on Fig 5.

**Response 25**: This paragraph references IASI ozone observations and anomalies presented in Figure 5. In the updated Figure 5, only significant anomalies (at the $2\sigma$ level) are displayed, enhancing visibility of the specific anomalies discussed in the text. Following Referee 1's suggestion, all mentioned anomalies are now indicated along with their corresponding lat-lon coordinates for easier localization on the figure.

**Point 26**: L. 701: For Zhu et al. (2023), it should be cited as the doi for final version of the paper, not the preprint doi. The correct doi is: https://doi.org/10.5194/acp-23-13355-2023.

**Response 26**: We thank the referee for providing the updated reference.

---

## Referee Report (RR1)

**Re-review of "Evidence of a Transient Ozone Depletion Event in the Early Hunga Plume Above the Indian Ocean" by Millet et al.**

The manuscript has been substantially revised in response to referee comments. In general, the authors have done a good job in responding to the points raised by the reviewers, and the manuscript has been considerably improved. However, several new issues have been introduced through the revision process, some comments on the previous draft have still not been adequately addressed, and a few things that escaped my notice during the initial review have become more obvious now that the manuscript has been cleaned up. Although many of my comments on the revised draft are minor corrections that should be easy to deal with, I do have a number of more major substantive concerns that need to be addressed before the paper is accepted for publication.

General comment:
One major comment from my previous review that has not been resolved in the revised draft is the magnitude of the IASI ozone anomaly and its apparent discrepancy with MLS-based estimates. The authors have redone their analysis of MLS measurements and now find average negative anomalies in ozone of 0.7 ± 0.5 ppmv at 17–12 hPa and 0.6 ± 0.5 ppmv at 26–32 hPa. Unlike in the original manuscript, these anomalies are now barely significant at $1\sigma$. However, the IASI analysis was also redone, and the maximum (and highly significant) TCO anomaly of 40.1 ± 4.8 DU is slightly larger than it was before. A stratospheric column ozone (SCO) anomaly is now also calculated; its maximum value (also highly significant) is 49.9 ± 4.7 DU. TCO / SCO anomalies of this magnitude will be met with skepticism by many readers. As demonstrated in my previous review, even an anomaly as large as 1 ppmv applied uniformly over the entire range from 40 to 1 hPa (the bulk of the stratospheric ozone layer) would not come close to producing an SCO anomaly of 50 DU. In their response, the authors state that "a direct comparison between IASI total column ozone measurements and MLS stratospheric ozone measurements is not appropriate, as the two instruments sample different atmospheric layers and use distinct observation geometries and techniques". This statement misses the point – ozone is ozone, no matter who is measuring it. For convenience, the plot included in my previous review was based on MLS measurements, but it did not depend on them – the same analysis could be done with any ozone profile. I encourage the authors to do such an exercise themselves – take an ozone profile (from anywhere), compute the SCO from it, and then calculate the SCO anomaly based on perturbations to that ozone profile of different amplitudes. This should give a sense of the magnitude and vertical extent of the perturbation necessary to bring about an SCO anomaly of 50 DU. I feel that some discussion about the credibility of the large column ozone anomalies estimated from IASI data – and their inconsistency with the MLS-based estimates – should be added to the text.

Specific comments and questions:
Both major substantive issues and minor points of clarification, wording suggestions, and grammar / typo corrections are listed together in sequential order through the manuscript. Line numbers refer to the "clean" version of the revised manuscript, not the tracked-changes file.

- Abstract: The authors need to be mindful that many readers will look to the Abstract to get a basic sense of the paper (and they may not go beyond that). Therefore the Abstract needs to do a much better job of summarizing the study and clearly highlighting its novel aspects. For example, the fact that this is the first presentation of IASI data in the context of the Hunga eruption should be emphasized here. In addition, the finding that the reduction in ozone appears to have been confined to two distinct layers associated with two separate aerosol clouds is one of the few new aspects of this study and should be more clearly articulated.
- L5: delete "The" in front of "Ozone"
- L9: Given the poor vertical resolution of IASI data, it is not really appropriate to refer to IASI and MLS "profiles" together in the same sentence. I suggest replacing "profiles" with "measurements" in this line.
- L11-12: The TCO result is not actually covered in the main body of the paper, but I think it should be – see my more detailed comments on this point below.
- L18: There is no need to define the acronym "UVR" as it is not used again in the manuscript.
- L34-35: such as that --> such as those; add a comma after "Calbuco (2015)"
- L44: Why is the word "implied" used here? Stratospheric ozone losses and radiative changes have been documented following volcanic eruptions, as noted in this manuscript.
- L52: clouds (PSCs) volume --> cloud (PSC) volume
- L71: An early paper discussing the influence of the excess humidity from Hunga in accelerating conversion of $SO_2$ to sulfate aerosols by Zhu et al. [2022, Comm Earth & Environ, 10.1038/s43247-022-00580-w] should also be cited for this point.
- L72: The paper by Sicard et al. has now been published, so the citation needs to be updated both here in the text and in the reference list. Moreover, other papers should also be cited for the Hunga-induced stratospheric cooling, such as those by Sellitto et al. (2022), Coy et al. (2022), and Schoeberl et al. (2022, GRL, 10.1029/2022GL100248).
- L75-83: I do not think that the listing of stratospheric chemical reactions has added useful information to this paper. I understand that in their comments on the previous draft one of the other referees suggesting discussing in more detail the influence of chlorine and sulfur compounds on stratospheric ozone, including showing some chemical reactions. But in response to that comment the authors have simply listed the set of "key heterogeneous reactions" given in the review paper by Solomon et al. (1999), with absolutely no accompanying text to put these reactions into context or give a sense of which ones are generally more important following volcanic eruptions. Zhu et al. (2023) and Evan et al. (2023, in the supplementary material), both already cited in the manuscript, provide a detailed description of the post-Hunga heterogeneous chemical reactions inside and outside the plume. In addition, Wilmouth et al. (2023, PNAS, 10.1073/pnas.2301994120) and Santee et al. (2023, JGR-A, 10.1029/2023JD039169) discuss the stratospheric chemical processing in the months following the eruption. Thus I feel that the authors would be better off deleting the material in these lines and simply referring readers to the lengthy explanations in those previous papers. If the authors want to retain these equations in the paper, then more in-depth discussion of how they are relevant needs to be added to the text.
- L88-89: It is not appropriate to say that Evan et al. "documented" a doubling of ozone loss via $O_3+Cl$ – they merely reported the results shown by the modeling study of Zhu et al. (2023). Moreover, while the rate of that particular reaction did double, the rates of other reactions

changed by even greater amounts, so it is not clear why that one has been singled out. It would be better to make a more general statement that the rates of key reactions increased substantially, leading to the 5% depletion of stratospheric ozone over the Indian Ocean observed by Evan et al.

- L97: To avoid repeating "eruption", it would be better to say "impacts of Hunga on ozone".
- L97: Although the unique aspects of this study are articulated more clearly in the revised draft than they were initially, I think that it would help to add here something along these lines: "… post-eruption. The goal is not to elucidate the chemical mechanisms giving rise to the observed low ozone, as they were investigated in detail by Evan et al. (2023) and Zhu et al. (2023). Rather, the objectives of the present manuscript can be summarized …".
- L100: traversed by --> obtained within
- L104-106: The sentence "Satellite observations of ozone profiles and columns were exclusively acquired within this region, complementing the ground-based data while offering global coverage and regular monitoring" is problematic. It could be interpreted as saying that the satellites did not make measurements outside of this region, which is not only inaccurate but also potentially confusing since their global nature is mentioned. I suggest instead saying "This study focuses exclusively on satellite measurements acquired in this region."
- L139: using MLS data at level 2 and version 4 (v4) --> using version 4 (v4) MLS level 2 data
- L140-142: The implications of the two different approaches to obtaining instrument pointing information for the MLS data are unclear, and actually this detail is not of much interest for the average reader. It would be better to delete the two sentences devoted to this topic and simply state that the extraordinary enhancement in $H_2O$ from Hunga degraded the accuracy of some of the v5 MLS data products in the first few weeks following the eruption.
- L144-145: For clarity, it would be better to rewrite the first two sentences of this paragraph as: "Following the recommendations of Millán et al. (2022), the MLS profiles for January 2022 are sourced exclusively from level 2 v4 measurements (Livesey et al., 2020). The MLS profiles are categorized as Hunga-influenced or non-influenced using criteria detailed in the next paragraph."
- L148: in any --> on any; to mean --> to the mean
- L152-153: All v5 ozone and water vapor profiles within a 5-degree radius of each of the January 2022 Hunga-influenced profiles were collected, regardless of the satellite's ascending or descending node --> All v5 ozone and water vapor profiles (on both ascending and descending sides of the orbit) within a 5-degree radius of each of the January 2022 Hunga-influenced profiles were collected
- L160-161: Assuming that I have understood correctly, for clarity change "locations showing high water vapor and a negative ozone anomaly" to "locations showing both high water vapor and a negative ozone anomaly".
- L162: 23 January --> 23 January 2022
- L164: Both profile groups --> The two profile groups
- L169-170: Although the authors' response letter makes it clear that v4 and v5 $O_3$ and v5 $H_2O$ data were screened but v4 $H_2O$ data were not screened, the manuscript itself is confusing on this point. First it is stated that "all quality flags … were used on the raw profiles (with the exception of the v4 $H_2O$ profiles)". Then it is stated that "Only the v5 and v4 $O_3$ profiles were screened". These two statements are contradictory.

- L189-190: The statement "the altitude of the tropopause, as estimated by the instrument" implies that the IASI dataset includes a retrieval of tropopause height. Similar statements are made on L271-272 and in the Fig. 1 caption. Is that really the case, or is tropopause height taken from meteorological analyses? Please clarify and amend these statements as needed.
- L194-199: The new paragraph on IASI retrievals requires clarification on several points:
  - The "significant ozone perturbations" were seen in the ozone retrievals from UV-visible instruments, not in ozone itself.
  - spectral ranges of ozone and $SO_2$ do not overlap in the IASI ozone retrieval --> spectral ranges used for ozone and $SO_2$ in the IASI retrieval algorithms do not overlap
  - I do not understand what is meant by "ozone vertical variability" – given IASI's very coarse vertical resolution, it might be better to omit the word "vertical" here.
- L209: top of the atmosphere irradiance --> top-of-the-atmosphere irradiance
- L214 & 215: near-real time --> near-real-time
- L237-238: In my original review I noted that the MLS ozone dataset has been very well validated and used extensively in prior studies. In fact, these data have been central to literally hundreds of scientific studies looking at regions all around the globe, including multiple papers by different groups examining the effects of Hunga on stratospheric ozone. Thus, the skepticism about their validity inherent in the statement "Prior to drawing any conclusions based on the MLS ozone profiles, it is essential to verify their agreement with precise local lidar observations during unperturbed conditions" is completely unwarranted. This language should be moderated. If indeed comparisons between MLS and Maïdo DIAL $O_3$ profiles have not been done previously, as stated in the response letter, then that represents a new contribution whose unique value should be articulated here.
- L239-240: Two points: (1) What does "all recovered profiles" mean? Why "recovered"? This word is used again in L258. (2) The phrases "within a 5-degree region around the lidar site" and "setting the inter-comparison radius to a maximum of 5°" are redundant.
- L240-242: First, these two sentences are also highly redundant and should be merged. Second, "both orbit types" should be "both sides of the orbit". I recommend rewriting these sentences as "We averaged together MLS v5 ozone profiles from both the ascending and the descending sides of the Aura orbit, which have acquisition times near Reunion around 10:15 and 21:45 UTC, respectively. On the other hand, …".
- L253: The statement "$O_{3\,MLS}(z)$ represents the MLS ozone value from averaging kernel at an altitude $z$" makes no sense. $O_{3\,MLS}(z)$ represents the retrieved MLS ozone value. The MLS averaging kernels were (or should have been) applied to the lidar data for this comparison.
- L257: at different layers --> in different layers
- L271: The comma after "retrieval" should be a semicolon.
- L293: assumed to be of 0.02 --> assumed to be about 0.02
- L299: Results show --> Results in Fig. 2b show; both instruments --> the two instruments
- L305: Add a pointer to Fig. 2a after "respectively".
- L315: also increasing the standard variation --> which also increases the standard deviation
- L317-318: Again, the language used here – "MLS appears to be a suitable substitute for lidar data in studying ozone levels" and "… supports the use of MLS data across the region" – gives the impression that the reliability of MLS $O_3$ data for this purpose was in doubt. This wording should be toned down. I suggest at least adding "as expected" to the first phrase and simply

deleting the second one. In fact, the sentence about the representativeness of Reunion data for the Indian Ocean region works better logically without that statement. Also: strong agreement --> good agreement

- L320-326: Why is this discussion of the ozone annual cycle of relevance for this paper? If this information is needed to help interpret any results shown here, then that needs to be made clear; otherwise, this text seems to be a pointless digression.
- L330: The authors state that they compared two datasets, but actually they made two sets of comparisons involving four different datasets altogether: MLS vs DIAL and IASI vs SAOZ.
- L333: The panel titles in Fig. 4 ("MLS & Lidar comparison" and "IASI & SAOZ comparison") give no hint of which way the subtraction goes, nor does the figure caption make it clear. In the text, the results are characterized as "MLS–DIAL" and "SAOZ–IASI". Please clarify whether these differences are "spaceborne" minus "ground-based" data or vice versa; also, if they are not already, make the two sets of differences consistent in terms of direction (i.e., to be parallel with MLS–DIAL, the TCO differences should be taken as IASI–SAOZ, not SAOZ–IASI).
- L340: with higher and --> with larger biases and
- L344: I find this discussion of the relative bias between MLS and DIAL $O_3$ confusing. It is stated that "the bias decreases to 0.24 ± 2.12%" from 40 to 45 km. However, to me it looks like the bias goes from roughly +0.5% at 40 km to nearly –1% at 45 km; that is, the relative bias grows in magnitude but changes sign over this altitude range. I do not see where the quoted value of 0.24% comes from.
- L345: difference and error --> relative bias and standard error
- L347: bias of ozone --> bias in the ozone
- L349-350: Note also that the increased difference and error at altitudes lower than 20 km may be due to the reduced satellite accuracy and precision --> Note also that the increased relative bias and standard error at altitudes below 20 km may be due to the reduced satellite accuracy and precision at those levels
- L353-354: Here again I am confused by the wording of the text. It is stated that "MLS profiles tend to slightly under-estimate ozone concentrations relative to DIAL". But the relative bias shown in Fig. 4a is positive through most of the vertical domain; since the differences were characterized as "MLS–DIAL" on L333, those results indicate that MLS over- (not under-) estimates DIAL concentrations. This needs to be clarified.
- L356-357: The relative dispersion RMSD=3.26% for the IASI/SAOZ comparison is characterized as "very low". But for the MLS/DIAL comparison, RMSD=1.27%. Why was that value described a "low" (L354) while the larger value for IASI/SAOZ is "very low"?
- L360: plume (25-30 km) being --> plumes (25-30 km) that are
- L364: "it appears relevant to use" is very odd wording. I suggest simply saying "we now use".
- L367 & Fig. 5: The figure has been greatly improved. However, it is virtually impossible to see the red contours on the SCO panels (a1-a9), where they could be mistaken for very high DU values. It would be better to simply omit the total $SO_2$ contours from those panels and amend the text in this line accordingly.
- L369: the selection criterion --> the Hunga-influenced selection criterion
- L370-371: I have two comments about "reveal an east-to-west displacement of both plumes … supports previous studies": (1) it's not clear what "both" means here. This word immediately follows "$H_2O$ and ozone anomalies", so the reader naturally associates it with

those two quantities, but the deficit in ozone does not constitute a "plume". I assume that SO$_2$ and H$_2$O are meant. In any case this needs to be clarified, perhaps by saying something along the lines of "Hunga-affected air masses" instead. (2) The east-to-west displacement of the Hunga plume is not a new result "revealed" by Fig. 5. It was reported in several previous studies, including Millán et al. (2022), Khaykin et al. (2022), and others. As the authors noted in their response, the prior studies did not specifically talk about ozone. Nevertheless, they did identify the movement of the Hunga plume, so they should be credited here; the vague allusion to "previous studies" is not sufficient.

- L372: Three points: (1) It would be better not to repeat "rapid" in this line; (2) influence of H$_2$O --> influence of excess H$_2$O; (3) Zhu et al. (2022) could also be cited for the rapid conversion of SO$_2$ to sulfate.
- L374: illustrating --> suggesting
- L375: the Hunga --> Hunga
- L381-382: The last sentence in this paragraph essentially repeats what was said in L372-373. The repetition is confusing since the reader is expecting new information to be conveyed.
- L384: record anomalies of –49.9 ± 4.7 DU were recorded 76.5°E --> a record anomaly of –49.9 ± 4.7 DU was measured at 76.5° E
- L385-386: this IASI SCO anomaly is more than 14 times below the average variability --> the magnitude of this IASI SCO anomaly is more than 14 times larger than the climatological variability
- L386: anomaly map … suggests --> anomaly maps … suggest
- L390: emphasize --> indicate
- L395: Two points: (1) selected by the criterion --> identified by the selection criterion; (2) it would be good here to remind readers what the two groups of Hunga-influenced profiles are.
- L400: by one of --> by each of
- L403: highest --> higher-altitude
- L405: lowest --> lower-altitude
- L406-407: ozone reduction in --> low ozone at
- L407-409: Presumably the ozone anomalies for the two clouds stated in absolute units (ppmv, DU/km) in L404 and L406 are computed from the MLS climatology. It is then a bit jarring to have another set of ozone anomaly values for the two clouds relative to the average lidar profile given in percent terms. This approach precludes easy comparison of the magnitude of the ozone anomalies based on MLS climatology with those based on lidar data. The MLS-climatology-based anomalies should also be quoted in terms of percent. Moreover, it is not clear why the anomalies calculated by differencing the Hunga-influenced MLS profiles and the average lidar profile are emphasized over those based purely on MLS data.
- L408: highest --> higher; this change should be reflected in the Fig. 6 legends as well
- L409: lowest --> lower; this change should be reflected in the Fig. 6 legends as well
- L409: coherent with Evan et al. (2023) who --> consistent with those of Evan et al. (2023), who
- L413: Two points: (1) "confirms previous research" – both Evan et al. (2023) and Zhu et al. (2023) should be explicitly cited here; (2) "the ozone anomaly is linked to a reduction of the

ozone layer": by definition, a negative anomaly is a reduction – what is at issue here is the cause. It would be better to say "the ozone anomaly arose from chemical loss".

- L415-422: Some acknowledgment that the results of this trajectory investigation are further confirmation of the passage of the Hunga plume over Reunion as established by Baron et al. (2023) and Evan et al. (2023) is needed in this paragraph; i.e., those papers should be cited.

- L420: trajectories simulation --> trajectory simulation

- L426-427: As with the Abstract, the authors should bear in mind that many readers will skip most of the detailed discussion in the text and jump straight to the Conclusions. Therefore the Conclusions section needs to do a better job of summarizing the study and identifying its novel aspects. For example, the last sentence of the first paragraph could be amended to better capture the diversity of measurements used: "… using IASI, MLS, and OMPS satellite observations, in conjunction with ground-based measurements from Reunion". The fact that this is the first presentation of IASI data in the context of Hunga should also be emphasized.

- L431: was passing --> passed

- L434: levels --> abundances

- L436: "TCO" and "SCO" should be redefined in the Conclusions or just written out. In addition, I find it strange that the TCO result is considered sufficiently important that it is highlighted in the Conclusions (and the Abstract, as noted above), yet was relegated to an Appendix. I come back to this point below.

- L437: indicated --> indicated that

- L437-440: The final sentences in the manuscript are not well composed. In my opinion they could be rewritten to better convey the message: "Hunga-influenced MLS profiles show a significant reduction in ozone over the 30–12 hPa pressure range. Ozone depletion occurred in two distinct layers, associated with two separate sulfate aerosol clouds. Within the higher-altitude (17.78–12.12 hPa) aerosol cloud, ozone decreased by an average of 0.7 ± 0.5 ppmv (1.1 ± 0.7 DU/km). Within the lower-altitude (31.62–26.10 hPa) aerosol cloud, ozone decreased by an average of 0.6 ± 0.5 ppmv (1.7 ± 1.4 DU/km)."

- L440: The paper ends rather abruptly. In addition to rewriting the last few sentences as suggested above, the authors should consider adding some sort of final sentence to put their results into context. For example, they could say something about how their finding that the observed ozone reduction appeared to be confined within two distinct aerosol layers adds new perspective to the studies that had previously reported chemical ozone loss in the week following the eruption.

- Appendix A: Although I do not disagree that Fig. A1, while helpful, is the sort of ancillary material that belongs in supplementary information rather than the main body of the paper, I am less convinced that that is true of the accompanying text. To me, if a result is sufficiently noteworthy to report in both the Abstract and the Conclusions, then it should be discussed in the paper itself, not just in an Appendix. I was struck by this when I got to the Conclusions and found a number (for the max TCO anomaly) that I had not seen in reading the paper. I feel that the TCO information in this short paragraph should be integrated into the discussion in Section 3.4 (which can still refer to Fig. A1, as it already does now).

- L444: anomalies, both in --> anomalies in both

- L445-448: Clarification is needed in several places in these lines:

- o this IASI TCO anomaly is more than 3 times below the climatological variability --> the magnitude of this IASI TCO anomaly is more than 3 times larger than the climatological variability
- o this anomaly is about 5 times below the variability --> the magnitude of this anomaly is about 5 times larger than the variability
- o When I first read this paragraph, I thought that it contradicted what was stated earlier. If this paragraph is kept in the Appendix (i.e., if the authors choose not to integrate it into the main text as suggested), then to reduce the possibility of confusion, I suggest adding this sentence at the end: "As discussed in Section 3.4 in the main text, the magnitude of the anomaly in SCO exceeds the climatological variability to an even greater degree."
- L459-472: The authors may wish to review the Acknowledgments carefully – there are some typos and missing words.
- L489-490: The second entry for Baron et al. (2023) appears to point to the preprint of a paper that has now been published and thus should be deleted.
- L650-652: As noted earlier, the paper by Sicard et al. has now been published, so the citation needs to be updated.
- L673: The Earth observing system microwave limb sounder (EOS MLS) on the aura Satellite --> The Earth Observing System Microwave Limb Sounder (EOS MLS) on the Aura satellite
- L680: 2018 --> 2022
- Figure 2 caption: between 2003 to 2021 --> between 2003 and 2021
- Figure 3 caption: average … profile from 2013—2021 observations at --> average … profile calculated from observations taken over 2013–2021 at
- Figure 4 caption: The solid red and dashed blue lines in panel (b) should be explained.
- Figure 5 & caption: Two points: (1) the selection criterion --> the Hunga-influenced selection criterion; (2) As noted above, the red $SO_2$ contours should be omitted from panels (a1)-(a9).
- Figure 6 caption: highest and lowest --> higher-altitude and lower-altitude
- Figure 7 caption: for the 23.5 km --> for the trajectory ending at 23.5 km

---

## Referee Report (RR2)

**Third review of "Evidence of a Transient Ozone Depletion Event in the Early Hunga Plume Above the Indian Ocean" by Millet et al.**

The manuscript has once again been substantially revised in response to referee comments, including major changes in the analysis approach (with consequent considerable effects on the magnitudes of the calculated ozone anomalies) and the addition of a coauthor. With these latest changes, the manuscript has again been greatly improved. However, some new issues, mostly instances of unclear wording, have been introduced through the revision process. Thus minor corrections are still needed before the paper can be published.

Specific comments and questions:
Both substantive issues and minor points of clarification, wording suggestions, and grammar / typo corrections are listed together in sequential order through the manuscript. Line numbers refer to the "clean" version of the revised manuscript, not the tracked-changes file.

- L5: while also incorporating --> and also incorporates
- L10: "Revealed" has already been used in this abstract; this word should not be overused. In addition, the term "ozone depletion" is easily misinterpreted. For clarity, I suggest rewriting this sentence as "IASI ozone spatial distributions showed marked decreases in total and stratospheric ozone on that date, with the 5$^{th}$ percentile …".
- L12-13: As currently worded, non-specialist readers could misinterpret this sentence as saying that MLS measures aerosol. Rearranging can alleviate this problem: "A key finding, as shown by MLS profiles, is that the ozone reduction was confined to two distinct layers, each associated with a separate aerosol cloud." Since this is indeed a key finding, it is curious that the authors have chosen not to include any details about the magnitudes of the ozone anomalies in these two layers, whereas this information is provided in the Conclusions.
- L53: more surface for --> more particle surfaces for
- L74-77: This discussion mixes processes occurring over different timescales and is therefore very likely to confuse readers. Evan et al. (2023) and Zhu et al. (2023) talk about the chemical processing and ozone loss that occurred in the Hunga plume within the first week of the eruption. These companion papers should be discussed together. In contrast, the studies by Santee et al. (2023), Wilmouth et al. (2023), and Zhang et al. (2024) focus on perturbations in stratospheric composition observed months after the eruption. The distinction between these two sets of studies should be made more clearly. Moreover, although it is good to mention them for completeness, the studies of the chemical processing in subsequent months are less relevant to this manuscript, which concentrates on the immediate aftermath of the eruption. I suggest re-writing these sentences for clarity. Maybe something along these lines would work: "In this context, Evan et al. (2023) provided evidence of HCl activation on sulfate aerosols within the fresh volcanic plume, and Zhu et al. (2023) elucidated the mechanisms giving rise to the changes observed immediately following the event. (For completeness, we note that comprehensive discussions of the stratospheric chemical processes at work in subsequent months can be found in Wilmouth et al. (2023), Santee et al. (2023), and Zhang et al. (2024).)

- L93: Again, Zhang et al. (2024) is not concerned with the immediate aftermath of the eruption (but rather focuses on the following SH winter, JJA) and does not discuss the same processes as the papers by Evan et al. and Zhu et al. Hence the reference to Zhang et al. (2024) here should be deleted.
- L126: "ozone TCO" is redundant, so delete "ozone".
- L133: "data should be used to study observations --> data should be used to study conditions; within the Hunga plume --> within the fresh Hunga plume for the first few weeks after the eruption
- L136: ozone --> MLS ozone
- L154: due to sedimentation --> due to particle sedimentation
- L161-164: This discussion is not quite correct. Neither v4 nor v5 MLS $H_2O$ measurements should be quality screened for the first ~3 weeks after the eruption. Standard filtering protocols should be applied to the $O_3$ data in both versions, as indicated here, but not to either version of the $H_2O$ data.
- L180: in (Boynard et al., 2018) --> by Boynard et al. (2018)
- L182: to the top of the atmosphere (~60 km): 60 km is not the top of the atmosphere
- L214: near-real time --> near-real-time
- L321: I'm not sure that ACP style will allow the ampersands ("&") in these lines, and in any case I do not think that their meaning is clear. I suggest just using a forward slash instead (e.g., "MLS/DIAL"). Alternatively, "vs" might also work.
- L330-345: I find this discussion a bit confusing. First it is stated that MLS has a relative bias and error with respect to DIAL measurements of 0.11 ± 0.20% in the 20–40 km altitude range. In this case a statement such as "MLS slightly overestimates DIAL in this region" would be appropriate. But then it is stated that over the whole altitude range, the linear regression $y = 1.00\,x$ shows that "MLS profiles tend to slightly over-estimate ozone concentrations relative to DIAL ... irrespective of the altitude". I do not see how the statement "slightly over-estimate" is justified given the value of "1.00" in the linear relationship.
- L347: "an elevated correlation" --> "a fairly strong correlation" (the word "elevated" raises the question "compared to what?")
- L350: altitudes of the Hunga volcanic plume ... that are --> altitudes of the Hunga-affected layers ... that are
- L351: low deviation --> low relative deviation
- L375-376: This wording is unclear. To avoid misinterpretation, it would be better to rewrite this sentence as "IASI recorded the highest number of negative ozone anomalies linked to Hunga on 20 and 21 January (panels (a6)-(a7) and (b6)-(b7) of Figures 5 and A1).
- 378-379: It's possible that I have misunderstood the point here, but to me it seems that the sentence "These values significantly exceed climatological variability" is redundant with "meaning this anomaly is more than three times larger than the typical variation". The first sentence should be either deleted or rewritten to clarify what information it provides that is not covered in the second sentence.
- L400: with respect the --> with respect to the
- L400-405: This discussion is opaque and hard to follow. For one thing, "resp.", used repeatedly in these lines, is not a common abbreviation, and I am not sure what it means

here. I believe that the authors intend to provide percent anomalies for the upper and lower aerosol clouds relative to both the MLS averaged Indian Ocean profile and the mean lidar profile from DIAL, but if so this is a very awkward way to go about doing so. It is also confusing to call an anomaly expressed in terms of percent a "volume mixing ratio anomaly". Finally, panel (e) of Figure 6 is no longer referenced in the text. I think that it would be much clearer to not only rewrite these sentences, but also to rearrange this entire paragraph such that the percent anomalies for each layer are given immediately following their associated absolute anomalies. Assuming that I have understood correctly, I suggest something like: "The ozone mean anomaly associated with the higher-altitude aerosol cloud is (1σ) significant at the 12 hPa level and barely (1σ) significant at the 14 hPa pressure level, with an average anomaly relative to the average background MLS profile of –0.7 ± 0.6 ppmv (–1.0 ± 1.0 DU/km) across these two pressure levels. In percentage terms, this corresponds to –5.5 ± 4.7% and –6.3 ± 4.8% with respect to the average MLS profile over the Indian Ocean (Figure 6e) and the mean lidar profile (Figure 3), respectively. For the lower-altitude aerosol cloud, (1σ) significant ozone anomalies occur across the 21–32 hPa pressure range, with a mean anomaly of –0.6 ± 0.5 ppmv (–1.7 ± 1.4 DU/km), corresponding to –7.5 ± 7.0% and –8.5 ± 8.1% with respect to the mean MLS Indian Ocean and the mean lidar profiles, respectively."

- L417: This construction ("the latter shows") appears to point only to Figure 5. For clarity, this should be rewritten as "… in Figs. 5 and A1; these two figures also show a westward …".
- L421: amounts water --> amounts of water
- L423-425: The way these sentences are written makes it sound like IASI "observations are derived from IASI, MLS, and OMPS satellite data", which makes no sense. This problem can be solved by re-wording / rearranging: "Here we use satellite observations from IASI, MLS, and OMPS, complemented by ground-based measurements from Reunion, to provide a detailed view of the evolution of … Indian Ocean. This study presents the first analysis of IASI data in the context of Hunga."
- L436: exceed --> exceeding
- L438-440: Anomalies expressed in terms of percent will be more meaningful to many readers than the values given here. It would be good to add the corresponding relative anomalies in a manner similar to that suggested above.
- Figure 6 caption: Panels (a-b) presents --> Panels (a-b) present; panels (c-d) shows --> panels (c-d) show; influenced by one of the aerosol clouds --> influenced by the aerosol clouds

---

## Editor Decision (ED1)

**Fourth review of "Evidence of a Transient Ozone Depletion Event in the Early Hunga Plume Above the Indian Ocean" by Millet et al.**

Further revisions to the manuscript have been made in response to referee comments. In addition, an error in the analysis of the MLS/DIAL $O_3$ comparisons was identified and corrected. With these latest changes, the manuscript has again been improved. As before, however, the revisions have introduced a few new instances of unclear wording, as well as some results that I feel are over-interpreted. In addition, a couple of other minor points that escaped my notice last time around were more obvious as I read through the latest draft. Thus I suggest below some additional corrections that should be made before the paper is published.

Specific comments and questions:
Both substantive issues and minor points of clarification, wording suggestions, and grammar / typo corrections are listed together in sequential order through the manuscript. Line numbers refer to the "clean" version of the revised manuscript, not the tracked-changes file.

- L24: Why "indeed" here? I suggest deleting this word.
- L82: As noted previously, the study by Wilmouth et al. (2023) pertains to a different time period than those by Evan et al. (2023) and Zhu et al. (2023). Thus, to avoid confusion, add "also" after "highlighted" in this sentence (alternatively, the whole clause about the Wilmouth paper could be deleted).
- L83: The second part of this sentence ("and the slowing down of the NOx cycle") is also potentially confusing, since of course that effect decreases, not increases, ozone destruction, and the point of this sentence is to describe the mechanisms leading to chemical ozone loss.
- L101-102: The phrase "relies exclusively on satellite measurements from this area" could be mis-interpreted as contradicting the previous sentence stating that ground-based data are used with satellite data. Some rewording / rearrangement would eliminate the ambiguity: "relies on satellite measurements obtained exclusively within this area".
- L136-137: The added text in this sentence (which I realize I suggested) has led to some repetition. To reduce redundancy and use more precise language, I recommend changing "for the first weeks after the eruption" in L136 to "for the first three weeks after the eruption" and then changing "during the first few weeks after the eruption" in L137 to "immediately after the eruption".
- L156-157: This new sentence ("Specifically, …Hunga-influenced") is largely redundant with the original sentence in L159-160 ("As a result, … Hunga-influenced"). Only one of these sentences is needed. If the authors choose to retain the first one, then "occurs" should be "occurred". Also, "considered as" --> "considered to be".
- L164-167: The description of the MLS quality screening is still unclear. For clarity, "for the first three weeks following the eruption" should be added after "with the exception of the v4 and v5 $H_2O$ profiles". (At least, that is what should have been done—the MLS $H_2O$ data outside of the immediate aftermath of the event should have been quality filtered.) On the other hand, this paragraph is about ozone. Therefore, a better approach would be to add "$O_3$" after "raw" in L166 for clarity and then simply delete the parenthetical about the $H_2O$ profiles.

- L183-184: (Boynard et al., 2018) --> Boynard et al. (2018) [i.e., move the parentheses and delete the comma]
- L242: determined --> calculated
- L285: OMPS LP --> OMPS-LP
- L318: I still feel that "MLS appears to be a suitable substitute for lidar data" is too weak (even if the comparisons turned out to be not quite as good as originally thought). I suggest "MLS data are" rather than "MLS appears to be".
- L336: The statement that "the average relative bias decreases" between 40 and 45 km gives the wrong impression. It would be more accurate to say "Between 40 and 45 km, the average relative bias decreases slightly in magnitude but changes sign."
- L336-337: I do not see how the statement "below 20 km, it shows an average of 10.81 ± 38.08 %" can be correct. For one thing, below 20 km the relative bias values are mostly negative. In addition, given that most of the negative relative bias values visible in Fig. 4a have magnitudes of less than 6%, I'm not even sure that "–10%" would be correct, unless the spikes currently cut off at the left-hand edge of the plot are considerably larger than that. If the authors want to quote percentage biases below 20 km, then the x-axis range should be expanded to show these values. However, I would argue that a layer-average bias is not very informative in the face of such large oscillations in the profile. Moreover, I do not believe that this structure is meaningful. Figure 3 shows that the MLS average profile over Reunion is smoothly varying below 20 km. The small wiggles in the DIAL profile are obscured by the thickness of the green and orange lines used for the MLS profiles. The reason for the fairly large relative errors at these altitudes is that the ozone mixing ratios are very small, approaching zero. In this situation—dividing by near-zero values in Eqn. (1)—relative errors become large, exaggerating the discrepancies between the two data sets. The authors state in L342-343 that the increased relative bias below 20 km is attributable to reduced satellite accuracy and precision and a smaller number of available lidar measurements, but, while those factors play a role, I believe that the larger relative biases are mainly due to the very low $O_3$ mixing ratios at these levels. This point needs to be made in the text. It might be more appropriate to cite raw (absolute) rather than relative biases in this region.
- L364: Reminding readers of the selection criteria is helpful. However, to ensure that this does not come across as new information, it would be good to add "As described in Section 2.1.3" at the beginning of this sentence.
- L368: All three of these references should be written as "et al., 202x" [i.e., add commas and delete the parentheses]
- L400-401: With the addition of "with excess water vapor" after "aerosol cloud", the last part of this sentence ("and water vapor excess at the same pressure ranges") is not needed. In fact, I'm not convinced that this sentence is necessary at all, as the details are given in the next paragraph.
- L405: This sentence points to Fig. 6e for the MLS Indian Ocean profile and Fig. 3 for the lidar profile. But isn't the January mean lidar profile also shown in Fig. 6e? If so, then it would be easier on readers to simply refer to Fig. 6e for both mean profiles; that is "… Indian Ocean (Fig. 6e, purple line)" and "… lidar profile (Fig. 6, green line)".
- L408: add a comma after "(2023)"
- L421: delete "also" (not needed with "Additionally")

---

## Author Response (AR2)

**Author's response**

This document provides a point-by-point response to the reviewers including a list of all relevant changes made in the manuscript. All authors sincerely thank each anonymous referee once again for their thorough and detailed reviews, which helped identify flaws and clarify ambiguous aspects of the article.

This document is organized into two sections, each addressing the comments and feedback from a specific referee. The responses to Referee 1 begin on page 2 and those to Referee 2 on page 16.

**Response to Referee 1 Comments**

We would like to once again express our sincere thanks and appreciation to Referee 1 for their thorough and detailed review, as well as for suggesting numerous grammar and typographical corrections. Their comments highlighted clear flaws, which we hope to have addressed in the revised version, as well as areas of ambiguity in the article.

Our responses follow the structure of the review document and are divided into two sections: 1) a response the general comment, and 2) responses to specific comments and questions. Referee comments are written in black and authors answers are in blue.

**General comment:**

**Point 1**: One major comment from my previous review that has not been resolved in the revised draft is the magnitude of the IASI ozone anomaly and its apparent discrepancy with MLS-based estimates. The authors have redone their analysis of MLS measurements and now find average negative anomalies in ozone of $0.7 \pm 0.5$ ppmv at 17-12 hPa and $0.6 \pm 0.5$ ppmv at 26-32 hPa. Unlike in the original manuscript, these anomalies are now barely significant at $1\sigma$. However, the IASI analysis was also redone, and the maximum (and highly significant) TCO anomaly of $40.1 \pm 4.8$ DU is slightly larger than it was before. A stratospheric column ozone (SCO) anomaly is now also calculated; its maximum value (also highly significant) is $49.9 \pm 4.7$ DU. TCO / SCO anomalies of this magnitude will be met with skepticism by many readers. As demonstrated in my previous review, even an anomaly as large as 1 ppmv applied uniformly over the entire range from 40 to 1 hPa (the bulk of the stratospheric ozone layer) would not come close to producing an SCO anomaly of 50 DU. In their response, the authors state that "a direct comparison between IASI total column ozone measurements and MLS stratospheric ozone measurements is not appropriate, as the two instruments sample different atmospheric layers and use distinct observation geometries and techniques". This statement misses the point – ozone is ozone, no matter who is measuring it. For convenience, the plot included in my previous review was based on MLS measurements, but it did not depend on them – the same analysis could be done with any ozone profile. I encourage the authors to do such an exercise themselves – take an ozone profile (from anywhere), compute the SCO from it, and then calculate the SCO anomaly based on perturbations to that ozone profile of different amplitudes. This should give a sense of the magnitude and vertical extent of the perturbation necessary to bring about an SCO anomaly of 50 DU. I feel that some discussion about the credibility of the large column ozone anomalies estimated from IASI data – and their inconsistency with the MLS-based estimates – should be added to the text.

**Response 1**: We appreciate the referee's continued attention to this issue. Following the suggested experiment, we confirm that applying a 1 ppmv anomaly uniformly across the stratospheric column does not produce an SCO anomaly of 50 DU. Such an anomaly is not observed in either MLS or IASI profiles and is roughly twice the maximum anomaly derived from MLS data. This finding prompted a reassessment of our method for computing IASI column anomalies.

To refine our approach, we consulted Anne Boynard, an expert in IASI ozone retrievals and now a co-author of this study. With her guidance, we identified that over-estimated anomalies were partly due to the re-sampling of L2 IASI daily observations onto the regular grid. The issue arose from using nearest-neighbor interpolation instead of averaging, which introduced artifacts. To correct this, we now compute the average of all daily L2 observations over a $1°\times1°$ global grid, aligning with standard

IASI processing procedures.

Additionally, instead of assuming a constant monthly ozone background, we now define background ozone levels as the daily average of IASI/Metop-B data from 2014 to 2021, representing daily means rather than a single January mean. We also apply quality filters, retaining only profiles and columns with more than two degrees of freedom and a retrieval quality filter of 1 to ensure the use of the most reliable observations. Finally, anomalies are now reported only when based on at least three L2 observations.

Figure 1 illustrates the impact of these improvements. The upper panel shows SCO anomalies from the previous approach, while the lower panel presents results from the revised method where at least three observations are available. The figure highlights how the averaging process reduces artifacts.

Furthermore, based on Anne Boynard's input and the referee's earlier comments during the first review, we introduced a regional statistical metric to better capture event-related anomalies while minimizing extreme values. Specifically, we now report the 5th percentile of anomalies, calculated only within the SO2 cloud identified from IASI data. For example, on 21 January, we now report a peak 5th percentile anomaly of -18.6 DU for TCO and at -14.5 DU for SCO.

[Figure]

Figure 1: (**a**) IASI SCO anomaly map for 21 January, computed using nearest-neighbor interpolation instead of averaging.(**b**) IASI SCO anomaly map for 21 January, computed using averaging instead of nearest-neighbor interpolation.)

Previously, Figure 5 of the article displayed only anomalies significant at the 2-sigma level to prevent artifacts. With the refined methodology, we now display all anomalies. In contrast, Figure 1b presents only anomalies based on at least three observations to improve clarity in illustrating the new method.

**Specific comments and questions:**

**Point Abstract**: The authors need to be mindful that many readers will look to the Abstract to get

a basic sense of the paper (and they may not go beyond that). Therefore the Abstract needs to do a much better job of summarizing the study and clearly highlighting its novel aspects. For example, the fact that this is the first presentation of IASI data in the context of the Hunga eruption should be emphasized here. In addition, the finding that the reduction in ozone appears to have been confined to two distinct layers associated with two separate aerosol clouds is one of the few new aspects of this study and should be more clearly articulated.

**Response Abstract**: We thank the referee for the suggested revision. The Abstract has been revised to more clearly emphasize the novel findings of the study.

**Point L5**: Delete "The" in front of "Ozone".

**Response L5**: This point has been addressed.

**Point L9**: Given the poor vertical resolution of IASI data, it is not really appropriate to refer to IASI and MLS "profiles" together in the same sentence. I suggest replacing "profiles" with "measurements" in this line.

**Response L9**: In response to the referee's first point, the abstract has been rewritten: "This study presents the first analysis of Infrared Atmospheric Sounding Interferometer (IASI) ozone data to investigate the impact of the Hunga eruption, while also incorporating Microwave Limb Sounder (MLS) and Ozone Mapping and Profiler Suite Limb Profiler (OMPS-LP) data, as well as ground-based measurements from Reunion."

**Point L11-12**: The TCO result is not actually covered in the main body of the paper, but I think it should be – see my more detailed comments on this point below.

**Response L11-12**: The TCO maps for 15-23 January 2022 were added to the Appendix and corresponding results were integrated into the main body of the paper.

**Point L18**: There is no need to define the acronym "UVR" as it is not used again in the manuscript.

**Response L18**: This point has been addressed.

**Point L34-35**: such as that –> such as those; add a comma after "Calbuco (2015)".

**Response L34-35**: This point has been addressed.s

**Point L44**: Why is the word "implied" used here? Stratospheric ozone losses and radiative changes have been documented following volcanic eruptions, as noted in this manuscript.

**Response L44**: We apologize for the confusion. The word "implied" was not the appropriate word; we intended to convey "resulting". The sentence has been revised to: "Because of the resulting ozone losses and radiative forcing anomalies ...".

**Point L52**: clouds (PSCs) volume –> cloud (PSC) volume

**Response L52**: This point has been addressed.

**Point L71**: An early paper discussing the influence of the excess humidity from Hunga in accelerating conversion of SO2 to sulfate aerosols by Zhu et al. [2022, Comm Earth & Environ, 10.1038/s43247-022-00580-w] should also be cited for this point.

**Response L71**: We thank the referee for highlighting this reference, which has now been added to the manuscript.

**Point L72**: The paper by Sicard et al. has now been published, so the citation needs to be updated both here in the text and in the reference list. Moreover, other papers should also be cited for the Hunga-induced stratospheric cooling, such as those by Sellitto et al. (2022), Coy et al. (2022), and Schoeberl et al. (2022, GRL, 10.1029/2022GL100248).

**Response L72**: The citation has been updated and suggested references were included into the discussion for the Hunga-induced stratospheric cooling.

**Point L75-83**: I do not think that the listing of stratospheric chemical reactions has added useful information to this paper. I understand that in their comments on the previous draft one of the other referees suggesting discussing in more detail the influence of chlorine and sulfur compounds on stratospheric ozone, including showing some chemical reactions. But in response to that comment the authors have simply listed the set of "key heterogeneous reactions" given in the review paper by Solomon et al. (1999), with absolutely no accompanying text to put these reactions into context or give a sense of which ones are generally more important following volcanic eruptions. Zhu et al. (2023) and Evan et al. (2023, in the supplementary material), both already cited in the manuscript, provide a detailed description of the post-Hunga heterogeneous chemical reactions inside and outside the plume. In addition, Wilmouth et al. (2023, PNAS, 10.1073/pnas.2301994120) and Santee et al. (2023, JGR-A, 10.1029/2023JD039169) discuss the stratospheric chemical processing in the months following the eruption. Thus I feel that the authors would be better off deleting the material in these lines and simply referring readers to the lengthy explanations in those previous papers. If the authors want to retain these equations in the paper, then more in depth discussion of how they are relevant needs to be added to the text.

**Response L75-83**: As per the referee's suggestion, we removed the list of stratospheric chemical reactions and now refer to the recommended references instead.

**Point 88-89**: It is not appropriate to say that Evan et al. "documented" a doubling of ozone loss via O3+Cl – they merely reported the results shown by the modeling study of Zhu et al. (2023). Moreover, while the rate of that particular reaction did double, the rates of other reactions changed by even greater amounts, so it is not clear why that one has been singled out. It would be better to make a more general statement that the rates of key reactions increased substantially, leading to the 5% depletion of stratospheric ozone over the Indian Ocean observed by Evan et al.

**Response 88-89**: We thank the referee for this suggested revision. This point has been addressed.

**Point L97**: To avoid repeating "eruption", it would be better to say "impacts of Hunga on ozone".

**Response L97**: This point has been addressed.

**Point L97**: Although the unique aspects of this study are articulated more clearly in the revised draft than they were initially, I think that it would help to add here something along these lines: "... post-eruption. The goal is not to elucidate the chemical mechanisms giving rise to the observed low ozone, as they were investigated in detail by Evan et al. (2023) and Zhu et al. (2023). Rather, the objectives of the present manuscript can be summarized ...".

**Response L97**: We thank the referee for this clarification regarding the objectives, which has been incorporated into the manuscript.

**Point L100**: traversed by –> obtained within.

**Response L100**: This point has been addressed.

**Point L104-106**: The sentence "Satellite observations of ozone profiles and columns were exclusively acquired within this region, complementing the ground-based data while offering global coverage and regular monitoring" is problematic. It could be interpreted as saying that the satellites did not make measurements outside of this region, which is not only inaccurate but also potentially confusing since their global nature is mentioned. I suggest instead saying "This study focuses exclusively on satellite measurements acquired in this region."

**Response L104-106**: This revision has been incorporated into the manuscript for clarity.

**Point L139**: using MLS data at level 2 and version 4 (v4) –> using version 4 (v4) MLS level 2 data.

**Response L139**: This point has been addressed.

**Point L140-142**: The implications of the two different approaches to obtaining instrument pointing information for the MLS data are unclear, and actually this detail is not of much interest for the average reader. It would be better to delete the two sentences devoted to this topic and simply state that the extraordinary enhancement in H2O from Hunga degraded the accuracy of some of the v5 MLS data products in the first few weeks following the eruption.

**Response L140-142**: This point has been addressed.

**Point L144-145**: For clarity, it would be better to rewrite the first two sentences of this paragraph as: "Following the recommendations of Millán et al. (2022), the MLS profiles for January 2022 are sourced exclusively from level 2 v4 measurements (Livesey et al., 2020). The MLS profiles are categorized as Hunga-influenced or non-influenced using criteria detailed in the next paragraph."

**Response L144-145**: This point has been addressed.

**Point L148**: in any –> on any; to mean –> to the mean.

**Response L148**: This point has been addressed.

**Point L152-153**: All v5 ozone and water vapor profiles within a 5-degree radius of each of the January 2022 Hunga-influenced profiles were collected, regardless of the satellite's ascending or descending node –> All v5 ozone and water vapor profiles (on both ascending and descending sides of the orbit) within a 5-degree radius of each of the January 2022 Hunga influenced profiles were collected

**Response L152-153**: This point has been addressed.

**Point L160-161**: Assuming that I have understood correctly, for clarity change 'locations showing high water vapor and a negative ozone anomaly" to "locations showing both high water vapor and a negative ozone anomaly".

**Response L160-161**: This point was correctly understood. However, following Referee 2's suggestions, the criterion has been adjusted to consider only water vapor anomalies, and the corresponding revision has been made.

**Point L162**: 23 January –> 23 January 2022.

**Response L162**: This point has been addressed.

**Point L164**: Both profile groups –> The two profile groups.

**Response L164**: This point has been addressed.

**Point L169-170**: Although the authors' response letter makes it clear that v4 and v5 O3 and v5 H2O data were screened but v4 H2O data were not screened, the manuscript itself is confusing on this point. First it is stated that "all quality flags ... were used on the raw profiles (with the exception of the v4 H2O profiles)". Then it is stated that "Only the v5 and v4 O3 profiles were screened". These two statements are contradictory.

**Response L169-170**: We apologize for the confusion. The last sentence should have read: "Only the v5 O3, v5 H2O, and v4 O3 profiles were screened...", but was omitted to avoid redundancy.

**Point L189-190**: The statement "the altitude of the tropopause, as estimated by the instrument" implies that the IASI dataset includes a retrieval of tropopause height. Similar statements are made on L271-272 and in the Fig. 1 caption. Is that really the case, or is tropopause height taken from meteorological analyses? Please clarify and amend these statements as needed.

**Response L189-190**: The IASI dataset does indeed provide an estimate of tropopause altitude. These sentences have been revised for clarity.

**Point L194-199**: The new paragraph on IASI retrievals requires clarification on several points:

- **Point 1**: The "significant ozone perturbations" were seen in the ozone retrievals from UV-visible instruments, not in ozone itself.

- **Response 1**: This point has been addressed.

- **Point 2**: spectral ranges of ozone and SO2 do not overlap in the IASI ozone retrieval –> spectral ranges used for ozone and SO2 in the IASI retrieval algorithms do not overlap

- **Response 2**: This point has been addressed.

- **Point 3**: I do not understand what is meant by "ozone vertical variability" – given IASI's very coarse vertical resolution, it might be better to omit the word "vertical" here.

- **Response 3**: We omitted the word "vertical" in the revised version of the manuscript.

**Point L209**: top of the atmosphere irradiance –> top-of-the-atmosphere irradiance.

**Response L209**: This point has been addressed.

**Point L214 & 215**: near-real time –> near-real-time.

**Response L214 & 215**: This point has been addressed.

**Point L237-238**: In my original review I noted that the MLS ozone dataset has been very well validated and used extensively in prior studies. In fact, these data have been central to literally hundreds of scientific studies looking at regions all around the globe, including multiple papers by different groups examining the effects of Hunga on stratospheric ozone. Thus, the skepticism about their validity inherent in the statement "Prior to drawing any conclusions based on the MLS ozone profiles, it is

essential to verify their agreement with precise local lidar observations during unperturbed conditions" is completely unwarranted. This language should be moderated. If indeed comparisons between MLS and Maïdo DIAL O3 profiles have not been done previously, as stated in the response letter, then that represents a new contribution whose unique value should be articulated here.

**Response L237-238**: We apologize for the unintended skepticism regarding MLS measurements. The sentence has been revised to address this and to emphasize the first comparison between these two datasets.

**Point L239-240:**: Two points: (1) What does "all recovered profiles" mean? Why "recovered"? This word is used again in L258. (2) The phrases "within a 5-degree region around the lidar site" and "setting the inter-comparison radius to a maximum of 5°" are redundant.

**Response L239-240:**: We recognize "recovered" was an unnecessary and confusing word that we omitted in the revised version. To avoid redundancy, we also deleted the sentence "setting the inter-comparison radius to a maximum of 5°".

**Point L240-242**: First, these two sentences are also highly redundant and should be merged. Second, "both orbit types" should be "both sides of the orbit". I recommend rewriting these sentences as "We averaged together MLS v5 ozone profiles from both the ascending and the descending sides of the Aura orbit, which have acquisition times near Reunion around 10:15 and 21:45 UTC, respectively. On the other hand, ...".

**Response L240-242**: We thank the referee for this suggested revision. This point has been addressed.

**Point L253**: The statement "O3 MLS(z) represents the MLS ozone value from averaging kernel at an altitude z" makes no sense. O3 MLS(z) represents the retrieved MLS ozone value. The MLS averaging kernels were (or should have been) applied to the lidar data for this comparison.

**Response L253**: The MLS averaging kernels were indeed applied to the lidar data. This statement has been revised for clarity.

**Point L257**: at different layers –> in different layers.

**Response L257**: This point has been addressed.

**Point L271**: The comma after "retrieval" should be a semicolon.

**Response L271**: This point has been addressed.

**Point L293**: assumed to be of 0.02 –> assumed to be about 0.02.

**Response L293**: This point has been addressed.

**Point L299**: Results show –> Results in Fig. 2b show; both instruments –> the two instruments.

**Response L299**: This point has been addressed.

**Point L305**: Add a pointer to Fig. 2a after "respectively".

**Response L305**: This point has been addressed.

**Point L315**: also increasing the standard variation –> which also increases the standard deviation.

**Response L315**: This point has been addressed.

**Point L317-318**: Again, the language used here – "MLS appears to be a suitable substitute for lidar data in studying ozone levels" and "... supports the use of MLS data across the region" – gives the impression that the reliability of MLS O3 data for this purpose was in doubt. This wording should be toned down. I suggest at least adding "as expected" to the first phrase and simply deleting the second one. In fact, the sentence about the representativeness of Reunion data for the Indian Ocean region works better logically without that statement. Also: strong agreement –> good agreement.

**Response L317-318**: The suggested changes have been applied to the manuscript. This point has been addressed.

**Point L320-326**: Why is this discussion of the ozone annual cycle of relevance for this paper? If this information is needed to help interpret any results shown here, then that needs to be made clear; otherwise, this text seems to be a pointless digression.

**Response L320-326**: This paragraph has been omitted in the revised version.

**Point L330**: The authors state that they compared two datasets, but actually they made two sets of comparisons involving four different datasets altogether: MLS vs DIAL and IASI vs SAOZ.

**Response L330**: This point has been clarified.

**Point L333**: The panel titles in Fig. 4 ("MLS & Lidar comparison" and "IASI & SAOZ comparison") give no hint of which way the subtraction goes, nor does the figure caption make it clear. In the text, the results are characterized as "MLS–DIAL" and "SAOZ-IASI". Please clarify whether these differences are "spaceborne" minus "ground-based" data or vice versa; also, if they are not already, make the two sets of differences consistent in terms of direction (i.e., to be parallel with MLS–DIAL, the TCO differences should be taken as IASI–SAOZ, not SAOZ–IASI).

**Response L333**: The MLS & Lidar comparison is performed as spaceborne minus ground-based, as specified in Equation (1). This information has been added to both the caption and the text for clarity. However, the comparison between IASI and SAOZ data does not involve any subtraction. Instead, the figure shows IASI TCO as a function of SAOZ TCO, from which statistical quantities are derived. To avoid confusion, we have revised the text in L333 from "... the MLS-DIAL and SAOZ-IASI comparisons ..." to "... the MLS & DIAL and IASI & SAOZ comparisons ...".

**Point L340**: with higher and –> with larger biases and.

**Response L340**: This point has been addressed.

**Point L344**: I find this discussion of the relative bias between MLS and DIAL O3 confusing. It is stated that "the bias decreases to $0.24 \pm 2.12\%$" from 40 to 45 km. However, to me it looks like the bias goes from roughly +0.5% at 40 km to nearly -1% at 45 km; that is, the relative bias grows in magnitude but changes sign over this altitude range. I do not see where the quoted value of 0.24% comes from.

**Response L344**: We apologize for the confusion. The value $0.24 \pm 2.12\%$ referred to the average relative bias between 40 and 45 km. This point has been revised for clarity.

**Point L345**: difference and error –> relative bias and standard error.

**Response L345**: This point has been addressed.

**Point L347**: bias of ozone –> bias in the ozone.

**Response L347**: This point has been addressed.

**Point L349-350**: Note also that the increased difference and error at altitudes lower than 20 km may be due to the reduced satellite accuracy and precision –> Note also that the increased relative bias and standard error at altitudes below 20 km may be due to the reduced satellite accuracy and precision at those levels.

**Response L349-350**: This point has been addressed.

**Point L353-354**: Here again I am confused by the wording of the text. It is stated that "MLS profiles tend to slightly under-estimate ozone concentrations relative to DIAL". But the relative bias shown in Fig. 4a is positive through most of the vertical domain; since the differences were characterized as "MLS–DIAL" on L333, those results indicate that MLS over- (not under-) estimates DIAL concentrations. This needs to be clarified.

**Response L353-354**: Prompted by the referee's comment, we have re-calculated the linear regression between MLS and lidar data. The updated regression equation ($y = 1.00x$, previously $y = 0.99x$) is very slightly greater than 1.00 but rounds down, indicating a minor over-estimation of MLS, consistent with the observed mean bias profile (as opposed to a minor under-estimation with the previous value). This clarification has been incorporated into the text.

**Point L356-357**: The relative dispersion RMSD=3.26% for the IASI/SAOZ comparison is characterized as "very low". But for the MLS/DIAL comparison, RMSD=1.27%. Why was that value described a "low" (L354) while the larger value for IASI/SAOZ is "very low"?

**Response L356-357**: We apologize for the wording issue. The text has been revised to describe the relative dispersion for IASI/SAOZ as "low" and for MLS/DIAL as "very low.

**Point L360**: plume (25-30 km) being –> plumes (25-30 km) that are.

**Response L360**: This point has been addressed.

**Point L364**: "it appears relevant to use" is very odd wording. I suggest simply saying "we now use".

**Response L364**: This point has been addressed.

**Point L367  Fig. 5**: The figure has been greatly improved. However, it is virtually impossible to see the red contours on the SCO panels (a1-a9), where they could be mistaken for very high DU values. It would be better to simply omit the total SO2 contours from those panels and amend the text in this line accordingly.

**Response L367  Fig. 5**: Total SO2 contours were omitted from panels a1-a9.

**Point L369**: the selection criterion –> the Hunga-influenced selection criterion.

**Response L369**: This point has been addressed.

**Point L370-371**: I have two comments about "reveal an east-to-west displacement of both plumes ... supports previous studies": (1) it's not clear what "both" means here. This word immediately follows

"H2O and ozone anomalies", so the reader naturally associates it with those two quantities, but the deficit in ozone does not constitute a "plume". I assume that SO2 and H2O are meant. In any case this needs to be clarified, perhaps by saying something along the lines of "Hunga-affected air masses" instead. (2) The east-to-west displacement of the Hunga plume is not a new result "revealed" by Fig. 5. It was reported in several previous studies, including Millán et al. (2022), Khaykin et al. (2022), and others. As the authors noted in their response, the prior studies did not specifically talk about ozone. Nevertheless, they did identify the movement of the Hunga plume, so they should be credited here; the vague allusion to "previous studies" is not sufficient.

**Response L370-371**: The suggested wording has been implemented, and appropriate references have been added.

**Point L372**: Three points: (1) It would be better not to repeat "rapid" in this line; (2) influence of H2O –> influence of excess H2O; (3) Zhu et al. (2022) could also be cited for the rapid conversion of SO2 to sulfate.

**Response L372**: This point has been addressed.

**Point L374**: illustrating –> suggesting.

**Response L374**: This point has been addressed.

**Point L375**: the Hunga –> Hunga.

**Response L375**: This point has been addressed.

**Point L381-382**: The last sentence in this paragraph essentially repeats what was said in L372-373. The repetition is confusing since the reader is expecting new information to be conveyed.

**Response L381-382**: The repetitive sentence has been removed.

**Point L384**: record anomalies of -49.9 $\pm$ 4.7 DU were recorded 76.5°E –> a record anomaly of -49.9 $\pm$ 4.7 DU was measured at 76.5°E.

**Response L384**: This point has been addressed.

**Point L385-386**: this IASI SCO anomaly is more than 14 times below the average variability –> the magnitude of this IASI SCO anomaly is more than 14 times larger than the climatological variability.

**Response L385-386**: This point has been addressed.

**Point L386**: anomaly map ... suggests –> anomaly maps ... suggest.

**Response L386**: This point has been addressed.

**Point L390**: emphasize –> indicate.

**Response L390**: This point has been addressed.

**Point L395**: Two points: (1) selected by the criterion –> identified by the selection criterion; (2) it would be good here to remind readers what the two groups of Hunga-influenced profiles are.

**Response L395**: This point has been addressed.

**Point L400**: by one of –> by each of.

**Response L400**: This point has been addressed.

**Point L403**: highest –> higher-altitude.

**Response L403**: This point has been addressed.

**Point L405**: lowest –> lower-altitude.

**Response L405**: This point has been addressed.

**Point L406-407**: ozone reduction in –> low ozone at.

**Response L406-407**: This point has been addressed.

**Point L407-409**: Presumably the ozone anomalies for the two clouds stated in absolute units (ppmv, DU/km) in L404 and L406 are computed from the MLS climatology. It is then a bit jarring to have another set of ozone anomaly values for the two clouds relative to the average lidar profile given in percent terms. This approach precludes easy comparison of the magnitude of the ozone anomalies based on MLS climatology with those based on lidar data. The MLS-climatology-based anomalies should also be quoted in terms of percent. Moreover, it is not clear why the anomalies calculated by differencing the Hunga-influenced MLS profiles and the average lidar profile are emphasized over those based purely on MLS data.

**Response L407-409**: The ozone anomalies for the two clouds are indeed computed from the MLS climatology, and we have clarified this in the text. To facilitate comparison, we now also express the MLS-climatology-based anomalies in percent terms. This ensures that lidar-based and MLS-based anomalies can be directly compared, rather than relying solely on the lidar-derived values.

**Point L408**: highest –> higher; this change should be reflected in the Fig. 6 legends as well.

**Response L408**: This point has been addressed.

**Point L409**: lowest –> lower; this change should be reflected in the Fig. 6 legends as well.

**Response L409**: This point has been addressed.

**Point L409**: coherent with Evan et al. (2023) who –> consistent with those of Evan et al. (2023), who.

**Response L409**: This point has been addressed.

**Point L413**: Two points: (1) "confirms previous research" – both Evan et al. (2023) and Zhu et al. (2023) should be explicitly cited here; (2) "the ozone anomaly is linked to a reduction of the ozone layer": by definition, a negative anomaly is a reduction – what is at issue here is the cause. It would be better to say "the ozone anomaly arose from chemical loss".

**Response L413**: We thank the referee for the suggested revision. This point has been addressed.

**Point L415-422**: Some acknowledgment that the results of this trajectory investigation are further confirmation of the passage of the Hunga plume over Reunion as established by Baron et al. (2023) and Evan et al. (2023) is needed in this paragraph; i.e., those papers should be cited.

**Response L415-422**: This point has been addressed.

**Point L420**: trajectories simulation –> trajectory simulation.

**Response L420**: This point has been addressed.

**Point L426-427**: As with the Abstract, the authors should bear in mind that many readers will skip most of the detailed discussion in the text and jump straight to the Conclusions. Therefore the Conclusions section needs to do a better job of summarizing the study and identifying its novel aspects. For example, the last sentence of the first paragraph could be amended to better capture the diversity of measurements used: "... using IASI, MLS, and OMPS satellite observations, in conjunction with ground-based measurements from Reunion". The fact that this is the first presentation of IASI data in the context of Hunga should also be emphasized.

**Response L426-427**: We thank the referee for the suggested revision. The Conclusions have been revised to better summarize the main results of the study.

**Point L431**: was passing –> passed.

**Response L431**: This point has been addressed.

**Point L434**: levels –> abundances.

**Response L434**: This point has been addressed.

**Point L436**: "TCO" and "SCO" should be redefined in the Conclusions or just written out. In addition, I find it strange that the TCO result is considered sufficiently important that it is highlighted in the Conclusions (and the Abstract, as noted above), yet was relegated to an Appendix. I come back to this point below.

**Response L436**: "TCO" and "SCO" have been redefined in the Conclusions, and the TCO results have been integrated into the main body of the article.

**Point L437**: indicated –> indicated that.

**Response L437**: This point has been addressed.

**Point L437-440**: The final sentences in the manuscript are not well composed. In my opinion they could be rewritten to better convey the message: "Hunga-influenced MLS profiles show a significant reduction in ozone over the 30-12 hPa pressure range. Ozone depletion occurred in two distinct layers, associated with two separate sulfate aerosol clouds. Within the higher altitude (17.78-12.12 hPa) aerosol cloud, ozone decreased by an average of $0.7 \pm 0.5$ ppmv ($1.1 \pm 0.7$ DU/km). Within the lower-altitude (31.62-26.10 hPa) aerosol cloud, ozone decreased by an average of $0.6 \pm 0.5$ ppmv ($1.7 \pm 1.4$ DU/km)."

**Response L437-440**: We thank the referee for the suggested revision which has been incorporated into the Conclusions.

**Point L440**: The paper ends rather abruptly. In addition to rewriting the last few sentences as suggested above, the authors should consider adding some sort of final sentence to put their results into context. For example, they could say something about how their finding that the observed ozone reduction appeared to be confined within two distinct aerosol layers adds new perspective to the studies that had previously reported chemical ozone loss in the week following the eruption.

**Response L440**: This point has been addressed.

**Point Appendix A**: Although I do not disagree that Fig. A1, while helpful, is the sort of ancillary material that belongs in supplementary information rather than the main body of the paper, I am less convinced that that is true of the accompanying text. To me, if a result is sufficiently noteworthy to report in both the Abstract and the Conclusions, then it should be discussed in the paper itself, not just in an Appendix. I was struck by this when I got to the Conclusions and found a number (for the max TCO anomaly) that I had not seen in reading the paper. I feel that the TCO information in this short paragraph should be integrated into the discussion in Section 3.4 (which can still refer to Fig. A1, as it already does now).

**Response Appendix A**: Information regarding total ozone was integrated into the discussion in Section 3.4.

**Point L444**: anomalies, both in –> anomalies in both.

**Response L444**: This point has been addressed.

**Point L445-448**: Clarification is needed in several places in these lines:

- **Point 1**: this IASI TCO anomaly is more than 3 times below the climatological variability –> the magnitude of this IASI TCO anomaly is more than 3 times larger than the climatological variability.

- **Response 1**: This point has been addressed.

- **Point 2**: this anomaly is about 5 times below the variability –> the magnitude of this anomaly is about 5 times larger than the variability

- **Response 2**: This point has been addressed.

- **Point 3**: When I first read this paragraph, I thought that it contradicted what was stated earlier. If this paragraph is kept in the Appendix (i.e., if the authors choose not to integrate it into the main text as suggested), then to reduce the possibility of confusion, I suggest adding this sentence at the end: "As discussed in Section 3.4 in the main text, the magnitude of the anomaly in SCO exceeds the climatological variability to an even greater degree."

- **Response 3**: This point has been addressed.

**Point L459-472**: The authors may wish to review the Acknowledgments carefully – there are some typos and missing words.

**Response L459-472**: This point has been addressed.

**Point L489-490**: The second entry for Baron et al. (2023) appears to point to the preprint of a paper that has now been published and thus should be deleted.

**Response L489-490**: This entry does not cite a publication but instead refers to the dataset from Baron (2023), which corresponds to aerosol lidar observations at Maïdo.

**Point L650-652**: As noted earlier, the paper by Sicard et al. has now been published, so the citation needs to be updated.

**Response L650-652**: This citation has been updated.

**Point L673**: The Earth observing system microwave limb sounder (EOS MLS) on the aura Satellite –> The Earth Observing System Microwave Limb Sounder (EOS MLS) on the Aura satellite.

**Response L673**: This point has been addressed.

**Point L680**: 2018 –> 2022.

**Response L680**: This point has been addressed.

**Point Figure 2 caption**: between 2003 to 2021 –> between 2003 and 2021.

**Response Figure 2 caption**: This point has been addressed.

**Point Figure 3 caption**: average ... profile from 2013-2021 observations at –> average ... profile calculated from observations taken over 2013-2021 at.

**Response Figure 3 caption**: This point has been addressed.

**Point Figure 4 caption**: The solid red and dashed blue lines in panel (b) should be explained.

**Response Figure 4 caption**: The solid red and dashed blue lines represent the linear regression line and the 1:1 line, respectively. This information was added in the caption of Figure 4.

**Point Figure 5 & caption**: Two points: (1) the selection criterion –> the Hunga-influenced selection criterion; (2) As noted above, the red SO2 contours should be omitted from panels (a1)-(a9).

**Response Figure 5 & caption**: This point has been addressed.

**Point Figure 6 caption**: highest and lowest –> higher-altitude and lower-altitude.

**Response Figure 6 caption**: This point has been addressed.

**Point Figure 7 caption**: for the 23.5 km –> for the trajectory ending at 23.5 km.

**Response Figure 7 caption**: This point has been addressed.

**Response to Referee 2 Comments**

We would like to once again express our thanks and appreciation to Referee 2 for their review. The comments identified flaws and unclear points in the article, providing an excellent opportunity to improve its overall quality.

Our responses follow the structure of the review document and are divided into two sections: 1) response to the main comment, and 2) responses to minor comments. Referee comments are written in black and authors answers are in blue.

**Main comment:**

**Point 1**: The criterion for diagnosing Hunga-influenced profiles includes the presence of positive $H_2O$ anomalies and negative ozone anomalies at 25 or 28 km. However, the authors then use these same profiles—selected based on the presence of negative ozone anomalies at the certain levels—to argue that negative ozone anomalies exist at these levels, which are caused by aerosol clouds. This constitutes a circular argument because the negative ozone anomalies are both part of the diagnostic criterion and used as evidence to support the claim. This reasoning is not appropriate, as it relies on the defined criterion to prove the relationship. To strengthen this argument, the authors need to provide additional evidence linking the negative ozone anomalies to aerosol clouds. For example, the authors could demonstrate that these negative ozone anomalies spatially overlap with the regions of aerosol clouds at 25 or 28 km. That said, I agree that the negative ozone anomalies are related to the Hunga event. In Figure 6, the co-occurrence of positive $H_2O$ anomalies and negative ozone anomalies at the same levels supports this connection. However, the specific relationship with aerosol clouds remains unsubstantiated and requires further confirmation. At the very least, the authors should weaken their claims about this linkage to better align with the evidence presented.

**Response 1**: We appreciate the referee's insight regarding the circular reasoning in our initial criterion. Based on the studies of Legras et al. (2022) and Schoeberl et al. (2022), we have refined our approach to avoid this issue. These studies show that the Hunga aerosol and water vapor plumes initially coincided before diverging due to aerosol sedimentation. Legras et al. (2022) further indicate that during the first phase of sedimentation (until about 20 February), water vapor followed the descending aerosol. Given that our study period (15-23 January 2022) falls within this phase, we assume that the water vapor and aerosol clouds overlap. To eliminate circular reasoning, we now base our criterion solely on MLS water vapor profiles. Originally, we identified Hunga-influenced profiles by selecting those with negative ozone anomalies and water vapor mixing ratios exceeding 100 ppmv in the 10-100 hPa range, yielding 72 profiles. We have revised this by selecting only profiles where water vapor exceeds 100 ppmv specifically at the 26.10 hPa and 14.67 hPa pressure levels (corresponding to ∼25 km and ∼28 km, the estimated altitudes of the plumes). This refinement reduces the number of selected profiles to 47 (26 at 14.67 hPa and 21 at 26.10 hPa), particularly decreasing the number of profiles linked to the higher-altitude aerosol cloud. Although the results remain significant, this adjustment reduces the significance of the ozone anomaly associated with the higher aerosol cloud.

**Minor comments:**

**Point 1**: Line 54: The phrase "offering more surface for halogen-ozone reactions" is inaccurate because reactions involving ozone do not require surface area. Consider rephrasing it to "heterogeneous reactions".

**Response 1**: This point has been addressed.

**Point 2**: Paragraph starting from Line 63 (Introduction): In the discussions on heterogeneous reactions, Zhang et al. (2024; https://doi.org/10.1029/2024GL108649) should be cited, which provides a detailed analysis of heterogeneous reactions triggered by the Hunga event. Similarly, in the section discussing gas-phase processes, Wilmouth et al. (2023; https://doi.org/10.1073/pnas.230199412) should be included, which highlights the importance of water vapor on chemistry.

**Response 2**: We appreciate the referee's suggestion and have incorporated these references into the manuscript as requested.

**Point 3**: In Figure 5b: It is surprising to see many data points exceeding 2-sigma uncertainty, with some even showing significantly positive anomalies.

**Response 3**: Initially, the significance of the anomaly was determined based on the uncertainty of the daily observations, which sometimes led to anomalies exceeding the 2-sigma uncertainty due to day-to-day variability. However, with the assistance of Anne Boynard, a new co-author of this study, we have refined the processing of IASI maps to align with standard IASI procedures. This includes applying data quality filters that retain only profiles and columns with more than two degrees of freedom and a retrieval quality filter of 1, ensuring that only the most reliable observations are used in our analysis. More importantly, anomalies are now computed by averaging nearby points rather than using nearest-neighbor interpolation. Consequently, we now display all anomalies for both TCO and SCO, rather than only significant ones, and high positive anomalies occur less frequently.

**Point 4**: Line 414, "This observation confirms previous research and indicates that the ozone anomaly is linked to a reduction of the ozone layer." The logic of the statement is unclear.

**Response 4**: This statement has been changed to: "This observation confirms previous research (Evan et al., 2023; Zhu et al., 2023) and indicates that the ozone anomaly arose from chemical loss."

---

## Author Response (AR3)

**Author's response**

This document provides a point-by-point response to the reviewers including a list of all relevant changes made in the manuscript. All authors sincerely thank each anonymous referee once more for their helpful reviews.

This document is organized into three sections. First, we would like to draw the referees' attention to updated results concerning the DIAL/MLS comparison. The last two sections address the comments and feedback from a specific referee. The responses to Referee 1 begin on page 3 and those to Referee 2 on page 7.

**Updated result**

An important point worth mentioning is a flaw in our previous comparison between DIAL and MLS data. Shortly after the last re-submission, we realized that the comparison mistakenly involved DIAL data with applied averaging kernels being compared to the original DIAL data itself. This explains why the mean bias profile appeared close to 0 across most of the altitude range. After correcting this error and properly comparing the DIAL data (with averaging kernels applied) to the MLS profiles, we obtained updated results (see Figure 1). As expected, the agreement is now slightly weaker, but the mean bias in the altitude range of interest (20–40 km) remains low at 2.76±1.40 %, and should not affect any of the MLS-based conclusions presented in the article.

[Figure]

Figure 1: (Left) DIAL/MLS mean bias profile included in the previous version and (Right) updated DIAL/MLS mean bias profile.

**Response to Referee 1 Comments**

Once again, we would like to express our sincere thanks to the referee for their suggested re-wordings and clarifications. Their comments helped identify important ambiguities, which we have aimed to address in the revised version.

Referee comments are written in black and authors answers are in blue.

**Specific comments and questions:**

**Point L5**: while also incorporating –> and also incorporates

**Response L5**: This point has been addressed.

**Point L10**: "Revealed" has already been used in this abstract; this word should not be overused. In addition, the term "ozone depletion" is easily misinterpreted. For clarity, I suggest rewriting this sentence as "IASI ozone spatial distributions showed marked decreases in total and stratospheric ozone on that date, with the 5th percentile...".

**Response L10**: This point has been addressed.

**Point L12-13**: As currently worded, non-specialist readers could misinterpret this sentence as saying that MLS measures aerosol. Rearranging can alleviate this problem: "A key finding, as shown by MLS profiles, is that the ozone reduction was confined to two distinct layers, each associated with a separate aerosol cloud." Since this is indeed a key finding, it is curious that the authors have chosen not to include any details about the magnitudes of the ozone anomalies in these two layers, whereas this information is provided in the Conclusions.

**Response L12-13**: We have re-worded the message as suggested and added information regarding the magnitude of ozone anomalies.

**Point L53**: more surface for –> more particle surfaces for

**Response L53**: This point has been addressed.

**Point L74-77**: This discussion mixes processes occurring over different timescales and is therefore very likely to confuse readers. Evan et al. (2023) and Zhu et al. (2023) talk about the chemical processing and ozone loss that occurred in the Hunga plume within the first week of the eruption. These companion papers should be discussed together. In contrast, the studies by Santee et al. (2023), Wilmouth et al. (2023), and Zhang et al. (2024) focus on perturbations in stratospheric composition observed months after the eruption. The distinct on between these two sets of studies should be made more clearly. Moreover, although it is good to mention them for completeness, the studies of the chemical processing in subsequent months are less relevant to this manuscript, which concentrates on the immediate aftermath of the eruption. I suggest re-writing these sentences for clarity. Maybe something along these lines would work: "In this context, Evan et al. (2023) provided evidence of HCl activation on sulfate aerosols within the fresh volcanic plume, and Zhu et al. (2023) elucidated the mechanisms giving rise to the changes observed immediately following the event. (For completeness, we note that comprehensive discussions of the stratospheric chemical processes at

work in subsequent months can be found in Wilmouth et al. (2023), Santee et al. (2023), and Zhang et al. (2024).)

**Response L74-77**: This point has been addressed.

**Point L93**: Again, Zhang et al. (2024) is not concerned with the immediate aftermath of the eruption (but rather focuses on the following SH winter, JJA) and does not discuss the same processes as the papers by Evan et al. and Zhu et al. Hence the reference to Zhang et al. (2024) here should be deleted.

**Response L93**: This point has been addressed.

**Point L126**: "ozone TCO" is redundant, so delete "ozone".

**Response L126**: This point has been addressed.

**Point L133**: "data should be used to study observations –> data should be used to study conditions; within the Hunga plume –> within the fresh Hunga plume for the first few weeks after the eruption

**Response L133**: This point has been addressed.

**Point L136**: ozone –> MLS ozone

**Response L136**: This point has been addressed.

**Point L154**: due to sedimentation –> due to particle sedimentation

**Response L154**: This point has been addressed.

**Point L161-164**: This discussion is not quite correct. Neither v4 nor v5 MLS H2O measurements should be quality screened for the first ∼3 weeks a er the eruption. Standard filtering protocols should be applied to the O3 data in both versions, as indicated here, but not to either version of the H2O data.

**Response L161-164**: We apologize for the confusion. Filtering criteria were indeed not applied to either the v4 or v5 MLS H2O profiles. This has been clarified in the manuscript.

**Point L180**: in (Boynard et al., 2018) –> by Boynard et al. (2018)

**Response L180**: This point has been addressed.

**Point L182**: to the top of the atmosphere (∼60 km): 60 km is not the top of the atmosphere

**Response L182**: This point has been addressed.

**Point L214**: near-real time –> near-real-time

**Response L214**: This point has been addressed.

**Point L321**: I'm not sure that ACP style will allow the ampersands ("&") in these lines, and in any case I do not think that their meaning is clear. I suggest just using a forward slash instead (e.g., "MLS/DIAL"). Alternatively, "vs" might also work.

**Response L321**: We have replaced the ampersands with the forward slash, both in the text and within

**Point L330-345**: I find this discussion a bit confusing. First it is stated that MLS has a relative bias and error with respect to DIAL measurements of 0.11 ± 0.20% in the 20–40 km altitude range. In this case a statement such as "MLS slightly overestimates DIAL in this region" would be appropriate. But then it is stated that over the whole altitude range, the linear regression y = 1.00 x shows that "MLS profiles tend to slightly over-estimate ozone concentrations relative to DIAL ... irrespective of the altitude". I do not see how the statement "slightly overestimate" is justified given the value of "1.00" in the linear relationship.

**Response L330-345**: As mentioned in our comment on the second page of this document, the comparison between DIAL and MLS data was updated, and the corresponding results were revised accordingly. The updated results should now eliminate any confusion, as both the mean relative bias profile and the linear regression (y = 1.02x) clearly indicate a slight over-estimation by MLS relative to the DIAL data.

**Point L347**: "an elevated correlation" –> "a fairly strong correlation" (the word "elevated" raises the question "compared to what?")

**Response L347**: This point has been addressed.

**Point L350**: altitudes of the Hunga volcanic plume ... that are –> altitudes of the Hunga-affected layers ... that are

**Response L350**: This point has been addressed.

**Point L351**: low deviation –> low relative deviation

**Response L351**: This point has been addressed.

**Point L375-376**: This wording is unclear. To avoid misinterpretation, it would be better to rewrite this sentence as "IASI recorded the highest number of negative ozone anomalies linked to Hunga on 20 and 21 January (panels (a6)-(a7) and (b6)-(b7) of Figures 5 and A1).

**Response L375-376**: This point has been addressed.

**Point L182**: It's possible that I have misunderstood the point here, but to me it seems that the sentence "These values significantly exceed climatological variability" is redundant with "meaning this anomaly is more than three times larger than the typical variation". The first sentence should be either deleted or rewritten to clarify what information it provides that is not covered in the second sentence.

**Response L182**: We have deleted the first sentence to avoid repetition, as suggested.

**Point L400**: with respect the –> with respect to the

**Response L400**: This point has been addressed.

**Point L400-405**: This discussion is opaque and hard to follow. For one thing, "resp.", used repeatedly in these lines, is not a common abbreviation, and I am not sure what it means here. I believe that the authors intend to provide percent anomalies for the upper and lower aerosol clouds relative to both the MLS averaged Indian Ocean profile and the mean lidar profile from DIAL, but if so this is a very awkward way to go about doing so. It is also confusing to call an anomaly expressed in terms of

percent a "volume mixing ratio anomaly". Finally, panel (e) of Figure 6 is no longer referenced in the text. I think that it would be much clearer to not only rewrite these sentences, but also to rearrange this entire paragraph such that the percent anomalies for each layer are given immediately following their associated absolute anomalies. Assuming that I have understood correctly, I suggest something like: "The ozone mean anomaly associated with the higher-altitude aerosol cloud is ($1\sigma$) significant at the 12 hPa level and barely ($1\sigma$) significant at the 14 hPa pressure level, with an average anomaly relative to the average background MLS profile of -0.7 $\pm$ 0.6 ppmv (-1.0 $\pm$ 1.0 DU/km) across these two pressure levels. In percentage terms, this corresponds to –5.5 $\pm$ 4.7% and -6.3 $\pm$ 4.8% with respect to the average MLS profile over the Indian Ocean (Figure 6e) and the mean lidar profile (Figure 3), respectively. For the lower-altitude aerosol cloud, ($1\pm$) significant ozone anomalies occur across the 21-32 hPa pressure range, with a mean anomaly of -0.6 $\pm$ 0.5 ppmv (-1.7 $\pm$ 1.4 DU/km), corresponding to (-7.5 $\pm$ 7.0% and -8.5 $\pm$ 8.1% with respect to the mean MLS Indian Ocean and the mean lidar profiles, respectively."

**Response L400-405**: We thank the referee for this suggested clarification. This point has been addressed.

**Point L417**: This construction ("the latter shows") appears to point only to Figure 5. For clarity, this should be rewritten as "... in Figs. 5 and A1; these two figures also show a westward ...".

**Response L417**: This point has been addressed.

**Point L421**: amounts water –> amounts of water

**Response L421**: This point has been addressed.

**Point L423-425**: The way these sentences are written makes it sound like IASI "observations are derived from IASI, MLS, and OMPS satellite data", which makes no sense. This problem can be solved by re-wording / rearranging: "Here we use satellite observations from IASI, MLS, and OMPS, complemented by ground-based measurements from Reunion, to provide a detailed view of the evolution of ... Indian Ocean. This study presents the first analysis of IASI data in the context of Hunga."

**Response L423-425**: This point has been addressed.

**Point L436**: exceed –> exceeding

**Response L436**: This point has been addressed.

**Point L438-440**: Anomalies expressed in terms of percent will be more meaningful to many readers than the values given here. It would be good to add the corresponding relative anomalies in a manner similar to that suggested above.

**Response L438-440**: This point has been addressed.

**Point Figure 6 caption**: Panels (a-b) presents –> Panels (a-b) present; panels (c-d) shows –> panels (c-d) show; influenced by one of the aerosol clouds –> influenced by the aerosol clouds

**Response Figure 6 caption**: This point has been addressed.

**Response to Referee 2 Comments**

We wish to express our sincere thanks to the referee for their suggested clarifications.

Referee comments are written in black and authors answers are in blue.

**Specific comments and questions:**

**Point 1**: Throughout the paper–including the abstract, introduction, and conclusion–the authors emphasize that "the ozone loss happened in two layers of aerosol clouds" (e.g., line 12), implying that aerosols were the primary driver of the observed ozone loss. However, Zhu et al. (2023) attributed the ozone anomaly to a combination of processes, with aerosols being a contributing but not dominant factor. Therefore, the use of "associated with aerosol clouds" may overstate the role of aerosols. I suggest replacing "aerosol clouds" with a broader term such as "Hunga plume" or "aerosol clouds with excess water vapor" to avoid potential misinterpretation.

**Response 1**: We acknowledge the referee's comment that the discussed impact on ozone results from the combined effects of water vapor and sulfate aerosols, rather than aerosols alone. To avoid overstating the role of aerosols, we have replaced most occurrences of "aerosol clouds" with the suggested terms.

**Point L27**: The paragraph discussing tropospheric ozone seems less relevant to the main focus of the paper. This is more of a comment than a request to remove it, as I respect the authors' writing style.

**Response L27**: This point has been addressed.

**Point L33**: For improved clarity, I recommend revising "global chemistry" to "global atmospheric chemistry".

**Response L33**: This point has been addressed.

**Point L40**: Consider simplifying "heterogeneous chemical reactions" to the more commonly used "heterogeneous reactions".

**Response L40**: This point has been addressed.

**Point L333**: The statement "Between 40 and 45 km, the average relative bias increases to 0.24 $\pm$ 2.12 %" seems inconsistent with Figure 4a, where the relative bias values between 40 and 45 km appear to be all above 0.24%. Please double-check this value for accuracy.

**Response L333**: As mentioned in our comment on the second page of this document, the comparison between DIAL and MLS data was updated, and the corresponding results were revised accordingly. The updated results should now eliminate any confusion.

**Point L360**: It would be helpful to briefly remind readers of the detail of the "Hunga-influenced selection criterion" at this point in the text or in the caption of Figure 5 for easier reference and clarity.

**Response L360**: As suggested, we have included a reminder detailing the selection criterion. Additionally, we added a clarification at its first mention (Lines 156–157 of the updated manuscript) to improve its initial presentation.

---

## Author Response (AR5)

**Author's response**

This document provides a point-by-point response to the reviewer including a list of all relevant changes made in the manuscript. All authors sincerely thank the anonymous referee once more for their helpful review.

**Response to Referee 1 Comments**

We would like to express our sincere thanks to the referee for their suggested re-wordings and clarifications. Referee comments are written in black and authors answers are in blue.

**Specific comments and questions:**

**Point L24**: Why "indeed" here? I suggest deleting this word.

**Response L24**: This point has been addressed.

**Point L82**: As noted previously, the study by Wilmouth et al. (2023) pertains to a different time period than those by Evan et al. (2023) and Zhu et al. (2023). Thus, to avoid confusion, add "also" after "highlighted" in this sentence (alternatively, the whole clause about the Wilmouth paper could be deleted).

**Response L82**: This point has been addressed.

**Point L83**: The second part of this sentence ("and the slowing down of the NOx cycle") is also potentially confusing, since of course that effect decreases, not increases, ozone destruction, and the point of this sentence is to describe the mechanisms leading to chemical ozone loss.

**Response L83**: Additional information on catalytic cycles was added to the manuscript to improve clarity. As noted by Zhang et al. (2024), the slowdown of the NOx cycle led to enhanced ClOx levels and subsequent ozone depletion.

**Point L101-102**: The phrase "relies exclusively on satellite measurements from this area" could be mis-interpreted as contradicting the previous sentence stating that ground-based data are used with satellite data. Some rewording / rearrangement would eliminate the ambiguity: "relies on satellite measurements obtained exclusively within this area".

**Response L-101-102**: This point has been addressed.

**Point L136-137**: The added text in this sentence (which I realize I suggested) has led to some repetition. To reduce redundancy and use more precise language, I recommend changing "for the first weeks after the eruption" in L136 to "for the first three weeks after the eruption" and then changing "during the first few weeks after the eruption" in L137 to "immediately after the eruption".

**Response L136-137**: This point has been addressed.

**Point L156-157**: This new sentence ("Specifically, ...Hunga-influenced") is largely redundant with the original sentence in L159-160 ("As a result, ... Hunga-influenced"). Only one of these sentences is needed. If the authors choose to retain the first one, then "occurs" should be "occurred". Also, "considered as" –> "considered to be".

**Response L156-157**: This point has been addressed.

**Point L164-167**: The description of the MLS quality screening is still unclear. For clarity, "for the first three weeks following the eruption" should be added after "with the exception of the v4 and v5 H2O

profiles". (At least, that is what should have been done-the MLS H2O data outside of the immediate aftermath of the event should have been quality filtered.) On the other hand, this paragraph is about ozone. Therefore, a better approach would be to add "O3" after "raw" in L166 for clarity and then simply delete the parenthetical about the H2O profiles.

**Response L164-167**: This point has been addressed.

**Point L183-184**: (Boynard et al., 2018) –> Boynard et al. (2018) [i.e., move the parentheses and delete the comma].

**Response L183-184**: This point has been addressed.

**Point L242**: determined –> calculated

**Response L242**: This point has been addressed.

**Point L285**: OMPS LP –> OMPS-LP

**Response L285**: This point has been addressed.

**Point L318**: I still feel that "MLS appears to be a suitable substitute for lidar data" is too weak (even if the comparisons turned out to be not quite as good as originally thought). I suggest "MLS data are" rather than "MLS appears to be".

**Response L318**: This point has been addressed.

**Point L336**: The statement that "the average relative bias decreases" between 40 and 45 km gives the wrong impression. It would be more accurate to say "Between 40 and 45 km, the average relative bias decreases slightly in magnitude but changes sign."

**Response L336**: This point has been addressed.

**Point L336-337**: I do not see how the statement "below 20 km, it shows an average of 10.81 ± 38.08 %" can be correct. For one thing, below 20 km the relative bias values are mostly negative. In addition, given that most of the negative relative bias values visible in Fig. 4a have magnitudes of less than 6 %, I'm not even sure that "-10 %" would be correct, unless the spikes currently cut off at the left-hand edge of the plot are considerably larger than that. If the authors want to quote percentage biases below 20 km, then the x-axis range should be expanded to show these values. However, I would argue that a layer-average bias is not very informative in the face of such large oscillations in the profile. Moreover, I do not believe that this structure is meaningful. Figure 3 shows that the MLS average profile over Reunion is smoothly varying below 20 km. The small wiggles in the DIAL profile are obscured by the thickness of the green and orange lines used for the MLS profiles. The reason for the fairly large relative errors at these altitudes is that the ozone mixing ratios are very small, approaching zero. In this situation–dividing by near-zero values in Eqn. (1)–relative errors become large, exaggerating the discrepancies between the two data sets. The authors state in L342-343 that the increased relative bias below 20 km is attributable to reduced satellite accuracy and precision and a smaller number of available lidar measurements, but, while those factors play a role, I believe that the larger relative biases are mainly due to the very low O3 mixing ratios at these levels. This point needs to be made in the text. It might be more appropriate to cite raw (absolute) rather than relative biases in this region.

**Response L336-337**: The bias values below 20 km were incorrect and have been corrected. During

the revision, we also found that the standard error of the mean bias was over-estimated by a factor of two; this has now been fixed. As suggested, the x-axis range was adjusted, and information on low ozone mixing ratios has been added.

**Point L**: Reminding readers of the selection criteria is helpful. However, to ensure that this does not come across as new information, it would be good to add "As described in Section 2.1.3" at the beginning of this sentence.

**Response L**: This point has been addressed.

**Point L368**: All three of these references should be written as "et al., 202x" [i.e., add commas and delete the parentheses]

**Response L368**: This point has been addressed.

**Point L400-401**: With the addition of "with excess water vapor" after "aerosol cloud", the last part of this sentence ("and water vapor excess at the same pressure ranges") is not needed. In fact, I'm not convinced that this sentence is necessary at all, as the details are given in the next paragraph.

**Response L400-401**: This point has been addressed and the sentence has been omitted.

**Point L405**: This sentence points to Fig. 6e for the MLS Indian Ocean profile and Fig. 3 for the lidar profile. But isn't the January mean lidar profile also shown in Fig. 6e? If so, then it would be easier on readers to simply refer to Fig. 6e for both mean profiles; that is "... Indian Ocean (Fig. 6e, purple line)" and "... lidar profile (Fig. 6, green line)".

**Response L405**: The MLS Indian Ocean profile and the lidar profile are indeed both present in Fig. 6e. The text has been simplified as suggested.

**Point L408**: add a comma after "(2023)"

**Response L408**: This point has been addressed.

**Point L421**: delete "also" (not needed with "Additionally")

**Response L421**: This point has been addressed.